



# Benchmarking and improving algorithms for attributing satellite-observed contrails to flights

Aaron Sarna[1], Vincent Meijer[2], Rémi Chevallier[3], Allie Duncan[1], Kyle McConnaughay[4], Scott Geraedts[1], and Kevin McCloskey[1]

[1]Google Research, Mountain View, CA, USA
[2]Faculty of Aerospace Engineering, Delft University of Technology, Delft, the Netherlands
[3]Federation ENAC ISAE SUPAERO ONERA, Université de Toulouse, France
[4]formerly at Google Research, Mountain View, CA, USA

**Correspondence:** Aaron Sarna (sarna@google.com)

**Abstract.** Contrail cirrus clouds persisting in ice-supersaturated air cause a substantial fraction of aviation's climate impact. One proposed method for the mitigation of this impact involves modifying flight paths to avoid particular regions of the atmosphere that are conducive to the formation of persistent contrails. Ascertaining which flight formed each observed contrail can be used to assess and improve contrail forecast models, as well as study the effectiveness of performing contrail avoidance. The problem of contrail-to-flight attribution is complicated by several factors, such as the time required for a contrail to become visible in satellite imagery, high air traffic densities and errors in wind data. Recent work has introduced automated algorithms for solving the attribution problem, but lack an evaluation against ground-truth data. In this work, we present a method for producing synthetic contrail observations with predetermined contrail-to-flight attributions which can be used to evaluate – or "benchmark" – and improve such attribution algorithms. The resulting performance metrics can be used to understand the implications of using this observational data in downstream tasks such as forecast model evaluation and analysis of contrail avoidance trials. We also introduce a novel, highly-scalable, contrail-to-flight attribution algorithm that leverages the characteristic compounding of error induced by simulating contrail advection using numerical weather models. The benchmark shows an improvement of about 30% in precision versus previous contrail-to-flight attribution algorithms, without compromising recall.

## 1 Introduction

Condensation trails (contrails) are the linear ice clouds that trail behind an aircraft as a result of the warm, moist engine exhaust mixing with colder, drier ambient air (Schumann, 1996). When the ambient air is sufficiently humid (i.e. supersaturated with respect to ice), these contrails can persist for several hours (Minnis et al., 1998). They perturb the Earth's energy budget by reflecting incoming solar radiation and reducing outgoing longwave radiation (Meerkötter et al., 1999). The net effect of all persistent contrails is estimated to be warming and of a magnitude comparable to that by aviation-emitted $CO_2$ (Lee et al., 2021).





(a)  (b)  (c)  (d)

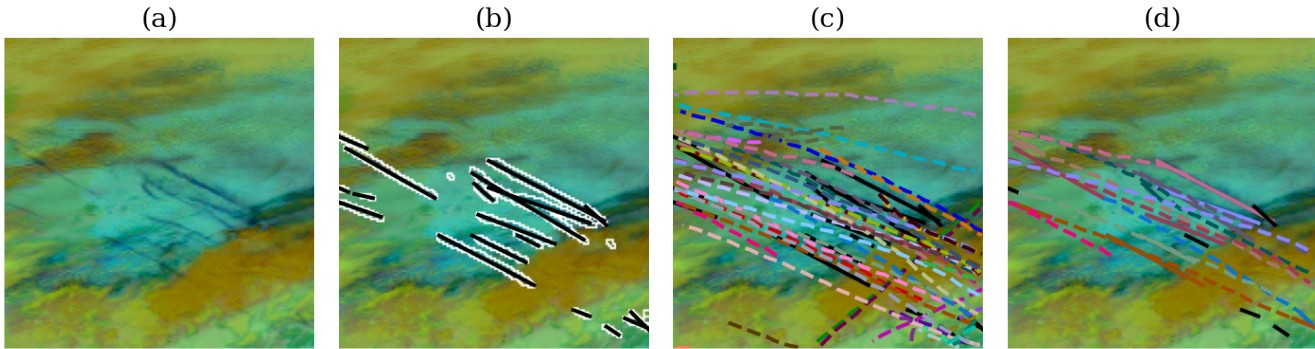

**Figure 1.** A high level visualization of a generic contrail-to-flight attribution process. All panes show a portion of a GOES-16 ABI image from 16:40 UTC on 6 May, 2019 over Ontario, Canada, rendered using the Ash color-scheme to map infrared brightness temperatures to the visible spectrum. In **(a)** we see just the image, with some contrails visible in dark blue and some other clouds in yellow and brown partially obscuring some of the contrails. In **(b)** we show the result of running an automated contrail detector on the image, with the detected contrail pixels outlined in white, and the results of linearizing the detector outputs as black line segments. Notably, some contrails appear segmented up due to the occlusion from other clouds. In **(c)** we take all flight paths that passed nearby in the preceding 2 hours and simulate their advection to the capture time of the GOES image. This estimates the expected location of a hypothetical contrail that each flight formed. Each advected flight is shown in a unique color, while the linearized contrails are still in black. Note that there is not a perfect alignment between observed contrails and flights, and in some cases there appear to be many candidate matches, while in others there appear to be none. In **(d)** we show the results of a contrail-to-flight attribution. Contrails that have been attributed are now colored the same as the flight they were attributed to, and only those flights are shown. Contrails in black were not attributed to a flight. The attributed flights are not always what appeared in **(c)** to be the best match, since the attribution algorithm can take into account additional signals, like temporal dynamics.

There exist several mitigation options for the climate impact of contrail cirrus, such as the use of alternative fuels (Voigt et al., 2021; Märkl et al., 2024) and trajectory modifications (Mannstein et al., 2005; Teoh et al., 2020; Frias et al., 2024). The latter approach, referred to as contrail avoidance, is motivated by the fact that ice-supersaturated regions (ISSRs) are relatively
rare and vertically thin (Gierens et al., 1999; Gierens and Spichtinger, 2000; Spichtinger et al., 2003), such that only a subset of aircraft trajectories would require minor modifications in order to achieve a major reduction in climate impact. Although such trajectory changes may lead to additional fuel burn and concomitant climate impacts, several simulation studies (Teoh et al., 2020; Frias et al., 2024; Borella et al., 2024) have assessed this trade-off and conclude that this is a cost-effective mitigation strategy. However, all these studies do assume that the required forecasts are perfect, whereas recent work has found that
existing forecast methods do not achieve this standard (Gierens et al., 2020; Geraedts et al., 2024; Meijer, 2024). Incorrectly forecasting a contrail formation region could lead to aircraft burning extra fuel for avoidance without actually reducing contrail climate impact. In the extreme, a sufficiently poor forecast model used for avoidance could even increase contrail climate impact. It is therefore imperative that the performance of existing forecasting approaches is characterized in detail, including the expected implications of their usage in modifying aircraft trajectories with the objective of reducing their warming impact
from both $CO_2$ and non-$CO_2$ sources.





One approach to the evaluation of these forecasts is to compare them with observations of contrails. Such observations can be obtained by means of in situ and remote sensing (Schumann et al., 2017). In situ measurements can provide detailed information on the microphysical contents of a contrail, but are expensive to obtain and therefore limited in number. On the other hand, remote sensing measurements provide less detailed information on any particular contrail but are widely available.

One example of such observations is the recognition of contrails in infrared satellite imagery. The automated detection of contrails in low-Earth orbit (LEO) satellite imagery has been carried out over the past two decades using variants of the computer vision algorithm introduced by Mannstein et al. (1999), but with less success when applied to geostationary satellite imagery of coarser resolution (Mannstein et al., 2010; Graf et al., 2012). Recent advances in image processing through deep learning and the introduction of higher resolution geostationary imagers such as the GOES-16 ABI has lead to the availability

of increasing amounts of contrail detection data (McCloskey et al., 2021; Meijer et al., 2022; Ng et al., 2023; Chevallier et al., 2023) due to the large spatial coverage and high temporal resolution of these instruments. Geostationary satellites are, however, limited in their ability to observe a contrail during its formation and identify the aircraft that produced it. Such information can be valuable for studying the relation between observed contrail properties and aircraft types (Gryspeerdt et al., 2024), validating contrail forecasts (Geraedts et al., 2024), and assessing the effectiveness of mitigation options. The lack of this

information is primarily due to the time required for a formed contrail to grow wide and/or optically thick enough to become visible in a particular instrument's imagery, during which the contrail advects away from the location it was formed. Although this advection process can be simulated using wind data obtained from numerical weather prediction (NWP) models, small errors in the prescribed wind values can lead to increasingly large errors in the simulated location of the contrail. In areas with sufficiently many flights, these errors significantly complicate the process of attributing observed contrails to flights.

Geostationary satellites are also limited in their abilities to infer the altitude of a detected contrail, which further inhibits their direct utility for evaluating contrail forecasts, especially if the forecasted contrail-formation regions are avoided by vertical deviations of the flight path (Teoh et al., 2020). Knowledge of the aircraft that formed a particular contrail can provide an estimate of the altitude of the contrail and also of the region that should be avoided.

Several approaches have been developed to address the problem of attributing contrails observed in satellite imagery to
flights. All of them to some degree follow the approach visualized in Figure 1: contrails visible in geostationary imagery (Figure 1(a)) are detected and often then individually transformed into a representative line segment (Figure 1(b)), joined with flight tracks advected with NWP weather data (Figure 1(c)), and finally contrail detections are attributed to flights using some form of cost optimization algorithm (Figure 1(d)). Duda et al. (2004) present a case-study of several widespread persistent contrails appearing over the Great Lakes region in the United States in 2000, as observed using imagery from the GOES-8

satellite. They advect flight tracks using NWP wind data to the satellite image capture times and compute the distance between these tracks and the observed contrails: the flight track with the minimum average distance to a given contrail for all applicable time steps was determined to have formed that contrail. Chevallier et al. (2023) introduce an algorithm that simultaneously tracks contrail instance masks over successive satellite images and attributes these to the flight that formed them, and apply this in a case study that considers 8 hours of satellite data over the continental United States. Their approach also relies on

advecting flight tracks using NWP wind data, and no evaluation of its performance is included. Geraedts et al. (2024) compare





advected flight tracks to linearized contrail detections in satellite imagery, and consider not only the distance between a flight track and a contrail, but also their relative orientation and the time between observation and aircraft passage. The performance of their automated approach is evaluated by comparing the algorithm's proposed attributions to those determined manually by three of the study's authors on 1000 randomly selected flight segments. Lastly, Gryspeerdt et al. (2024) associates detections

of the same contrail in successive satellite images by using NWP wind data, and then attributes the tracked contrail to a flight based on relative distance and orientation.

Benchmarking these attribution algorithms is complicated by the lack of ground-truth data. As discussed, the moment of formation of a particular contrail is not observed in geostationary satellite imagery. A ground-truth dataset for these attribution algorithms therefore requires observing the moment of formation using some higher-resolution instrument, possibly a

ground-based camera, and following the contrail until it becomes observable in the satellite imagery of interest. At the time of writing, no such dataset is available. Even if such a dataset were available, the metrics used to evaluate the performance of a contrail-to-flight attribution algorithm and their implications for downstream usage of the algorithm output data are relatively under-explored. For example, an attribution algorithm that is conservative in the amount of contrail-to-flight attributions in assigns by prioritizing quality over quantity may be suitable for the construction of a dataset for comparing a contrail forecast

model to satellite observations. However, such an algorithm would perhaps be less suitable for the evaluation of a large-scale contrail avoidance experiment using satellite imagery. Additionally, one attribution algorithm may perform better in certain circumstances (such as high air traffic density), which could further motivate choosing a particular approach over others.

In addition to providing observational data for the evaluation and improvement of contrail forecasts, automated contrail-to-flight attributions can be utilized in evaluating the effectiveness of contrail mitigation approaches such as contrail avoidance.

The potential of contrail avoidance has been assessed by modifying real-world flight trajectories in multiple experiments, or "trials". In 2021, the German Aerospace Center (DLR) and EUROCONTROL deviated a total of 212 flights in the airspace managed by the Maastricht Upper Area Control (MUAC) for the purposes of contrail avoidance (Sausen et al., 2023). The effect of these deviations was studied by means of an automated contrail detection algorithm (Mannstein et al., 1999, 2010) applied to Meteosat SEVIRI imagery. The trial did not attempt to attribute individual observed contrails to flights and instead

quantified only the presence or absence of observable contrails in particular sectors of the MUAC airspace. As a consequence, this required all flights in a particular sector to adhere to the requested deviation, limiting the total number of usable data points for the evaluation of the experiment. Although the MUAC/DLR trial lead to a statistically significant reduction in the number of observed contrails for days where flights performed contrail avoidance, Sausen et al. (2023) emphasize the need for better forecast and evaluation methods. In 2023, a partnership of American Airlines, Google, and Breakthrough Energy

also performed an avoidance feasibility trial involving 22 round-trip flights (Sonabend et al., Forthcoming). The evaluation, in this case, did involve using satellite imagery to determine whether each individual flight in the trial formed contrails, but the process was entirely manual and time-consuming, making scaling this approach infeasible. The use of automated contrail-to-flight attribution in the context of such trials (as opposed to manual labeling) has the potential downside of increasing the number of flights that need to be included in the trial in order to achieve statistical significance, due to increased measurement




noise from an algorithm compared to a human expert, and thereby increases the costs of running such a trial. Quantifying and reducing that measurement noise of such systems would counteract this effect, requiring fewer flights and lower costs.

There has additionally been recent interest in establishing Monitoring, Reporting, and Validation (MRV) systems for contrail climate impact, at the air-space, national, or continental levels. One recent example is the proposal for an MRV system for non-$CO_2$ effects of aviation in the European Union (Council of European Union, 2024). Among the goals of these systems are to monitor the contrail impact of each airline and encourage its reduction. For any such implementation, there will be a need both for assessing the quality of contrail forecasts as well as for accurate and scalable methods that can retrospectively determine contrail formation on a per-flight basis.

We thus conclude that there are several relevant applications of satellite-observed contrails that are attributed to the flights that formed them, but that this potential has not yet fully realized due to the limited scalability of manual approaches and the absence of a benchmark for the evaluation of their automated counterparts. For these reasons, this study introduces a new, scalable attribution algorithm, named "CoAtSaC," and a large-scale benchmark dataset of synthetic contrail detections with predetermined flight attributions, named "SynthOpenContrails." We show that this newly introduced dataset, in combination with a set of performance metrics, allows one to both benchmark and improve the performance of an attribution algorithm. The large size of the dataset allows for verifying the scalability of a particular attribution algorithm, as well as study its performance in differing conditions such as air traffic density, time of day, and time of year. Lastly, we show that our novel attribution algorithm provides a substantial improvement when compared to existing approaches evaluated on the new benchmark.

## 2 Methods

In this section we present a novel contrail-to-flight attribution algorithm as well as a dataset of synthetic contrail detections. We show how the dataset can help tune the algorithm, and then be used to compare the performance of different attribution algorithms.

### 2.1 Notation

In Section 2.2 and 2.3 we present a contrail-to-flight attribution algorithm and a method for producing a benchmark dataset of synthetic contrail detections with known flight attributions. Each of these includes a number of parameters that can be adjusted as needed to prioritize performance in different scenarios for the attribution algorithm and to generate a synthetic dataset that simulates different satellites and detection algorithms. These parameters are denoted by $C$, with a subscript, for coefficients, and $T$, with a subscript, for thresholds. The specific values of these parameters used in this study are provided in Table A4 for the CoAtSaC algorithm and Table A2 for the SynthOpenContrails dataset.

### 2.2 Contrail-to-Flight Attribution Algorithm

In this section we present a novel algorithm for attributing contrails to the flights that created them. We call this algorithm "CoAtSaC," short for "Contrail Attribution Sample Consensus".





### 2.2.1 Data

We obtain our contrail detections by running the contrail detection algorithm used in Ng et al. (2023) on infrared imagery from
the GOES-16 Advanced Baseline Imager's (ABI) Full Disk product (Goodman et al., 2019), which covers much of the western
hemisphere with approximately 2 km spatial resolution and scans every 10 minutes. The algorithm uses a convolutional neural
network to produce a prediction that each satellite pixel contains a contrail and thresholds the results to produce a binary mask.
It then fits line segments to the individual contrails in the mask. For consistency with Geraedts et al. (2024), we use the same
spatial region covering roughly the contiguous United States (specified in Appendix A1), and the same time periods between
April 4, 2019 and April 4, 2020, divided up the same way into several time spans between 4 and 22 hours long sampled
throughout the year, aiming to capture seasonal, day-of-week, and diurnal effects on contrail formation, requiring a minimum
of 36 hours of separation between time spans to ensure no overlap of flights or contrails between time spans. We divide these
time spans up into train, validation, and test splits. During algorithm development and tuning (see Section 2.4.3) we primarily
used the train split, but monitor performance on the validation split to ensure we do not "overfit" to the specific conditions
present in the train split. Only when computing the final results, as shown in Section 3 do we use the test split, which verifies
that the algorithm generalizes to previously unseen data. The specific time spans in each split can be found in Appendix A28.
We use flight trajectories provided by FlightAware (https://flightaware.com). This includes a mixture of Automatic Depen-
dent Surveillance-Broadcast (ADS-B) data received by ground-based stations and Aerion satellites (Garcia et al., 2015). For the
purposes of contrail attribution it is critical to recognize that this data is incomplete, since it may lack data on particular flights
as operators may request their data to be obfuscated or excluded. The implication is that there may be detectable contrails
formed by flights that are missing from our data and we need to be careful not to incorrectly attribute them to a different flight
that appears to be the best match among the flights in the data. We also make no assumptions about what fraction of flights are
missing or whether they are in some way biased with respect to likelihood of persistent contrail formation. We apply the same
filtering and preprocessing of flight data as in Geraedts et al. (2024), to filter out erroneous waypoints and those that could
not have formed contrails, and to achieve a uniform frequency of waypoints across all flights. For each time span of contrail
detections we load flight data starting 2 hours before the start of the span and ending at the end of the span, in order to account
for the delay between contrail formation and detection due to the 2 km spatial resolution of the GOES-16 ABI (Geraedts et al.,
2024).

The weather data we use comes from the European Centre for Medium-Range Weather Forecasts (ECMWF). For our attri-
bution algorithm we use the ARCO-ERA5 dataset (Carver and Merose, 2023), which is derived from the ERA5 reanalysis data
(Hersbach et al., 2020). This is hourly data at 0.25 degree resolution at 137 model levels separated by roughly 10 hPa.

For the purposes of our contrail attribution approach, we need to answer the question: for each flight waypoint, "where would
we expect a hypothetical contrail formed by the given aircraft at this waypoint to appear in a given satellite scan?" To answer
this, we simulate the advection of each waypoint to each of the subsequent 11 GOES-16 ABI Full Disk images (roughly 2 hours
at 10 minute intervals – see Appendix A2 for the implications of only advecting for 2 hours). We note that the GOES-16 ABI
does not capture the Full Disk scan instantaneously at the nominal scan-time, but rather it captures it as 22 west-to-east swaths,



starting in the north and moving south over the course of 10 minutes (see Appendix A3). This approach can be generalized to other geostationary satellites, as they have similar scan patterns (Okuyama et al., 2015). We therefore compute a "scantime-offset" for each flight waypoint, based on when its location would be captured by the GOES-16 ABI relative to the nominal scan start time (Meijer et al., 2024a). The set of target times for our advection is then the nominal scantimes of the 11 scans, with the scantime-offset added. A small amount of error is introduced by the fact that the scantime-offset is not updated as the

waypoint advects; if it advects across a swath boundary, the scantime offset would jump by roughly 30 seconds. The advection itself is performed exactly the same way as in Geraedts et al. (2024), which we detail in Appendix A4.

This approach to simulating flight advection is subject to a number of sources of error, including (but not limited to) inaccuracies in the interpolated weather data, approximations in sedimentation rate, and not accounting for all physical processes that can affect the vertical location of the contrail (e.g. radiative heating). We expect that these errors will compound over time, and

as a result our estimation of where a hypothetical contrail would appear in a given satellite image will be increasingly wrong as the hypothetical contrail grows older, with a strong correlation between the errors in successive satellite images.

Once all flights are advected, we will have advected flights and detected contrails at each satellite frame starting 2 hours before the initially defined time span, and ending 2 hours after. This is to ensure that the attribution algorithm can consider flights and contrail detections that are near the beginning and end of the time span in the context of their temporal dynamics.

However, the benchmark only evaluates the attributions that are within the original time span.

### 2.2.2 Single-Frame Attribution Algorithm from Geraedts et al. (2024)

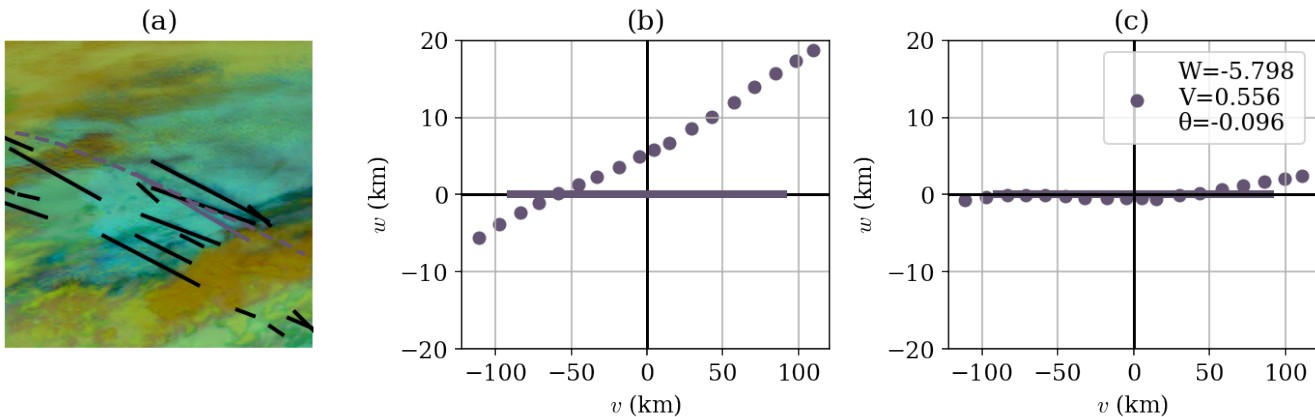

**Figure 2.** A visualization of the single frame matching process. This is the same scene as in Figure 1, but focusing in on a single flight and a single contrail, shown in purple in **(a)**. In **(b)** we show the same data on the $v - w$ plane, with the linear contrail defining the $v$-axis and the flight waypoints projected accordingly to points $(w_i, v_i)$. **(c)** shows the results of applying the transformation in Equation 1 after optimizing the parameters $W$, $V$, and $\theta$ in Equation 2, producing points $(\hat{w}_i, \hat{v}_i)$.





CoAtSaC is an extension of the single-frame attribution algorithm from Geraedts et al. (2024), with minor modifications, specified in Appendix A5. We summarize the Geraedts et al. (2024) algorithm here.

The algorithm starts with the definition of a new 2D coordinate system, which is an orthographic projection centered on a linearized detected contrail, with the contrail centered along the $v$ axis and the $w$ axis orthogonal to it. Distances along each axis are specified in kilometers. Advected waypoints of a single flight are parallax-corrected to the geostationary satellite's perspective and projected onto this plane to coordinates $(w_i, v_i)$. We compute the set of "overlapping" waypoints as those whose $v_i$ values are within the span of the contrail, with a small additional tolerance. Non-overlapping waypoints are excluded. An example of projecting a flight and linearized contrail to this space is shown in Figure 2(b).

In this projection, we aim to quantify the "agreement" between the advected flight and the detected contrail in terms of their relative orientation and distance. This can be thought of as measuring the implied advection error if this flight had formed this contrail, which we quantify in terms of the coordinate transformation

$$\hat{w}_i \rightarrow (w_i + W)\cos(\theta) + (v_i + V)\sin(\theta) \tag{1}$$
$$\hat{v}_i \rightarrow (v_i + V)\cos(\theta) - (w_i + W)\sin(\theta),$$

The parameters $W$ and $V$ are translations along the respective axes and $\theta$ is a rotation. These parameters are optimized by minimizing the objective function

$$S_{\text{attr}} = \underbrace{C_{\text{fit}}\frac{1}{N}\sum_{i=1}^{N}\hat{w}_i^2}_{\text{fit term}} + \underbrace{C_{\text{shift}}(V^2 + W^2) + C_{\text{angle}}(1 - \cos(\theta))}_{\text{regularization terms}} + \underbrace{C_{\text{age}}}_{\text{constant term}}. \tag{2}$$

which essentially tries to move the flight waypoints as close to the contrail (i.e. $v$-axis) as possible, subject to regularization terms. Coefficients $C_{\text{fit}}$, $C_{\text{shift}}$, $C_{\text{angle}}$, and $C_{\text{age}}$ vary with age to allow for a higher tolerance for advection error for flights that have advected longer. The result of running the optimization in Equation 2, is visualized in Figure 2(c), showing both the transformed flight waypoints and the optimized parameters of the transformation. If the minimized value of $S_{\text{attr}}$ is below a threshold of 3 then the associated flight is deemed to have formed the associated contrail, other than some additional logic to help resolve multiple flights being attributed to the same contrail, which can be found in Section 2.2 of Geraedts et al. (2024).

This approach has a few shortcomings that we aim to improve upon. Firstly, because the winds can be very different at different altitudes, a flight that is at a substantially different altitude than the contrail will likely advect at a different speed and might advect directly above or below the contrail but, in the two-dimensional projection, look perfectly aligned in a single frame. Such a flight could erroneously be attributed over the true flight that formed the contrail, which would advect along with the contrail, but might incur some error along the way. Secondly, it treats the advection error for each flight waypoint as independent between satellite frames, when in reality it is highly correlated. We aim to rectify these by leveraging the correlations between advection errors for the same flight segment as it advects over time.




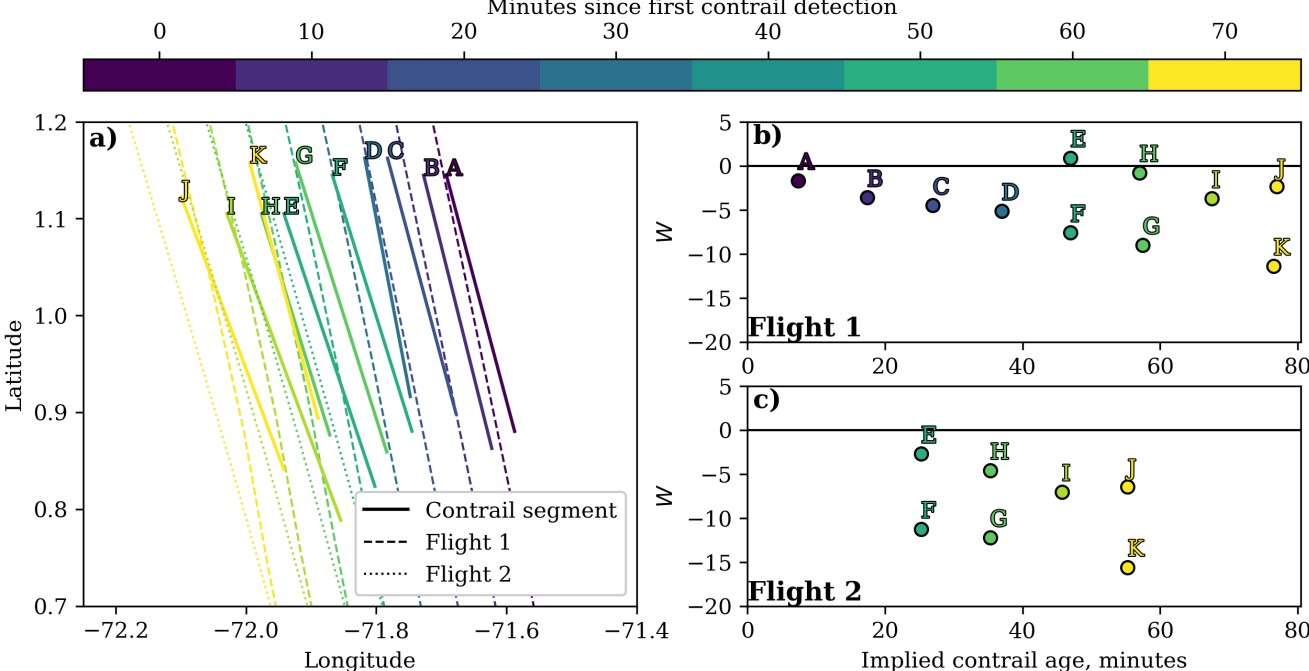

**Figure 3.** Visualization of a contrail-to-flight attribution problem involving two flights that both formed a contrail. **a)** Shows the detected linear contrails for a 70-minute period (covering 8 GOES-16 ABI images), accompanied by the flight tracks advected to the GOES-16 capture times. Each linear contrail and flight track is colored according to its corresponding image time. In **b)** and **c)** we show the value of the single-frame attribution parameter $W$, which approximately measures the advection error perpendicular to the contrail, for each possible flight and contrail detection pairing, as a function of the time between the passage of the flight and the moment of detection (i.e. the implied contrail age).

### 2.2.3 CoAtSaC Attribution Algorithm

CoAtSaC improves upon the single-frame algorithm by considering the temporal evolution of the transformation parameters $V$, $W$ and $\theta$ from Equation 1, with a particular focus on $W$. The algorithm is composed of 3 stages. The first stage, called "Fitting," looks at all single-frame attributions to a single flight and leverages the expected temporal evolution of $W$ in order to

group together detections of the same contrail in different frames. The second stage, called "Rejecting," combines the evidence from the first stage across multiple candidate flights for each contrail detection and uses that to determine a subset of the single-frame attributions which can be confidently rejected. The third stage reruns "Fitting," but without the potential confounders that were eliminated in the second stage.

To provide intuition for the "Fitting" stage, we consider the situation depicted in Figure 3(a), which shows two contrails

formed by two different flights over a period of 70 minutes. In Figure 3(a), "Flight 1" passes through the domain before "Flight 2" and forms a contrail that is detected in 7 consecutive GOES-16 ABI images (line segments A, B, C, D, F, G and K). The





contrail formed by "Flight 2" is first detected 40 minutes later (line segments E, H, I and J). The flight tracks, advected to the time corresponding to each relevant GOES-16 ABI image, are also shown in Figure 3(a) by use of dotted and dashed lines. Figure 3(b) and Figure 3(c) show the values of the transformation parameter $W$ for Flights 1 and 2 respectively. For the single-

frame attribution algorithm, an ambiguous situation occurs 40 minutes after the first contrail detection, when line segment E (which is the first detection of the contrail formed by Flight 2) is close to the advected flight tracks of both flights. In fact, the single-frame attribution score $S_{attr}$ between Flight 1 and line segment E is smaller than that between Flight 2 and line segment E (which is the correct one). Thus, a single-frame attribution algorithm may erroneously match flight 1 to line segment E. If we however consider the temporal evolution of the value of $W$ for both flights as shown in Figure 3(b) and Figure 3(c), we see

that for both flights we can identify two "groups" of single-frame matches that can be connected by a line. For flight 1, we can imagine points A, B, C, D, F, G and K to form one such line, and E, H, I, J the other. To understand why this is the case, we note that for a constant error in the wind data used for advection, we would expect a displacement error between the advected flight track and detected contrail that linearly increases with time, which roughly corresponds to $W$ increasing linearly with time. Depending on the number of contrails present in one particular situation, many such "lines" may be present in a figure like

Figure 3(b). Importantly, these lines will differ in the location of their intercept with the $W$-axis: we expect the line connecting detections of a contrail formed by a flight to intersect the $W$-axis near zero, implying that if the satellite could have observed this contrail forming, it would be exactly at the location of the flight waypoints before any advection. This assumption, that the $W$-axis intercept value of a line connecting the contrail detections formed by a flight should be near zero, is the key component of the CoAtSaC attribution algorithm. Considering Figure 3, this would lead us to attribute A, B, C, D, F, G and K to flight 1,

and the remaining detections, E, H, I, and J, to flight 2.

The algorithm, based on this intuition, requires access to all single-frame attributions for each flight, and the ability to analyze the temporal evolution of the $W$ parameter (see Appendix A6 for a discussion of why we do not use $V$ and $\theta$ also). For the time dimension of this analysis, we acknowledge that we do not know the ages of the detected contrails, but we do know how long the flight waypoints that are considered in each attribution have advected. We can thus take the mean over the advection

times of the overlapping waypoints and call this "implied contrail age." This is consistent with the age that was used to set the coefficient values in Equation 2, and it is important to note that the value can vary dramatically for the same detected contrail when attributed to different flights. In order to gain access to $W$ values that have a meaningful temporal evolution, we require a slight modification to the single-frame algorithm described in Section 2.2.2: we make the regularization coefficients, $C_{fit}$, $C_{shift}$, and $C_{angle}$ consistent regardless of contrail age, specifically fixing them at the values they would take on for a flight that had advected for 30 minutes. With this in place, we can observe the temporal evolution of $W$ for the single-frame attributions,

and plot them separately for each flight. As we saw in Figure 3, there is a clear pattern where detections of the same contrail in nearby frames result in a $W$ value that varies linearly in time, even when measured against a flight that did not form the contrail. We show a number of additional examples in Figure 4, including some where identifying the linear structures is more challenging due to there being large numbers of nearby contrails.

These more challenging examples motivate the need for an algorithm that is both noise-tolerant and able to leverage additional evidence to reduce the number of confounders. The "Fitting" stage of CoAtSaC adapts the Sequential Random Sample

 

Consensus (RANSAC) algorithm (Torr, 1998) to the task of extracting the multiple linear structures representing individual physical contrails from collections of single-frame attributions for a single flight represented in $W$ by implied contrail age space. By combining the linear groupings that include the same contrail detection across different flights, the "Rejecting" phase applies a series of heuristics aimed at eliminating the incorrect attributions. With those eliminated, "Fitting" can rerun, but with fewer confounders, and thereby produce more confident determinations of contrail attribution. We now present each stage in greater detail.

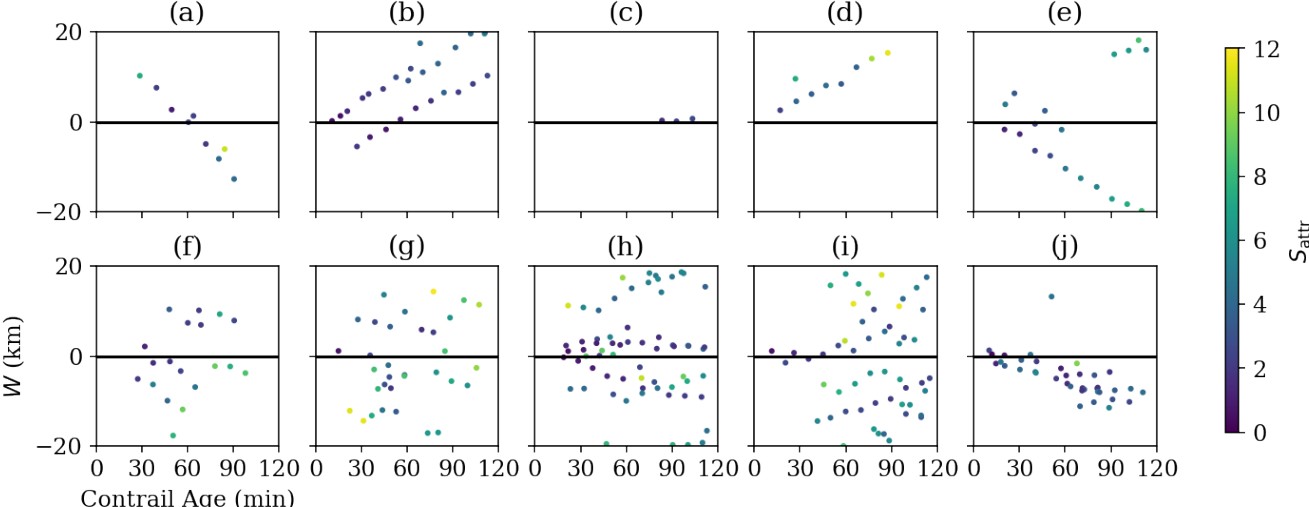

**Figure 4.** Examples of single-frame attributions for individual flights plotted on implied contrail age by $W$ axes. Unlike in Figure 3, color here indicates the single-frame score $S_{attr}$ for each attribution, rather than identifying the satellite frame. **(a)** 1 (possibly 2) contrails where the detections around 60 minutes have a small $W$ value and low $S_{attr}$. The single-frame algorithm would incorrectly attribute these detections to this flight, whereas, because of the large W-intercept, we can be confident that they were formed by a different flight. **(b)** 3 contrails, only one of which was likely actually caused by this flight. **(c)** A contrail with a shallow slope and near-0 W-intercept, but that is first detected long after this flight passed through. This might have been due to occlusions from other clouds, or due to a later flight forming a contrail near the advection path of this flight. **(d)** A case where the $S_{attr}$ values move out of the match range as the contrail ages, leaving them available to incorrectly match to other flights. **(e)** One long-lived contrail that is likely caused by this flight, with a few other nearby contrails that might make it tricky to fit lines correctly. **(f)** A few short-lived contrails nearby cause a danger of fitting spurious vertical lines across contrails, unless there is a prior to prefer shallow slopes. **(g)-(j)** Examples of higher contrail density that result in different degrees of difficulty in identifying the linear structures that track individual contrails.

**Fitting:** The intra-flight "Fitting" stage is visualized in Figure 5. We start by gathering all single-frame contrail attributions for one flight and filtering them to those that have an $S_{attr}$ score below a threshold, $T_S$, as is shown in Figure 5(a). Each single-frame attribution has an associated range of waypoints that ostensibly formed each contrail, so we can divide up the attributions for each flight such that each group is responsible for a range of waypoints that does not overlap any other group, as is shown in





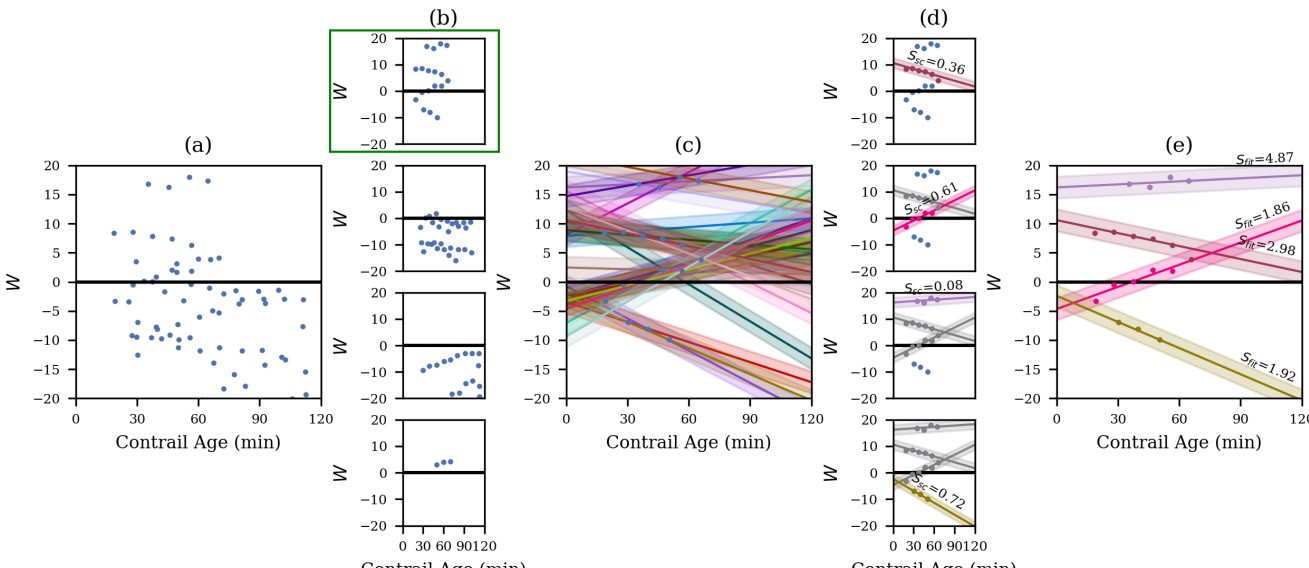

**Figure 5.** A visual depiction of the intra-flight "fitting" stage of CoAtSaC for a single flight. In **(a)** we enumerate all of the single-frame attributions to this flight with low enough $S_{attr}$ values and plot them in implied age by $W$ space. It is hard to extract much structure from this view. In **(b)** we divide the attributions into groups based on being attributed to overlapping ranges of flight waypoints, and plot each group separately. Here we can see the linear structures more clearly. In **(c)-(e)** we work just with the green-highlighted group plotted on top, but the same process would be independently applied to each of them. In **(c)** we enumerate all pairs of attributions, subject to the age gap and slope validity checks. In this example there are 18 single-frame attributions, resulting in 153 possible pairs, 29 of which satisfy the validity criteria. Each of the 29 pairs defines a line, which is plotted in an opaque distinct color, and a surrounding semi-transparent region where other attributions, not in the original pair, would be considered inliers to this line fit. In **(d)** we show the process of selecting from the fit lines, from top to bottom. In the top panel, we have all single-frame attributions available, so we pick the line that has the most inliers, 6 in this case, and break ties by the lower value of $S_{sc}$. The best line and its inliers are shown in maroon below its $S_{sc}$ value. In the second from the top, we show the first selected fit and its inliers in gray, depicting that we have removed the inliers. We then repeat the process of selecting the next best line from the remaining single-frame attributions, shown in pink. One of the single-frame attributions would have been an inlier to this fit, but it was already claimed by the previous fit, so it is excluded here. The process is repeated until no more candidate lines remain. In this example, 4 line fits are produced. In **(e)** we show the 4 lines and their inliers in distinct colors, along with their values for $S_{fit}$. Heading into the "Rejecting" stage, the pink and olive colored fits are below the score threshold of 3, and the purple fit is above it, which is what we expect given the $W$-intercepts. The maroon fit is very close to the threshold, and has a relatively high $W$-intercept, so ideally another flight will cause it to be rejected in the "Rejecting" stage. Note that the purple fit has the highest $S_{fit}$ score and the lowest $S_{sc}$ score, meaning that we are confident that it is the same physical contrail and also that it was not formed by this flight.





Figure 5(b). In most cases this step substantially simplifies the line fitting process. If any group contains just a single attribution we skip it, since we cannot fit a line to a single point. We then proceed with line fitting within each group independently.

Similar to RANSAC, we iteratively sample, without replacement, pairs of attributions from within a group from which we
can produce candidate lines. We can immediately filter out some of these pairs if they do not satisfy the criteria of being temporally within $T_t$ hours of each other, have an absolute slope $|\frac{dW}{dt}| < T_{\text{dW/dt}}$, and have overlapping attributed waypoints. The slope term, in particular, is important for avoiding fitting lines that span multiple linear structures in the data. If the allowed slopes were unbounded, an example like Figure 4(f) could end up with a near-vertical line that groups together what is likely 5 or 6 different contrails. This term, in effect, encodes an expected upper-bound on the rate of $W$ growth for a contrail. A pair
that passes all of these conditions defines a line, with slope $m$ and $W$-intercept $b$. The other attributions in the group are labeled as inliers or outliers to this line based on a residual threshold $T_{\text{res}}$. Specifically, an attribution with implied age $t_i$ and W value $W_i$ is an inlier if $(mt_i + b - W_i)^2 < T_{\text{res}}$. This threshold should generally be thought of as a function of measurement noise that is relatively independent across satellite frames, such as from contrail linearization and quantization of contrail location due to satellite image resolution. This process of computing fit lines and inliers is shown in Figure 5(c). We hereafter refer to the fit
line and its set of inliers as a "fit."

It is important to recognize that a goal at this stage of the algorithm is to group attributions to the same physical contrail as detected in different satellite frames into fits, regardless of whether they are correctly attributed to this flight. Attributions that correspond to the same contrail, but are formed by a different flight will still form lines, but they will generally have non-zero intercepts, $b$. The goal is then to identify these sets of attributions so they cannot be spuriously grouped with a different contrail
and made to look like they were created by this flight. To accomplish this we compute a score that attempts to measure how likely it is that all of the inliers of the current candidate fit are detections of the same contrail:

$$S_{\text{sc}} = C_{\text{slope}} m. \qquad (3)$$

$C_{\text{slope}}$ is a tunable hyperparameter that multiplies the slope of the fit line. We experimented with including terms based on the $R^2$ of the inliers with respect to the fit line and the number of unique satellite frames represented in the inliers, but the tuning
described in Section 2.4.3 found these not to be valuable.

Finally, we call the candidate fit the current best if it either has more inliers than the previous best, or it has the same number of inliers and a lower $S_{\text{sc}}$ value.

Since the data we are working with has relatively few points per line and potentially many lines per scene, we do not rely on the relatively small number of sample pairs used for typical RANSAC, as described in Fischler and Bolles (1981). We allow
for up to 5000 sampled pairs, without replacement, per group, which in the vast majority of cases results in an exhaustive sampling.

Once we have completed the sampling, if a valid best fit was found, we remove all of the fit's inliers from the set of candidate attributions in the group and run the fitting process again with the remainder, repeating until we either have fewer than 2 attributions remaining or we are unable to find a valid fit. This is shown in Figure 5(d).





At this point we have some number of fits for each flight. We can now make an initial determination for each fit whether it is likely to represent a contrail created by this flight or a different flight by computing a fit score:

$$S_{\text{fit}} = S_{\text{sc}} + C_{\text{int}}|b| + C_{\text{sing}}\min_{\text{inliers}}(S_{\text{attr}}),  \tag{4}$$

where $|b|$ is the absolute value of the $W$-intercept of the fit line and $C_{\text{int}}$ and $C_{\text{sing}}$ are tunable coefficients. This encodes the assumption that a small $W$-intercept , combined with a low smallest $S_{\text{attr}}$ (which mostly helps avoid substantial rotation error) are indicators that the contrail tracked by this fit was formed by this flight. The "Fitting" stage does not itself act on the $S_{\text{fit}}$ score. The first time "Fitting" is run, the subsequent "Rejecting" stage will consume these scores, and the final time it is run, these scores will determine the final attribution decisions.

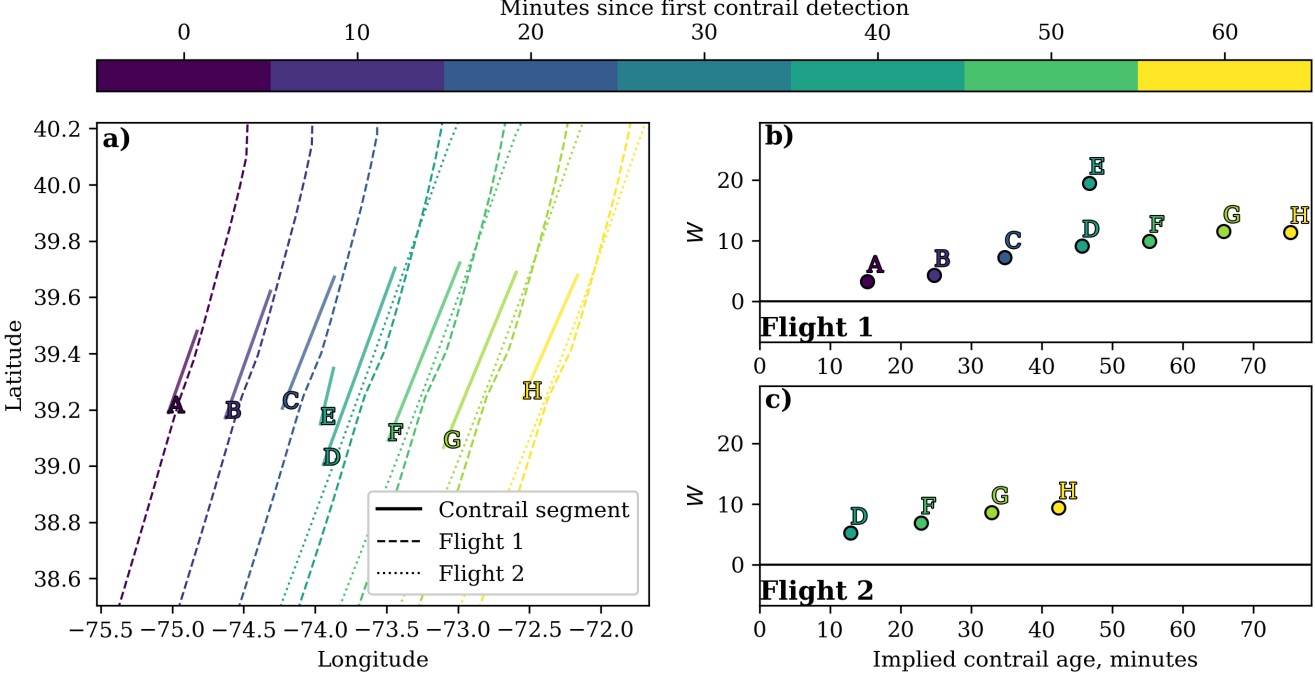

**Figure 6.** An example of the inter-flight "Rejecting" stage of CoAtSaC, presented in the same format as Figure 3. Flight 1 and Flight 2 both have linear fits that have shallow slopes and near-0 $W$-intercepts, which gives them low $S_{\text{fit}}$ values. They also both include contrail detections D, F, G, and H. Flight 1 additionally includes contrail detections A, B, and C in its fit. Because the relevant waypoints of Flight 2's flight path did not yet exist when contrails A, B, and C were detected, we can confidently reject these attributions for Flight 2, since it most likely just flew close to a pre-existing contrail.

**Rejecting**: Whereas the "Fitting" stage's goal was to group together attributions of the same physical contrail and produce a score indicating the likelihood of it being formed by the flight in question, using no additional evidence, the inter-flight "Rejecting" stage combines this evidence across flights to eliminate as many incorrect single-frame attributions as possible.





Without this stage, there is a strong possibility that the "Fitting" stage would produce fits for multiple flights containing the same contrail detections, all with $S_\mathrm{fit}$ scores below the target threshold. This is not inherently problematic, since there can be errors in the contrail detection process that result in merging together distinct contrails. Even when that is not the case, we could express some of the uncertainty in the algorithm by dividing the attribution between multiple candidate flights with different

confidences. However, there are cases where looking across the different flights that have fits containing the same contrail can be used to refine our results. The "Rejecting" stage accounts for two such cases. Both are identified by first gathering all fits that include the same contrail detection across all flights. The first case of interest is if any pair of fits share at least 2 contrail detections, and one of them also includes contrail detections that predate the other flight segment. In this case we can assume that the later flight just flew very close to the existing contrail, and we reject the single-frame attributions between the common

contrails and the later flight. An example of this is given in Figure 6, where Flight 1 and Flight 2 both look like good fits in isolation, but when we notice that Flight 1's fit includes all of the contrail observations from Flight 2's fit, as well as three more that were observed before Flight 2 even passed through, we can confidently say that Flight 2 did not form this contrail. The second case relies on the quality of the fits produced in the "Fitting" stage. As we saw in Figure 5, some fits it produces have $W$-intercepts far from 0, implying a low likelihood that the constituent single-frame attributions are correct. This, and other

measures of fit quality, factor into the $S_\mathrm{fit}$ score. We therefore compare these values for each of the fits and if any is more than a threshold $T_b$ higher than the lowest value, we reject all of its single-frame attributions as well. In Figure 5(e), the purple and maroon fits, and their constituent single-frame attributions, which have $W$-intercepts far from 0, should be eliminated by this process, assuming that the algorithm has access to the flights that did form those contrails.

Once the "Fitting" stage has produced an initial set of fits and the "Rejecting" stage had combined their evidence to reject

many confounding single-frame attributions, we rerun the "Fitting" stage with just the non-rejected single-frame attributions. In principle we could run more iterations of "Rejecting" and "Fitting," but in practice there are very few remaining contrails attributed to multiple flights after just doing "Fitting-Rejecting-Fitting," using our parameter values. The resulting fits define the final attribution decision, which is determined by $S_\mathrm{fit} < 3$, with the value 3 being chosen for consistency with Geraedts et al. (2024).

A critical benefit to this algorithm is that it is highly scalable. The "Fitting" stage can be parallelized over flights, and the "Rejecting" stage can be parallelized over contrail detections. This lends itself well to being implemented in the Dataflow Model (Akidau et al., 2015) using a framework like Apache Beam (Apache Software Foundation, 2024). In principle this enables the algorithm to scale to all flights and all contrail detections globally, where the speed of the algorithm is proportional to the number of compute nodes provided to it. This is in contrast to approaches that require optimizing over a full graph of

flights and contrail detections, which require holding the complete graph in memory of a single computer.

## 2.3 Synthetic Contrail Benchmark Dataset

We now turn to the questions of how to tune the 8 hyperparameters of CoAtSaC and how to evaluate its performance. In an ideal world we would use a dataset of ground-truth contrail attributions in geostationary imagery. Currently, no such dataset exists, since it is an extremely challenging task for even a skilled human to perform without additional evidence. In the absence of





such a dataset, the answer we propose is a synthetic contrail dataset. Specifically, we aim to provide a set of synthetic contrail detections that can be directly input to the attribution algorithm. The synthetic contrail detections should be as statistically similar as possible to real detections, while specifying which flight created each contrail. While not a strict requirement, we choose to produce the dataset corresponding to the capture times and pixel grid of real GOES-16 ABI scans, since that allows for both quantitatively and qualitatively comparing with the real contrail detections from the corresponding scan.

Importantly, these synthetic contrail detections are simulating a particular detection algorithm run over imagery from a particular geostationary satellite, including the flaws of both. They are not attempting to model reality or what an expert human labeler might produce for a given satellite image. It is not a goal of this dataset to have exactly the same flights that formed detectable contrails in reality also form contrails in this dataset, nor do the synthetic contrails need to end up being in exactly the same locations as the contrails the detection algorithm finds in the same scene. Ultimately the critical element is that the

dataset has similar statistics in terms of contrail density, dynamics, detectable lifetime, and advection error characteristics relative to the ERA5 weather data, so that we can measure the attribution algorithm's performance across all scenarios that it is likely to encounter on real contrail detections. An added benefit is for the dataset to provide physical properties of the synthetic contrails that allow for studying the attribution algorithm's performance as a function of these properties.

The dataset described here, which we name "SynthOpenContrails," is tuned towards the performance of the contrail detection

algorithm introduced by Ng et al. (2023) on GOES-16 ABI Full Disk imagery. The approach should be adaptable to other detection algorithms and other satellites, but some details and parameter values may need to change. We also expect that algorithms built around other detection methods should still be able to use SynthOpenContrails as-is, and we demonstrate this in Section 3 by evaluating the Chevallier et al. (2023) algorithm, with only minor modifications.

### 2.3.1   Data

The data used to produce the synthetic contrails consists of flight paths and historical weather data. We generate the dataset for the same spatial region and the same train, validation and test time spans as we run the attribution algorithm over, ensuring that it provides contrail detections also for the surrounding satellite frames that the attribution algorithm needs.

In order to avoid biasing towards our assumption that advecting flights for 2 hours is sufficient, we consider all flight waypoints that were flown at any point between 6 hours before the start of each time span and 3 hours after it ends. For flight

loading purposes we also dilate the spatial region by 720 km in each direction, to allow contrails formed by flights outside the region to advect in from all directions without presuming anything about the wind direction. We also do not perform any of the filtering on flight paths that was done for the inputs to the attribution algorithm in Section 2.2.1. We resample each flight to $C_{\text{Tflight}}$ seconds in between waypoints, such that there will end up being roughly 2 waypoints per GOES-16 ABI pixel at typical aircraft speeds.

In selecting weather data to use, it is important that we do not use the same weather data as is used for flight advection in the attribution algorithm itself, since that would result in having unrealistically low advection error. To that end, we use the control run of the ERA5 Ensemble of Data Assimilations (EDA), which has a coarser resolution than the nominal ERA5 reanalysis product. The ensemble data is at 3-hour intervals, 0.5625 degrees spatial resolution, and vertically discretized to 37





pressure levels separated by roughly 25-50 hPa. We unintentionally excluded the levels between 450 hPa and 975 hPa, which

led to some minor weather interpolation artifacts at the low end of the contrail formation altitudes (see Section 3.1.1). See

Appendix A7 for a discussion of the appropriateness of selecting this source of weather data.

### 2.3.2 Dataset Generation

---

**Algorithm 1** Pseudocode for generating synthetic contrails

---

**Input:** $AllFlights, AllFrames$

**Output:** $Rasters, LinearContrails$

  $FlightRastersPerFrame[AllFrames] \leftarrow []$

  **for** $Flight$ in $AllFlights$ **do**

    $CocipResult \leftarrow$ RunCoCiP($Flight$)

    **for** $Frame$ in $AllFrames$ **do**

      **if** $Frame$ in $CocipResult$ **then**

        $FrameResult \leftarrow CocipResult[Frame]$

        $Reprojected \leftarrow$ ReprojectGeostationary($FrameResult$)

        $Filtered \leftarrow$ FilterUndetectable($Reprojected$)

        $Adapted \leftarrow$ AdaptToDetector($Filtered$)

        $FlightRastersPerFrame[Frame]$.append(Rasterize($Adapted$))

      **end if**

    **end for**

  **end for**

  $Rasters[AllFrames] \leftarrow None$

  $LinearContrails[AllFrames] \leftarrow []$

  **for** $Frame$ in $AllFrames$ **do**

    $CombinedRaster \leftarrow$ CombineRasters($FlightRastersPerFrame[Frame]$)

    $Rasters[Frame] \leftarrow$ HandleOutbreaks($CombinedRaster$)

    $LinearContrails[Frame] \leftarrow$ Linearize($Rasters[Frame]$)

  **end for**

---

The process for generating the synthetic contrail detections is outlined in Algorithm 1. We summarize each subroutine below, with further details found in the appendices.

**RunCoCiP:** We simulate contrail formation and evolution using CoCiP (Schumann, 2012), which is a Lagrangian model simulating contrail formation and evolution, as implemented in the PyContrails library (Shapiro et al., 2024). We configure PyContrails as specified in Appendix A8. PyContrails can produce outputs only at fixed time intervals, so in order to capture the outputs we need at the times corresponding to GOES-16 scans with the correct scantime-offsets, we configure it to produce outputs at 30 second intervals for the duration of the longest-lived contrail formed by the provided flight. If a flight does not





form a contrail according to CoCiP, then PyContrails will have no outputs, so we do not consider this flight any further. For
flights that do form contrails, PyContrails outputs contrail properties for each contrail-forming input flight waypoint at each
timestep, but we are only interested in the properties that would manifest at the times that the GOES-16 ABI would capture it.
We therefore compute the scantime offset corresponding to the location of each output, and then select just the timestep that is
closest to each satellite scan plus scantime offset for each waypoint. This results in a maximum of 15 seconds of advection error,
which is negligible for our purposes (see Appendix A9). At this stage we divide each flight's outputs up by the corresponding
satellite scan and operate on them independently.

**ReprojectGeostationary:** The goal of this subroutine is to reproject CoCiP's outputs from its native frame of reference to
the perspective of the geostationary imager. CoCiP computes the parameters of the contrail plume cross-section at each flight
waypoint such that attributes like width and optical thickness are measured along a viewing ray that passes directly through
the center of the contrail to the center of the earth. In order to render off-nadir contrails in the perspective of a geostationary
satellite, we need to recompute these values using the viewing ray of the instrument. The details of how this is accomplished
are in Appendix A10.

**FilterUndetectable:** This subroutine's purpose is filtering CoCiP's outputs to just those that the Ng et al. (2023) detector
would be likely to find if they existed in reality. This amounts to codifying whether the training data for the detector would have
included a label for this contrail. It computes a per-waypoint detectability mask, taking into account a few criteria, detailed in
Appendix A11.

**AdaptToDetector:** Before actually rasterizing the CoCiP data, we apply some adaptations directly to CoCiP's outputs, in
order to better reflect the behavior of the detector being emulated. These are specified in Appendix A12.

**Rasterize:** In this subroutine, we map the filtered and adapted CoCiP outputs to pixel values in the geostationary imager's
native projection and resolution. The most important component is determining what quantity should be rasterized in order to
best imitate the detector. Since the Ng et al. (2023) detector exclusively consumes longwave infrared bands, what it is actually
detecting is changes in outgoing longwave radiation. To this end, the quantity that we can extract from CoCiP that will best
reflect detectability is opacity, which, by the Beer-Lambert law (Beer, 1852), can be expressed as $\kappa = 1 - e^{-\tau}$, where $\tau$ is
the contrail optical depth produced by CoCiP. Appendix A14 discusses the appropriateness of applying the Beer-Lambert law
here. The actual rasterization process adapts the process described in Appendix A12 of Schumann (2012) to geostationary
satellite imagery. This is detailed in Appendix A15. The output of this subroutine is an opacity value $\kappa_{\mathrm{ras}}$ for each pixel in the
geostationary image that a flight contributed to in a single frame, along with the relevant CoCiP metadata for each waypoint
that contributed to the pixel.

**CombineRasters:** We can then combine the rasters for all flights at the same timestep, keeping track of the per-flight contrail
parameters contributing to each pixel for later analysis. For simplicity, we resolve different flights contributing to the same pixel
in the final raster by taking the maximum. The more correct approach would be to sum the optical thicknesses before converting
to opacity, but CoCiP does not model these inter-flight effects and in practice it does not matter much for our use-case. In order
to simulate some of the smoothing effect that the detector has over the relatively noisy satellite imagery, we apply a gaussian





blur, with standard deviation of 1, without allowing any zero-valued pixels to become nonzero. We produce a binary contrail
mask by thresholding the raster by $\kappa_{\mathrm{ras}} > 1 - e^{-T_\tau}$.

**HandleOutbreaks:** This subrouting addresses the mismatch between how CoCiP and the Ng et al. (2023) detector operate
in regions of very high contrail density, which we refer to as "contrail outbreaks." Generally CoCiP will cover the entire region
in contrails, to the point where individual contrails cannot be identified, while the detector will only identify the few most
optically thick contrails. Appendix A17 details how we adapt theses regions to behave more like the detector.

**Linearize:** In this subroutine we map the rasterized opacities, which include per-pixel attribution metadata, to linear contrail
segments that can be used in a contrail-flight attribution algorithm. This process is a close analog to the processing Ng et al.
(2023) applies to real satellite imagery and the resulting detector outputs, but with some additional bookkeeping. The full
process is described in Appendix A18.

## 2.4 Tuning and Benchmarking

This section describes how we tune the parameters of the synthetic dataset generation pipeline on observations and use the
resulting dataset, called SynthOpenContrails, to tune the parameters of the CoAtSaC attribution algorithm, after which we
compute and compare benchmark metrics for multiple attribution algorithms.

### 2.4.1 Tuning the Synthetic Dataset Parameters

The pipeline we have described for generating synthetic contrails includes a number of parameters whose values need to be
determined. The intention here is to allow the same fundamental approach to be used to produce synthetic contrails that emulate
different detection algorithms or different satellite imagers, just by setting different values for the parameters. As mentioned
previously, for SynthOpenContrails we produce synthetic contrails using real flights and real weather at the capture times of
real GOES-16 ABI images. This allows us to tune towards matching the behavior of the Ng et al. (2023) detector on the real
data. We manually tune to quantitatively match the statistics for number of contrail pixels and number of linear contrail per
frame. We can further qualitatively compare by overlaying the real and synthetic contrail masks on sequences of GOES-16
ABI imagery. We use the Ash color scheme as used previously in Kulik (2019); Meijer et al. (2022); Ng et al. (2023) to map
infrared radiances to RGB imagery that makes optically thin ice clouds, like contrails, appear in dark blue. An example frame
of this imagery with both real and synthetic detections overlaid is shown in Figure 7. For tuning purposes, we compute the real
and synthetic contrail detections for the full validation set of time spans (details in Appendix A28) and apply our comparisons
over those. We note that there are likely multiple sets of parameters that match our real data equally well, and the parameters
used for SynthOpenContrails are just a single instantiation of this. We therefore caution against attempting to extract physical
insights from SynthOpenContrails, as it has been designed only for evaluating contrail-to-flight attribution and is in essence
a filtering of CoCiP simulations. The tuned parameter values we use for generating SynthOpenContrails are in Table A2. We
present some quantitative statistics and qualtitative observations of SynthOpenContrails in Appendix A21.



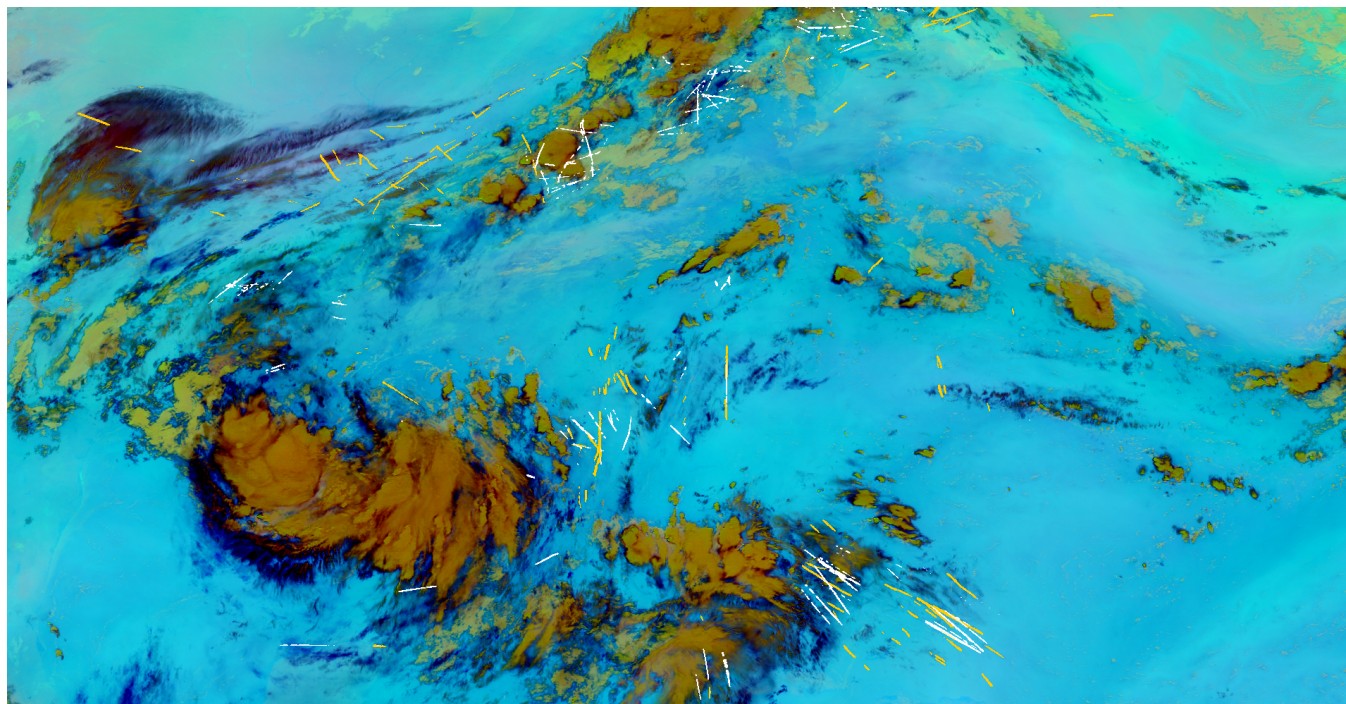

**Figure 7.** An Ash color scheme false-color GOES-16 ABI image from 11 July, 2019 at 12:40 UTC over the south-eastern United States, with the contrail mask produced by the Ng et al. (2023) detector in yellow and the SynthOpenContrails mask in white. The SynthOpenContrails contrails generally appear in the same regions as the detected contrails, but there is far from perfect alignment, which is unnecessary for the purposes of this dataset.

### 2.4.2 Benchmark Metrics

Here we define a set of metrics used as the top-line results for the benchmark. The metrics are divided into per-contrail metrics and per-flight metrics. Generally the per-flight metrics will better assess the binary determination of whether a flight formed a contrail, while the per-contrail metrics will be better for accounting for the number of contrails formed and how long they persisted.

- **Per-contrail recall**: The percentage of linear contrails to which the algorithm has attributed the correct flight.

- **Per-contrail precision**: The percentage of the attribution algorithm's attributions to linear contrails that are correct (note that the algorithm can choose not to attribute any flight to a linear contrail).

- **Per-flight recall**: The percentage of flights that formed at least one linear contrail in SynthOpenContrails to which the attribution algorithm has attributed at least one linear contrail (regardless of whether that specific attribution is correct).



470       – **Per-flight precision**: The percentage of flights to which the attribution algorithm has attributed at least one linear contrail that also formed at least one linear contrail in SynthOpenContrails.

The precise method for computing each metric can be found in Appendix A19.

For the purposes of the benchmark, these metrics should be computed globally over the entire dataset. There are, however, other ways to compute them that may be more suitable for specific use-cases. As we will show below in Figure A6, there is a

substantial variance in the number of contrails, and the numbers of flights forming contrails, across satellite frames. A potential downside of the global approach, then, is that the frames with larger numbers of contrails will dominate the metrics, and mask the performance in scenes with fewer contrails. For some applications it therefore might be preferable to compute metrics independently per-satellite-frame and subsequently aggregate the metric values across frames in order to give equal weight to different conditions. Another reasonable modification when computing the per-contrail metrics could be to weight each linear

contrail by its length or area, in order to better reflect the fraction of contrail coverage that is being correctly attributed.

As the goal is to assess the performance of the attribution algorithms in isolation, these metrics are all computed relative to the filtered and adapted view of CoCiP provided by SynthOpenContrails and do not attempt to account for performance relative to the raw CoCiP outputs.

### 2.4.3   Tuning and Benchmarking Attribution Algorithms

Given a dataset of synthetic linear contrails labeled with the flight that formed them, divided by time span into train, validation, and test splits, we can then apply it to both tuning and benchmarking an attribution algorithm. Specifically, we simply run the attribution algorithm using SynthOpenContrails's linear contrails instead of detector-produced contrails, and compare the resulting attributions to the ground-truth labels we have for each synthetic linear contrail. From that we can compute the metrics of interest, as defined in Section 2.4.2.

Critically, the flights used to generate SynthOpenContrails are from the same database as those we use for attributions, but that database is known to be incomplete: at a minimum military aircraft are unlikely to be fully present, which (Lee et al., 2021) estimates to be 5% of air traffic globally (and may be higher over our region of study). In order to ensure that the attribution algorithms can handle contrails formed by flights that are missing from the database, when tuning and benchmarking we conservatively exclude a fixed random sample of 20% of flights. The selection of this value imposes an upper bound on the

metrics, which may not be realistic for an MRV system which is run by a government with access to its own military aircraft locations. Because of this, the metrics should not be interpreted directly as the performance of an attribution algorithm in the real world in an absolute sense. They should, however, provide a relative measure of performance between different attribution algorithms. We provide a sensitivity analysis on the choice of 20% in Appendix A23.

Using this setup, we apply Google Vizier (Golovin et al., 2017) as a blackbox optimization service to search through the

space of parameters of CoAtSaC, aiming to find the optimal set producing the highest values for the 4 metrics of interest using the train-split of SynthOpenContrails. How one chooses to prioritize each of the metrics relative to each other — an increase in one often leads to a decrease in another — depends largely on the intended use-case for the attributions. If the goal is an



MRV system that aims to capture the largest possible fraction of contrail warming, while tolerating some inaccuracies in the specifics, contrail recall might be the most important metric. If instead one aims to generate training data for a contrail forecast

model, where noise in the labels could impair the model, flight precision might be the better metric. Using the attributions to evaluate a contrail avoidance trial might require more of a balance between the metrics, depending on the size of the trial. For the purposes of this study we slightly prioritized flight precision, while keeping the other metrics above reasonable performance thresholds. The parameters chosen by this tuning are in Table A4.

## 3 Results

### 510 3.1 Evaluating Attribution Algorithms on SynthOpenContrails

We compare the performance of CoAtSaC with the single-frame algorithm of Geraedts et al. (2024) and the tracking algorithm of Chevallier et al. (2023) on the metrics mentioned in Section 2.4.2 over the SynthOpenContrails test split. The tracking algorithm was slightly modified, and was only evaluated on part of the test split, as detailed in Appendix A25.

We compute each metric defined in Section 2.4.2 over the dataset in aggregate, and we also compute them independently

per satellite frame (providing a mean and standard deviation) since the full dataset metrics might be dominated by a few large contrail outbreaks. We reiterate the caution that these numbers should be interpreted as relative performance metrics amongst the different attribution algorithms: 20% of flights are artificially excluded in the evaluation so the upper bound on contrail recall is 80% (Appendix A23) and SynthOpenContrails design choices for outbreak handling (Appendix A17) and detectable contrail lifetime (Appendix A22) may influence the metrics.

The results are in Table 1. The high level takeaway is that CoAtSaC outperforms the tracking algorithm on every metric, and both outperform the single frame algorithm on all metrics other than global flight-level recall. Generally CoAtSaC's recall gains are fairly minor, while the precision gains are on the order of 30% better than the tracking algorithm and 50% better than the single-frame algorithm. The improvements being far higher in precision than recall is a consequence of the tuning strategy we used in Section 2.4.3, and we suspect that we could have tuned to higher recall at the expense of precision. The global flight

recall losses can be attributed to contrails that are only detectable in a single frame, which cannot be attributed correctly by CoAtSaC. More investigation into these losses can be found in Appendix A26.

### 3.1.1 Performance as a Function of Contrail Properties

Because the SynthOpenContrails contrails are rasterized directly from CoCiP's outputs, we can propagate the properties that CoCiP assigns to each contrail segment through to the final linear contrail instances and then analyze how attribution per-

formance varies with each property. For these analyses we only look at contrail-detection-level metrics, since many of the properties of interest can not be meaningfully aggregated to the flight level. Examples of these slicings are shown in Figure 8, where we see that while the relative performance of the algorithms remains relatively constant, across all of them performance





**Table 1.** Performance of attribution algorithms on SynthOpenContrails (test split) using the metrics defined in Section 2.4.2. Metrics are computed both over the full dataset (global) and separately per-frame (per-frame). The per-frame metrics are presented as "mean (std=standard deviation)" over all frames in the dataset. Refer to Section 2.4.3 for why these should be interpreted as relative performance metrics and do not reflect expected performance in the real world.

| Algorithm | Contrail Precision (global) | Contrail Recall (global) | Flight Precision (global) | Flight Recall (global) | Contrail Precision (per-frame) | Contrail Recall (per-frame) | Flight Precision (per-frame) | Flight Recall (per-frame) |
|---|---|---|---|---|---|---|---|---|
| **Single-Frame** (Geraedts et al., 2024) | 40.3% | 33.0% | 41.4% | **62.2%** | 46.4% (std=12.1) | 33.1% (std=5.4) | 50.1% (std=12.3) | 43.7% (std=6.0) |
| **Tracking** (Chevallier et al., 2023) | 50.4% | 28.6% | 50.3% | 46.7% | 51.4% (std=12.2) | 29.1% (std=7.6) | 55.6% (std=10.9) | 37.7% (std=8.4) |
| **CoAtSaC** (ours) | **66.9%** | **36.6%** | **68.4%** | 50.6% | **69.6% (std=8.7)** | **37.5% (std=5.6)** | **71.6% (std=8.5)** | **46.2% (std=5.8)** |

falls off with increasing contrail density and age, improves with length, and has more complex relationships with altitude, season, and time of day. Further analysis of these results and additional results slicings can be found in Appendix A27.

## 3.2 Training a Contrail Forecast

Sonabend et al. (Forthcoming) introduced a machine-learning approach to training a model that can forecast future regions of detectable contrail formation, using training data that is derived from the contrail-to-flight attributions produced by Geraedts et al. (2024). They used this model, together with CoCiP, to execute a successful contrail avoidance experiment. Here we retrained the same machine-learned model, replacing the attribution labels with CoAtSaC's run over the training time spans.
Following their method of evaluating forecasts on the attributions from the test-split of time spans, using CoAtSaC in place of the Geraedts et al. (2024) labels improves Sonabend et al. (Forthcoming)'s key evaluation metric of "Area Under the Receiver Operating Characteristic Curve" from 85.5% to 91.7%. On its own this does not directly confirm improved skill at forecasting contrail-likely regions because the evaluations are also using labels produced by the attribution algorithms. However, together with the substantial increase in attribution precision shown on the SynthOpenContrails dataset and the fact that decreased
label-noise is known to improve the performance of learned classification models (Frénay and Verleysen, 2013), improved skill at forecasting is a likely explanation.



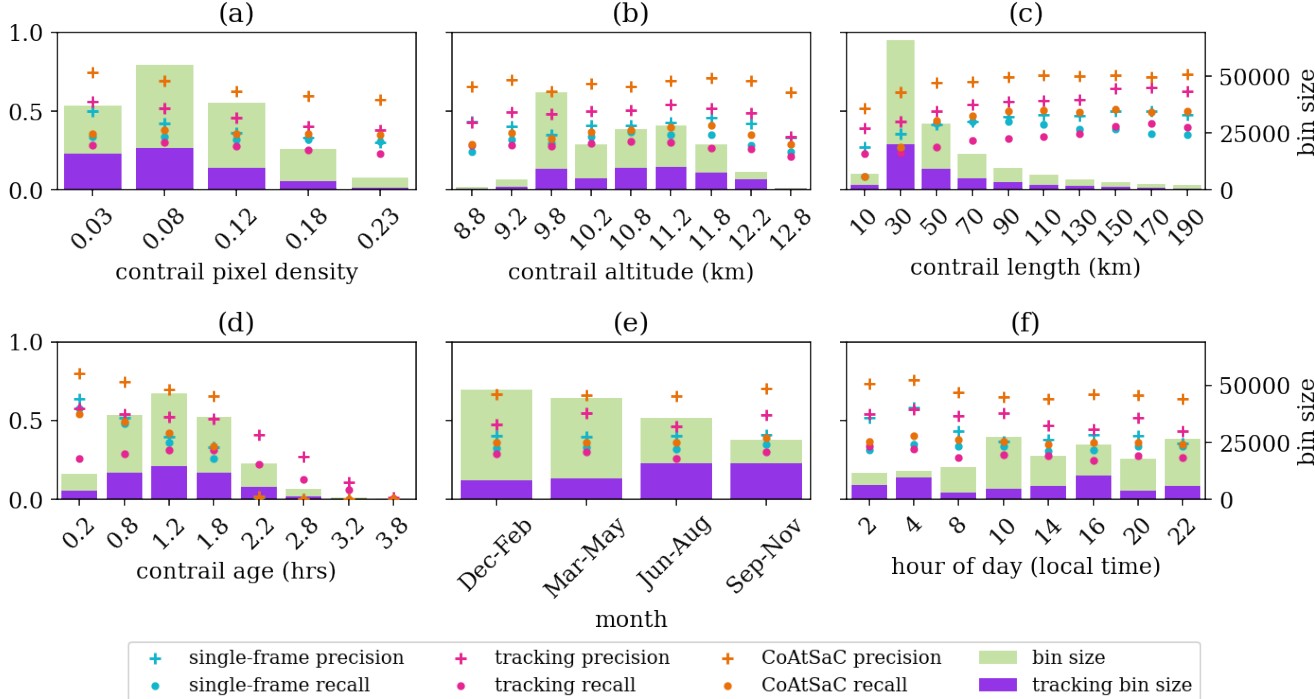

**Figure 8.** Performance metrics of each attribution algorithm shown as a function of various properties available to SynthOpenContrails. The recall and precision used here are contrail-detection level metrics. The green bars show the number of contrail detections in each bin for CoAtSaC and the single-frame algorithm, while the purple bars indicate the subset upon which the tracking dataset was evaluated. **(a)** shows performance binned by contrail pixel density (defined as fraction of contrail pixels in the 49x49 pixel window surrounding the center of the contrail). **(b)** shows performance binned by contrail pressure altitude. **(c)** shows performance binned by contrail length, as measured along the linearized contrail. **(d)** shows performance binned by contrail age. **(e)** shows performance binned by season. **(f)** shows performance binned by solar hour of the day at the contrail center.

## 4    Conclusions

We have presented a novel, highly scalable, contrail-to-flight attribution algorithm for geostationary satellite imagery (CoAt-SaC) and a large dataset of synthetic contrail detections (SynthOpenContrails). The SynthOpenContrails dataset allows us
to determine that the new algorithm substantially improves over the previous state of the art. It also allows us to study the performance of each algorithm as a function of contrail and scene properties.

The new attribution algorithm can potentially enable larger-scale live flight contrail avoidance trials, since the methods used to determine contrail formation in previous trials (Sausen et al., 2023; Sonabend et al., Forthcoming) would have difficulty scaling to larger number of flights. The resulting dataset of flights and contrails could also be used to evaluate contrail forecast
models, and to train machine learning contrail forecast models similar to Sonabend et al. (Forthcoming). It is also a necessary



step for observational approaches to become a main component of a contrails MRV system or a scope-3 emissions accounting system.

SynthOpenContrails should be helpful in continuing to improve the state of the art in contrail-to-flight attribution. In particular, it has made clear that there is a lot of room for improvement in areas of high contrail density, and that entirely different approaches to attribution might be necessary in those settings. It also seems clear that incorporating independent contrail altitude signals in the attribution algorithm has the potential for significant improvement, and future work will be needed to determine how to model those signals in a synthetic contrails context.

When generating synthetic data from CoCiP outputs, we found poor agreement between the CoCiP outputs and our detections. Differences on a per-contrail level are not surprising given uncertainties in weather data (Gierens et al., 2020; Agarwal et al., 2022), but we also found broader qualitative differences, in quantities such as overall contrail density. For the purposes of this study, distributional alignment between the statistics of the synthetic and real contrail detections was sufficient to evaluate a flight matching system, and we were able to achieve this by introducing variations in detectability as a function of contrail age and density. It would be valuable to disentangle which of these qualitative differences are accounting for errors in CoCiP's modeling, versus errors in the NWP modeling, versus classifying the subset of contrails that can theoretically be detected in geostationary imagery, versus the specific skill of a particular detection model. The answers to these questions could help improve all components of the system, from the detection models, to CoCiP and similar physics-based models of contrail formation and evolution, to the NWP models themselves. It can also inform which of these components can and should be used in either a predictive or retrospective context. One path towards disentangling these questions and validating some of the subjective decisions made in the synthetic dataset generation would be to build a high-fidelity large-scale dataset of real contrail detections with known flight attribution.

When evaluating an automated contrail monitoring system, one is concerned with the errors from both contrail attribution, which is the subject of this work, as well as the contrail detection, which is not. The methods in this work can only be used to compare different attribution algorithms which operate on the same contrail detections. A useful direction for future work would be a method of measuring the end-to-end performance of the overall detection and attribution system. Observation-based datasets that can track contrails from the moment of formation until they can be detected in a geostationary image, for example using ground cameras, could allow this. Since the ultimate goal is the reduction of contrail warming, the fraction of total contrail warming detected by a monitoring system could also be a useful metric. SynthOpenContrails could potentially provide a way to estimate this, since it does simulate the warming of each contrail, and whether that contrail is detectable or not. However the decisions around detectability in Sec. 2.3.2 were made with the goal of producing any dataset which qualitatively resembled available contrail detections. We have not established whether the decisions are a unique way of generating plausible detections or how the fraction of warming captured is sensitive to these decisions. We leave this for future work. Radiative transfer modeling such as in Driver et al. (2024) can also allow for the quantification of detectable warming. Observations of contrail warming on a per-contrail basis would also be very useful here.



*Data availability.* ERA5 data are available from the Copernicus Climate Change Service Climate Data Store (CDS): https://cds.climate.
copernicus.eu/cdsapp#!/dataset/reanalysis-era5-pressure-levels?tab=overview. Visualization of Contrail Detections on GOES-16 ABI data
can be found at https://contrails.webapps.google.com/. Raw GOES-16 data can be found at: https://console.cloud.google.com/marketplace/
product/noaa-public/goes.

The SynthOpenContrails dataset described in this paper is available upon request.

## Appendix A

### A1  Geographic region

The geographic region used both for the attribution algorithm and the synthetic dataset in this study is the same as in Geraedts
et al. (2024). It is the region bounded by great-circle arcs between (50.0783°N, 134.0295°W), (14.8865°N, 121.2314°W),
(10.4495°N, 63.1501°W), and (44.0734°N, 46.0663°W).

### A2  2 hour advection

The decision to advect flights for only two hours could limit the performance of the attribution algorithm. Many contrails do
persist and remain detectable in the GOES-16 ABI for longer than two hours (Vázquez-Navarro et al. (2015) showed this for
the Meteosat Second Generation satellite's SEVIRI instrument, which has lower spatial resolution than the GOES-16 ABI has),
and this decision makes it impossible to attribute these older observations properly, since the correct flight will not be available
to the attribution algorithm. Driver et al. (2024) finds that virtually all GOES-16 ABI detectable clear-sky contrails will become
so within the first 2 hours of their lifetime. This implies that if the goal of attribution is to whether a contrail forecast for a given
flight segment was correct, then 2 hour advection is usually sufficient. While CoAtSaC is benchmarked at two hour advection,
it is technically duration agnostic. Beyond two hours, however, we see a slight decrease in attribution performance, likely due
to increasing the number of candidate flights involved in the attributions decision for each observed contrail.

### A3  GOES-16 ABI scantime-offsets

Figure A1 shows the time interval between the nominal scan start-time for the GOES-16 ABI and when each pixel is actually
captured. The disc is divided into 22 west-to-east swaths, which are captured from north to south over the course of 10 minutes.
This needs to be taken into account when advecting flights for the purposes of contrail-to-flight attribution, since advecting to
the nominal scan-start time can introduce substantial error relative to when a detected contrail was actually captured.

### A4  Advection algorithm

We simulate the advection of flights in 3 dimensions using the third-order Runge-Kutta method (Bogacki and Shampine,
1989) with winds linearly interpolated from the weather data. Similar to Geraedts et al. (2024) we assume an initial wake
vortex downwash of 50 m and additional altitude loss due to sedimentation of the contrail's ice particles over time. In order to



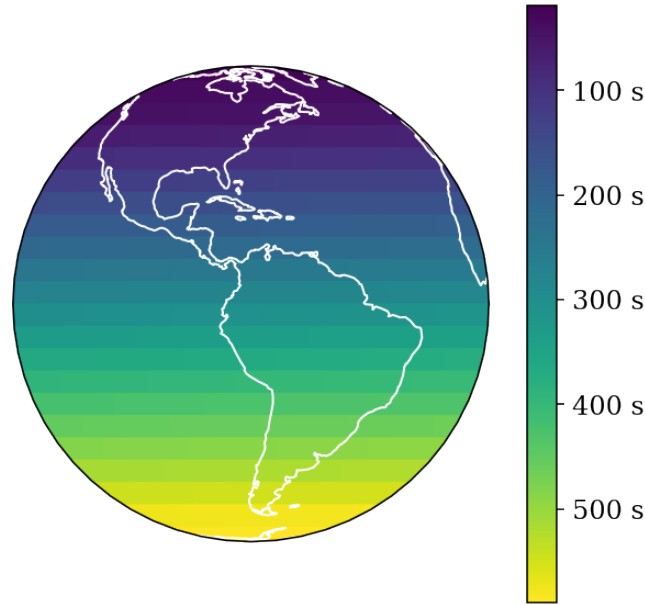

**Figure A1.** Scantime-offsets (the number of seconds after the nominal scan start time) for locations on the GOES-16 disk, when using the Scan Mode 6A (the current default scan pattern).

correctly compute sedimentation rates we would need to know the relative humidity along the advection path, but the ERA5 relative humidity values at flight cruising altitudes are known to be unreliable (Agarwal et al., 2022; Meijer, 2024). Since

one of our goals with contrail attribution is to evaluate contrail forecast models, most of which require relative humidity as an input, we want to avoid the attributions we produce having correlated errors with the forecasts, so we do not use relative humidity for computing sedimentation rates. Instead, we follow Geraedts et al. (2024) and sediment the contrail at a rate that is purely a function of contrail age based on a statistical fit to model data from Schumann (2012), which we would expect to be approximately correct on average, but not necessarily in the specifics.

**A5   Modifications to the Single-Frame Attribution Algorithm**

We make minor modifications to the Geraedts et al. (2024) single-frame algorithm.

Firstly, (Geraedts et al., 2024) divided flights up into 10-minute segments as part of preprocessing and attributed each segment independently. We skip this in order to avoid edge effects at segment boundaries.

Secondly, it is critical for CoAtSaC that the sign of $W$ does not change at random when looking at the same advected flight

and physical contrail in different satellite scans. For the single-frame algorithm, the sign is unimportant, as the values are always squared in Equation 2, so making it consistent has no negative effect on it. In order to impose consistency, we require that the advected flight be represented with $v$ values increasing with the timestamp of the original waypoint, and positive $w$ values being to the right with respect to the advected flight heading. Specifically, we perform the initial projection of flight





waypoints to the $v$-$w$ plane with coordinates $(w_i, v_i)$ without any such constraints, dropping any waypoints that are beyond

the span of the contrail. If the $v$ value for the earliest and latest waypoints in the overlapping set are $v_j$ and $v_k$, respectively, if $v_k < v_j$ then we multiply all of the $w_i$ and $v_i$ values by $-1$. For a an advected flight segment that is monotonic in $v$ as a function of time, this achieves the desired invariant. Occasionally there are advected flights that loop back on themselves, either due to unusual flight paths or unusual wind patterns, and these can result in having inconsistent signs for the $w$ values. We opt to tolerate failures in these cases, since contrails produced by these flight segments are highly unlikely to be attributed, or even

detected, successfully by an algorithm based on linearized detected contrails anyway.

## A6 Rationale for not Using Transformation Parameters $V$ and $\theta$

The CoAtSaC algorithm presented in subsubsection 2.2.3 focuses specifically on the $W$ parameter of Equations 1 and 2, but it only indirectly consumes the $V$ and $\theta$ values by way of thresholding the single frame $S_{\mathrm{attr}}$ values and incorporating $S_{\mathrm{attr}}$ into Equation 4. Here we discuss why the advection error implied by $V$ and $\theta$ carries less signal than that of $W$ for the purposes of

providing a signature useful for contrail-to-flight attribution.

The problem with $V$ is that if there is substantial error in the $v$ direction (parallel to the contrail) it manifests as changing the set of advected flight waypoints that are determined to be overlapping the contrail and are then input to Equation 2. This is tricky to resolve, since the contrail detections available at this stage are linear by construction, and most advected flight paths are also quite linear, so there are very few features to assist with proper alignment. A tracking-based approach, similar to

Chevallier et al. (2023), that directly consumes a contrail pixel mask or even raw radiances, could potentially align features of the detected contrails across frames, potentially also better aligning with any non-linearities in the advected flight path, to help minimize this drift in waypoint overlap.

The parameter $\theta$ also appears not to have much signal. We speculate that this is due to $\theta$ being a second-order effect, since it measures the change in advection error in the $w$ dimension over the length of the contrail. This measurement is made noisy by

the varying lengths of contrails and that they are often short relative to the spatial resolution of the weather data. Specifically, as can be seen in Figure A5, 21% of detected contrails have lengths shorter than the 31 km average grid size of the ERA5 weather data, and 59% are shorter than 62 km. This implies that variation in advection errors across a flight segment matching to shorter contrails will be dominated by the effects of the interpolation scheme in the weather data, while for longer contrails their will be more variance due to errors in the weather data itself.

## 660 A7 Use of ERA5 EDA Control Run for Synthetic Dataset Generation

In Section 2.3.1 we select the ERA5 EDA control run as the weather data to use for generating SynthOpenContrails. We note that the control run is not simply a lower resolution of the nominal, as the full EDA spread is used to set bias terms of the data assimilation process in computing the ERA5 nominal data (Hersbach et al., 2020). The important characteristic of the weather used for the dataset is that the differences, or error, between it and the weather used for advecting flights for the

attribution algorithm (ERA5 nominal is used for all algorithms evaluated in this study) be comparable to the error between the weather used for attribution and reality. Of course, not all aspects of the weather error actually matter for our use-case; we are



primarily concerned about the contributors to advection error. One way to measure this error is to look at the distribution of $W$ values from the single-frame attribution optimization outputs (regardless of final attribution determination) between flights advected with the ERA5 nominal product and real detected contrails, and compare it to the distribution of $W$ values for the

same advected flights computed against synthetic contrails generated (as described in Section 2.3) with particular weather data. If the distributions match, then the error characteristics are likely close enough for our purposes. We applied this test, using ERA5 nominal to advect flights and the first ERA5 ensemble member for generating synthetic contrails. The distribution of $W$ values for the real contrails have a standard deviation of 15.0 km and for the synthetic contrails it has a standard deviation of 15.2 km. The distributions are plotted in Figure A2. We acknowledge that matching the $W$ distribution does not capture all

components of advection error. Further research is required to determine, and maybe generate, a source of weather data that exactly matches every relevant characteristic of this error. We expect this will become more necessary as attribution algorithms start to approach perfect accuracy.

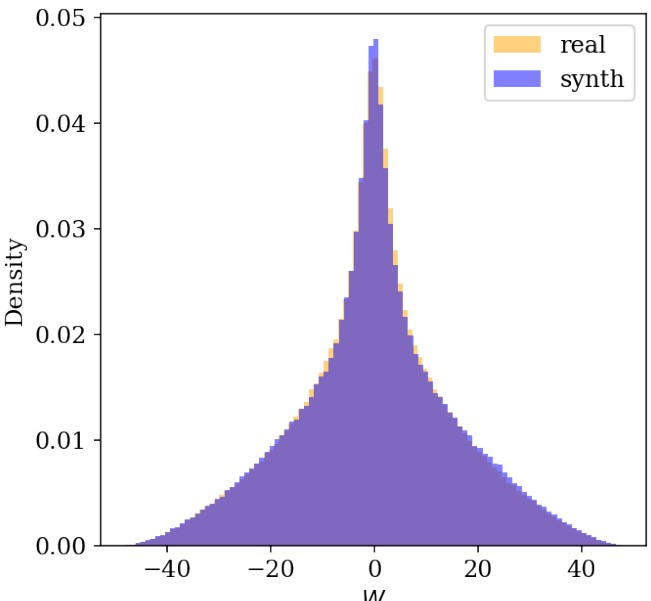

**Figure A2.** The distribution of $W$ values for all flight and contrail pairs in the validation time spans that produce an $S_{\text{attr}}$ value less than 12. The real contrail detection distribution is shown in orange and the SynthOpenContrails distribution is shown in blue. The distributions are nearly identical.

## A8  PyContrails Settings

Here we specify the settings we use for the PyContrails library's (Shapiro et al., 2024) implementation of CoCiP (Schumann,

680    2012).



In addition to flight track information and weather data, CoCiP requires aircraft performance data, specifically the aircraft wing span, aircraft mass, true air speed, fuel consumption per flight distance, soot number emission index, and the overall propulsion efficiency, which we estimate using the Poll-Schumann model (Poll and Schumann, 2021). The Poll-Schumann model is an open-source point-mass aircraft performance model that estimates fuel flow and other performance characteristics

for turbofan-powered aircraft across various flight regimes. It calculates flight performance based on inputs such as Mach number, aircraft mass, ambient temperature, and aircraft-specific characteristics. To generate the required emissions data for the CoCiP model, it incorporates the Fuel Flow Method 2 (DuBois and Paynter, 2006), and the Improved FOX (ImFOX) method (Zhang et al., 2022), in addition to the ICAO Aircraft Engine Emissions Databank.

In order to correct for known biases in ERA5 humidity at cruising altitudes (Agarwal et al., 2022; Meijer, 2024), we further

configure PyContrails to use "histogram matching" to scale the humidity values in the weather data to match quantiles of in-situ measurements from the In-service Aircraft for a Global Observing System (IAGOS) (Petzold et al., 2015).

We rely on the default PyContrails setting for the maximum contrail lifetime, which is 20 hours, although the longest lifetime we see in our dataset is 13 hours.

## A9 Advection Time Error in Synthetic Dataset Generation

In the RunCocip subroutine in Section 2.3, we configure CoCiP to provide outputs on 30 second intervals and map the true satellite capture time to the nearest CoCiP output time, which is a maximum of 15 seconds away. At 75 ms$^{-1}$ wind speeds this would incur 1125 m of advection error, which is only slightly more than half of the GOES-16 ABI nadir resolution. We measured the distribution of ERA5 first ensemble member wind speeds experienced by all flights in the dataset and found that 75 ms$^{-1}$ is more than 3 standard deviations above the mean (mean = 25.3 ms$^{-1}$, stddev = 15.7 ms$^{-1}$). Even the maximum

wind speed in the dataset (103 ms$^{-1}$) results in subpixel error. We therefore consider this error to be negligible for the purposes of our analysis.

## A10 ReprojectCoCiP

For each flight waypoint that forms a contrail at a given timestep, CoCiP models the contrail in a 3d space with axes x, y, z and origin at the advected waypoint location (units are meters). z is the vertical axis pointing from the center of Earth to the

contrail. x points along the horizontal plane orthogonal to z, along the contrail's length. y is the normal to x in the horizontal plane, with the positive direction to the right of the advected flight heading. Within this space, the contrail cross-section for a given waypoint is modeled as a 2D anisotropic Gaussian in the yz plane with covariance matrix

$$\sigma = \begin{bmatrix} \sigma_{yy} & \sigma_{yz} \\ \sigma_{yz} & \sigma_{zz} \end{bmatrix}. \tag{A1}$$

To obtain the cross-section parameters at locations between two waypoints, the Gaussian's parameters are interpolated linearly.

CoCiP defines the width, $B$, and depth $D$ as





$$B = (8\sigma_{\mathrm{yy}})^{\frac{1}{2}}, \tag{A2}$$

$$D = (8\sigma_{\mathrm{zz}})^{\frac{1}{2}}, \tag{A3}$$

(see Section 2.1 of Schumann (2012) for more details) and uses that width to compute optical depth properties. In order to render off-nadir contrails in the perspective of a geostationary satellite, we need to recompute these values using the viewing

ray of the instrument. We therefore compute a vector from each contrail waypoint to the satellite and project it onto the yz plane, calling it $z_{\mathrm{sat}}$. We then rotate $\sigma$ such that $z_{\mathrm{sat}}$ is now the positive vertical axis, and then recompute width, depth, and contrail optical depth from the resulting covariance matrix. This process is demonstrated in Figure A3.

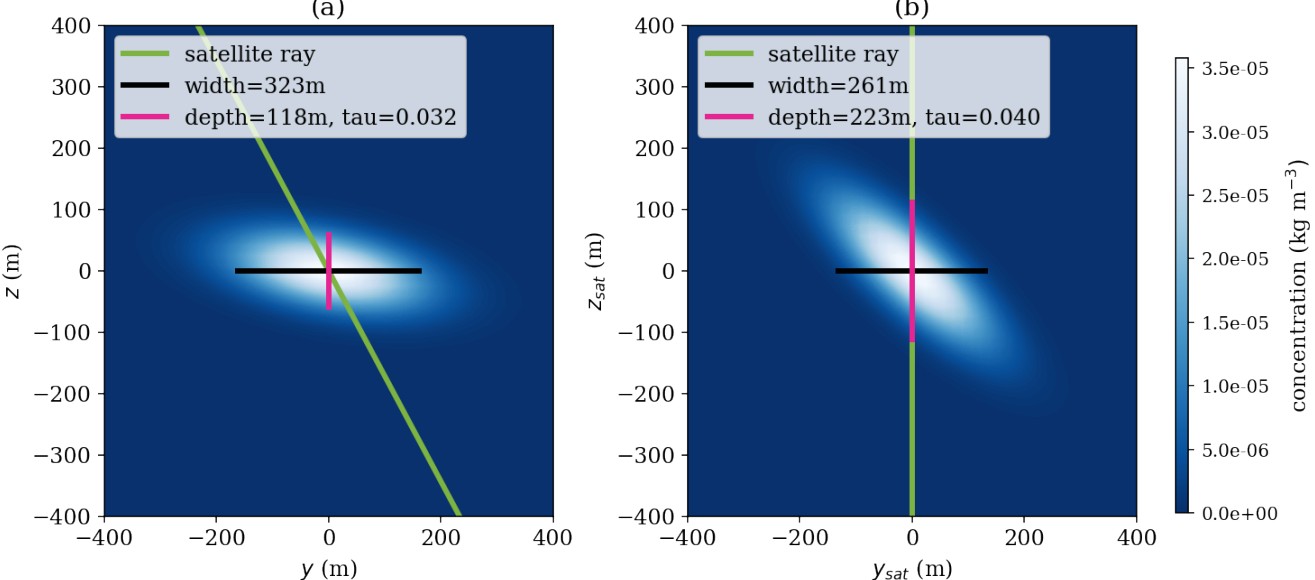

**Figure A3.** A simulated CoCiP plume ice particle concentration profile, placed at latitude $37°$ north, longitude $120°$ west, shown **(a)** in the native CoCiP coordinate system, and **(b)** recomputed from the GOES-16 perspective.

### A11   FilterUndetectable

The FilterUndetectable subroutine of the synthetic data generation pipeline aims to compute a detectability mask that filter's

CoCiP's outputs to just what the Ng et al. (2023) detector would detect. The criteria it uses are:

1. The maximum optical depth of the contrail cross-section at the waypoint must be above a threshold $T_{\tau}$.

2. As a proxy for other clouds limiting detectability, we require that the CoCiP-reported longwave radiative forcing be above a threshold $T_{\mathrm{rflw}}$.





3. The contrail width must be *below* a threshold $T_{\text{Bmax}}$. This is somewhat counter-intuitive, since generally we think of
   contrails being too narrow to be seen in geostationary imagery. The contrails that are too narrow will be filtered out
   naturally in the subsequent Rasterize subroutine, so we do not address them here. Here we are using width as a proxy for
   linearity. The labelers who labeled the detector training data were instructed to only label line-shaped contrails, because
   contrails that are past their linear phase are generally hard to distinguish from natural cirrus. Appendix A13 discusses
   why it is reasonable to use width as a proxy for linearity for the purposes of detectability.

4. The contrail length must be substantially larger than its width. The labeler instructions in Ng et al. (2023) required that
   a contrail be 3 times as long as it is wide. To simulate this, we say that a given contrail waypoint will only be detectable
   if it has a certain number of neighboring waypoints $n = b * C_{\text{l/B}}/C_{\text{Tflight}}$ that are also visible according to the previous
   criteria, where $b$ is the average width of the contrail detection in question, $C_{\text{l/B}}$ is a ratio of flight seconds per m of width,
   and $C_{\text{Tflight}}$ is the number of seconds between flight waypoints after the initial resampling described previously. In order
   to tolerate small gaps in visibility, we search for the $n$ visible neighbors in a window of $n * C_{\text{ndil}}$ waypoints in either
   direction, where $C_{\text{nfrac}} >= 1$ defines the amount by which we dilate the search window.

As a minor optimization, we qualitatively determined that we most closely match human detectability if we slightly loosen
these criteria. Specifically, if a contrail in the given timestep has any waypoints that pass all four criteria, we keep all of its
waypoints in the contrail that pass criteria 2 and 3. This helps avoid unnatural single-waypoint contrails and hard boundaries
that are not due to occlusion.

## A12 AdaptToDetector

Here we detail the adaptations made directly to the CoCiP outputs to better reflect the behaviors of the Ng et al. (2023) detector.

The first is related to condition 3 of the detectability criteria in the FilterDetectable subroutine (see Appendix A11). We
found that using a fixed width upper bound results in contrails that suddenly disappear in unrealistic ways. The reality is that
there is a decay in odds of detection as a contrail ages and becomes more dispersed and less linear. Since the value that will
eventually be rasterized in the Rasterize subroutine is directly derived from optical depth, to simulate this, we decay CoCiP's
optical depth $\tau$ based both on width and age of the contrail. Specifically we apply:

$$\tau' = \tau * \left(1 - \max\left(0, B - \frac{T_{\text{Bmax}} - C_{\text{decay}}}{C_{\text{decay}}}\right)\right) * \min\left(1, e^{T_{\text{age}} - a}\right), \tag{A4}$$

where $B$ is the contrail width in meters, $a$ is the contrail age in hours. This decays $\tau$ linearly to 0 as the contrail width grows
from $T_{\text{Bmax}} - C_{\text{decay}}$ to $T_{\text{Bmax}}$, and additionally applies a multiplicative exponential decay based on the contrail age, once it
becomes older than $T_{\text{age}}$ hours. See Appendix A13 for further discussion.

The second adaptation is a reflection of how the training data for the detector was labeled. Specifically, the tool that labelers
used to draw polygons around contrails did not allow for the polygon to be less than 2 pixels wide. Consequently, the contrail
masks in the OpenContrails dataset (Ng et al., 2023) are never less than 2 pixels wide, and the detector model learned this




behavior, even for contrails that are far narrower than what one would expect for a 2 pixel-wide contrail seen in the GOES-16 ABI. To instill this behavior in SynthOpenContrails, we artificially pad the widths (only after all of the aforementioned width-based filtering and adaptation) of contrails whose CoCiP-predicted widths are between $T_{\mathrm{padmin}}$ and $T_{\mathrm{padmax}}$ by $C_{\mathrm{pad}}$.

**A13   Width and Age Decay of Synthetic Detectability**

In both the FilterUndetectable and AdaptToDetector subroutines of the synthetic dataset generation described in Section 2.3,
CoCiP's predicted contrails growing very wide are interpreted as a proxy for the contrails becoming undetectable. Additionally, in Equation A4, contrail age being over a threshold is multiplicatively applied as a further decay of detectability. The justification for this lies in how CoCiP makes some simplifying assumptions that some physical processes can be partially or totally ignored because they apply only at smaller spatial scales than the contrail plume, whose cross-section CoCiP requires to be Gaussian. One of these processes is sub-grid-scale (SGS) turbulence. CoCiP takes SGS turbulence into account only as a factor
that slightly increases the rate of ice particle loss, which then is applied uniformly across the contrail cross-section, leading to a decrease in optical depth and total contrail lifetime (Section 2.12 of Schumann (2012)). While for CoCiP's own purposes this assumption of applying the effects of SGS uniformly across the contrail may be fine, for the purposes of detectability it creates a challenge, particularly when the contrail is wide enough to span multiple satellite pixels: non-uniformity in rates of ice particle extinction across the contrail would result in local variation in optical depth. This could manifest as irregular
widths, gaps, and deviation of the width-wise center of the contrail away from the advected waypoint location, all of which would contribute to becoming undetectable, and none of which are modeled by CoCiP. The width-based decay is introduced here as a simplified model of detectability loss due to these processes.

There are other approximations that CoCiP makes that likely also affect detectability. Since, by definition, SGS turbulence cannot be directly read from the NWP data, its magnitude is inferred to grow quadratically with wind shear (Equation A20
of Schumann (2012)), as derived from the Richardson number. CoCiP does not directly compute wind shear from the NWP values either, but instead applies an enhancement factor (Equation 39 of Schumann (2012)), which is a function only of contrail depth, to what would be computed directly from the NWP. This enhancement is inspired by Houchi et al. (2010), and it notably results in matching radiosonde shear measurements at a distribution level, but not in the specifics. In CoCiP, a contrail's width increases with age as a function primarily of both wind shear and vertical diffusivity (see Equation 29 of Schumann (2012)).
Vertical diffusivity is also a function of turbulence, but in this case CoCiP uses a fixed value for turbulence (Equation 35 of Schumann (2012)). Taken together, all of these simplifying assumptions, coupled with the relatively low spatio-temporal resolution of the weather data, result in the CoCiP contrails growing wider at a relatively uniform rate along the length of the contrail, when in fact there should often be more variation. This effect compounds with contrail age, and is not strictly dependent on contrail width; the age-based decay therefore aims to capture this effect.

**A14   Beer-Lambert Law Applicability**

In the Rasterize subroutine of Section 2.3 we apply the Beer-Lambert law (Beer, 1852) to map CoCiP's optical depth to opacity, $\kappa$, which is then directly rasterized and thresholded to determine a final synthetic contrail mask. CoCiP's optical depth



is computed at 550 nm wavelength, whereas the bands the detector uses are in the thermal infrared range (8.5 - 12 $\mu$m). Per Schumann et al. (2012), the absorption optical depth in the thermal infrared range is approximately half of the 550 nm optical
depth. Since the final mask will be determined by thresholding $\kappa$, this mismatch will simply result in a different threshold value being used. We find it reasonable to apply the Beer-Lambert law here, despite contrails not being a purely absorbing-medium, since in the thermal infrared bands the contribution of scattering to the optical depth of high ice clouds is negligible when compared to that of absorption (Jin et al., 2019). This would not hold if shortwave bands were used for detection.

## A15   Rasterizing CoCiP in a Geostationary Perspective

Here we detail the process of rasterizing CoCiP outputs in the perspective of a geostationary satellite. This is an adaptation of Appendix A12 of Schumann (2012).

At this stage we still operate on just a single flight and a single timestep. We first parallax correct each CoCiP waypoint location to the surface latitude and longitude where the satellite would see it. Due to an error, for this process we used the altitude output from PyContrails, which uses an International Standard Atmosphere (ISA) approximation to convert pressure
to geometric altitude, when it would have been more correct to use geopotential to compute it. In Appendix A16 we show that this error is negligible for our purposes. We then map those onto the satellite pixel grid, but supersampled (Akenine-Moller et al., 2019) to 8 times the true resolution in order to minimize aliasing in the final raster. For each pair $(i,j)$ of adjacent waypoints, with optical depths $(\tau_i', \tau_j')$ and widths $(B_i, B_j)$, we take a square kernel of pixels that includes both waypoints and all pixels that are within $\max(B_i, B_j)$ from the segment joining the waypoints. Within this kernel, we lookup the latitude and
longitude of the centers of each pixel, noting that the grid will be somewhat irregular due to the curvature of the Earth. We then compute the distance, $s$, in meters, from the center of each pixel to the closest point on the segment, and also the fraction $\alpha$ (this is called $w$ in Schumann (2012), but we want to avoid confusion with other variables of that name here) of the distance along the segment from $i$ to $j$ of this closest point. Following Appendix A12 of Schumann (2012) we can then compute the optical depth of the contrail in this pixel as

$$\tau_{\text{ras}} = \left(\alpha \tau_i' + (1-\alpha)\tau_j'\right) * \left(\frac{4}{\pi}\right)^{1/2} * \exp\left(-\frac{4s^2}{(\alpha B_i + (1-\alpha)B_j)^2}\right). \tag{A5}$$

Having populated the kernels for each pair of waypoints, we can then combine them back to the supersampled pixel grid, taking a maximum over different waypoint pairs that contribute to the same pixel. We can then downsample to the native satellite resolution, and convert to opacity: $\kappa_{\text{ras}} = 1 - e^{-\tau_{\text{ras}}}$.

## A16   Pressure Altitude Conversion

We analyzed the impact of applying parallax correction of advected flight waypoint locations relying on International Standard Atmposphere (ISA) approximations for converting pressures to geometric altitudes rather than using geopotential heights to be more precise. We took the PyContrails outputs for each waypoint at each timestep where it contributed to the final contrail masks in the SynthOpenContrails validation set. We measured the Euclidean distance in the GOES-16 ABI's native resolution



for infrared bands between the subpixel location that the waypoint would project to using the ISA altitude and the geopotential
height. We found the mean distance to be 0.200 pixels and the standard deviation to be 0.066 pixels. This suggests that the
error it contributes is likely negligible for the purposes of SynthOpenContrails, and likely also more generally for the class of
contrail-to-flight attribution in geostationary satellite imagery algorithms considered in this study.

### A17   HandleOutbreaks

In principle the CombineRasters subroutine should produce a final contrail mask, except that this results in certain large areas
that are almost entirely marked as contrails and the individual contrails can not be identified. These are usually in areas where
the satellite imagery does have very high contrail density, which we hereafter refer to as "contrail outbreaks." In the satellite
imagery contrail outbreaks do often appear as large areas with amorphous cirrus cloud cover no longer identifiable as individual
contrails other than certain areas of greater optical depth that are still linear. Generally speaking, the Ng et al. (2023) detector
will only identify these greater optical depth contrails in outbreak scenarios. It is also likely that the true contrail density is a bit
lower than what CoCiP predicts, since CoCiP does not model the inter-flight effects, where the formation of the first contrail
slightly dehydrates the atmosphere, making the next contrail less likely to form (Schumann et al., 2015). As the objective is
to simulate the detector's behavior, whether or not CoCiP is over-predicting, we need to modify the outputs in these outbreak
areas.

To accomplish this, for each contrail pixel in our mask we compute a local "contrail density" $\rho$ as the fraction of contrail
pixels in the $C_{\sigma k} \times C_{\sigma k}$ pixel neighborhood that surrounds it. We apply a logistic function

$$\sigma(\rho) = 1 - \frac{1}{1 + \exp(-C_{\sigma\gamma}(\rho + C_{\sigma\beta}))}, \tag{A6}$$

where $C_{\sigma\gamma}$ and $C_{\sigma\beta}$ are parameters controlling the rate and domain of scaling applied. We then scale the opacity for that pixel
as $\kappa'_{\mathrm{ras}} = \frac{\sigma(\rho)}{\sigma(0)}\kappa_{\mathrm{ras}}$. This process is demonstrated in Figure A4.

### A18   Linearizing Synthetic Contrail Masks

Here we detail the Linearize subroutine of the synthetic contrail generation process, which takes a single frame of rasterized
synthetic contrail opacities and maps them to individual line segments, each representing a single contrail.

First we reproject our rasterized contrail opacities into overlapping square 256 x 256 pixel tiles in the Universal Transverse
Mercator (UTM) projection, with the UTM zone selected per-tile, with a resolution of approximately 500 km of surface
distance along each side of the tile. The Ng et al. (2023) detector itself consumes tiles of satellite radiances with exactly the
same reprojections applied, in order to avoid many of the distortion issues in the native projection caused by being farther
from the satellite nadir. We then threshold the reprojected opacities using $1 - e^{-T_\tau}$, as before. We found that using OpenCV's
LineSegmentDetector, as described in Ng et al. (2023), sometimes poorly linearizes wider contrails (both synthetic and real)
producing two line segments at either edge of the contrail mask, rather than the desired single line segment in the middle. We
therefore use the line-kernel convolution-based algorithm described in McCloskey et al. (2021), which is based on Mannstein



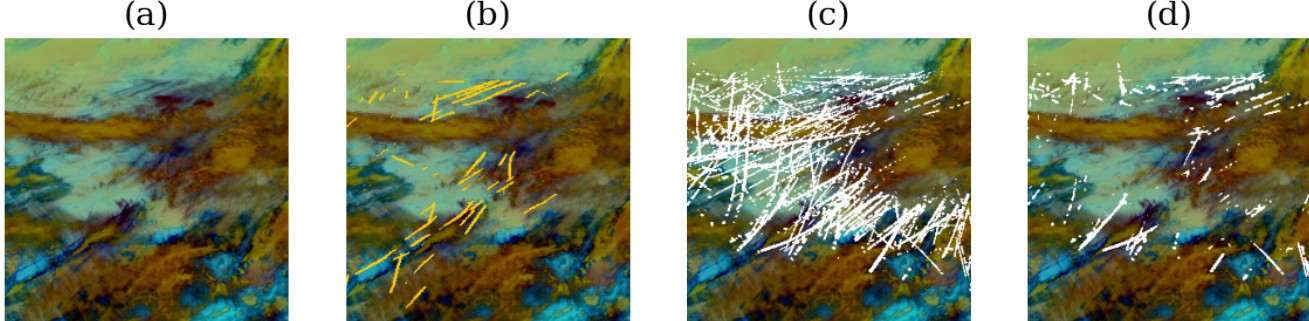

**Figure A4.** A demonstration of the special handling for contrail outbreaks. In **(a)** we show an Ash false color GOES-16 ABI image from 11 February, 2020 at 22:00 UTC, centered just off the coast of Delaware. Many contrails are visible in dark blue, along with some thinner cirrus that may also have originated as contrails. There are also mixed-phase clouds shown in brown that make some contrails hard to see. In **(b)** we overlay the detections from Ng et al. (2023) in yellow. In **(c)** we overlay the results of our synthetic contrails generation before Equation A6 is applied in white. The density of contrail pixels is substantially higher than in (b). In **(d)** we show the same thing, but after Equation A6 is applied. The density of contrail pixels is much more similar to (b).

et al. (1999), both for linearizing the real detector outputs and our synthetic contrail mask tiles. An additional benefit this approach provides is that this linearization algorithm declares which mask pixels in the tile correspond to each linear contrail that it produces, which allows us to maintain a mapping of the CoCiP output properties contributing to each pixel corresponding to each linear contrail. We then invert the UTM reprojection for these tile pixels to resolve which flights produced the pixels that comprised each linear contrail. In some cases more than one flight is deemed to have contributed to a single linear contrail,
either due to actual contrail overlap, or erroneously due to the linearization algorithm. In these cases we take a winner-takes-all approach and assign the linear contrail to the flight that is responsible for the most pixels. The final step is to deduplicate linear contrails from overlapping regions of neighboring tiles, and for this we exactly follow the process described in Ng et al. (2023).

**A19 Metric definitions**

Section 2.4.2 defined a set of metrics that can be measured for an attribution algorithm's outputs on SynthOpenContrails. Here
we further detail how each metric is computed.

Each metric is composed of cell values from Table A1. The values in each per-contrail cell, **A, B, C** are computed by joining each linear contrail in the benchmark dataset with any flight attributions that an algorithm made for that contrail. Each linear contrail will have 0 or more attributions associated with it. If there are 0 attributions, **C** is incremented. For each attribution, if the flight is the same as the true flight that formed the linear contrail, **A** is incremented. Otherwise **B** is incremented. The
per-flight cell values, **D, E, F**, are similarly computed by grouping together all linear contrails in the benchmark dataset by the flight that formed them and similarly grouping all attributions by attributed flight. Each flight will then have 0 or more linear contrails that it formed and 0 or more linear contrails attributed to it. If both are 0 then we ignore this flight. If the flight formed





linear contrails and there are attributions to it, we increment **D**. If it formed linear contrails but there were no attributions, we increment **F**. If there were attributions but it did not form any linear contrails, we increment **E**.

**Table A1.** A contingency table used for metric computation.

| | | flight $x$ formed linear contrail $y$ | | | | flight $x$ formed any linear contrail | |
|---|---|---|---|---|---|---|---|
| | | Yes | No | | | Yes | No |
| flight $x$ attributed to linear contrail $y$ | Yes | **A** | **B** | flight $x$ attributed to any linear contrail | Yes | **D** | **E** |
| | No | **C** | | | No | **F** | |

Once the table is populated, we compute the derived metrics using the specified formulae:

- **Per-contrail recall**: **A/(A + C)**

- **Per-contrail precision**: **A/(A + B)**

- **Per-flight recall**: **D/(D + F)**

- **Per-flight precision**: **D/(D + E)**

## A20    Parameter Values for SynthOpenContrails Generation

The specific parameters used for generating the SynthOpenContrails dataset are shown in Table A2.

### A21    Statistics and Qualitative Assessment of the SynthOpenContrails Dataset

We show some top-level statistics comparing SynthOpenContrails to real detections for the same space-time regions, per dataset split in Table A3. We can also look at the per-frame contrail-pixel and linear contrail counts, which are shown for the validation set in Figure A6. The pixel counts in aggregate are very similar, with there being a few time spans where SynthOpenContrails has meaningfully more contrail pixels, and one notable span where the real detection masks have many more pixels. On the whole, the peaks and valleys align very well. The linear contrail counts also match the overall trends, but the total counts are somewhat farther apart. The vast majority of the discrepancy comes from a single time span with a large outbreak, where our adjustments to reduce the number of synthetic contrails in outbreaks seems to have overcompensated. We hope that future work can find a better approach to handling these cases. We can also compare the lengths of the linear contrails between real data and SynthOpenContrails. The distribution of lengths is shown in Figure A5. The distributions match quite well, but SynthOpenContrails skews slightly shorter.

We also qualitatively evaluated the dataset for how well it matches the Ng et al. (2023) detector outputs for the corresponding GOES-16 ABI scans in terms of geographic distribution of contrails, temporal dynamics, and the appearance of individual



**Table A2.** The parameter values used for generating SynthOpenContrails.

| Parameter | Description | Value | Units |
|---|---|---|---|
| $C_{\text{Tflight}}$ | Flight paths are resampled to this frequency before being input to CoCiP | 5 | s |
| $T_\tau$ | A threshold on CoCiP's contrail optical depth used both for determining early stage contrail detectability and for thresholding the final raster to produce a contrail mask | 0.04 | unitless |
| $T_{\text{rflw}}$ | A minimum threshold on the CoCiP-predicted longwave radiative forcing used to determine if a contrail segment will be detectable | 7 | $\text{Wm}^{-2}$ |
| $T_{\text{Bmax}}$ | The maximum width of a contrail that is likely to be linear enough to be detectable | 12500 | m |
| $C_{\text{l/B}}$ | A ratio of contrail flight seconds to meters of contrail width, used to specify how many neighboring waypoints need to have formed a contrail for a waypoint to be detectable | 0.01 | $\text{sm}^{-1}$ |
| $C_{\text{ndil}}$ | A factor by which the search window for neighboring contrail-forming waypoints is dilated in order to tolerate small gaps | 1.43 | unitless |
| $C_{\text{decay}}$ | The rasterized optical depth is decayed linearly to zero between $T_{\text{Bmax}} - C_{\text{decay}}$ and $T_{\text{Bmax}}$ | 5000 | m |
| $T_{\text{age}}$ | The contrail age above which the rasterized optical depth is decayed exponentially | 1.5 | h |
| $T_{\text{padmin}}$ | The minimum contrail width for which a padding is applied to the width before rasterization | 500 | m |
| $T_{\text{padmax}}$ | The maximum contrail width for which a padding is applied to the width before rasterization | 2500 | m |
| $C_{\text{pad}}$ | The amount by which the contrail width is padded before rasterization when the width is between $T_{\text{padmin}}$ and $T_{\text{padmax}}$ | 1000 | m |
| $C_{\sigma k}$ | The size of the kernel used for computing contrail pixel density for outbreak handling | 49 | px |
| $C_{\sigma\gamma}$ | Controls the rate of scaling applied in Equation A6 | 6 | unitless |
| $C_{\sigma\beta}$ | Controls the domain of scaling applied in Equation A6 | -0.1 | unitless |

contrails in the mask. Of these characteristics, all appeared qualitatively similar, in the authors' opinions, with the exception of certain aspects of individual contrail appearance, as expanded below. To accomplish this comparison, we rendered the SynthOpenContrails mask and the Ng et al. (2023) detector mask in different colors on top of the corresponding false-color GOES-16 ABI imagery. An example of this is shown in Figure 7. We use the Ash color scheme as used previously in Kulik (2019); Meijer et al. (2022); Ng et al. (2023) to map infrared radiances to RGB imagery that makes optically thin ice clouds,

like contrails, appear in dark blue. We observe that the SynthOpenContrails contrail detections appear generally in the same regions as the real detections, but there is far from perfect alignment. While there are a few instances where the SynthOpenContrails mask actually exposes contrails visible in the Ash imagery that the detector missed, the vast majority of the time the real detector better reflects what a skilled human would see in the satellite imagery. This is consistent with previous work (Gierens et al. (2020); Agarwal et al. (2022); Geraedts et al. (2024)) which finds that NWP data has difficulty predicting con-

trail formation at the per-flight level. The temporal dynamics frame-to-frame do appear qualitatively similar to those of real detections. We reiterate that for the purposes of our contrail-flight attribution system benchmark, it is not necessary that SynthOpenContrails be correct as to which flights actually formed contrails; it is only necessary that the distribution of properties





of the synthetic data are similar to the real data. The individual synthetic contrails look qualitatively fairly similar to their detector-produced counterparts in overall form. The most noticeable difference is that the synthetic contrails have a slightly

higher rate of appearing discontinuous. This may be a result of CoCiP evaluating each waypoint pair independently, in contrast with the smoothing tendencies of the detector. This could perhaps be rectified by a slight blurring of the CoCiP outputs across neighboring waypoints prior to rasterization. The fact that more discontinuous contrails are present in SynthOpenContrails masks does not affect CoAtSaC, as it only utilizes the linearizations of the contrail mask, which are for the most part unaffected by the discontinuities. Any attribution algorithm that directly uses the pixels within the contrail mask, however, may be

affected, and this discrepancy should therefore be explored in greater detail for such approaches.

We can also measure the distributions of various other properties of these synthetic contrails by propagating the metadata from CoCiP through the rasterization and linearization processes. We look at these in depth in Section 3.1.1 in the context of slicing performance of attribution algorithms along various axes. In Appendix A22 we document the fraction of CoCiP contrails that are rasterized and linearized as a function of contrail age.

**Table A3.** Statistics of the SynthOpenContrails splits. Values for the corresponding detector outputs on real satellite imagery are in parentheses, where applicable.

|  | **Train** | **Validation** | **Test** |
| --- | --- | --- | --- |
| Satellite Frames | 4,536 | 1,512 | 1,505 |
| Contrail Pixels | 76,698,642 (74,948,579) | 24,244,788 (24,225,800) | 26,206,579 (23,868,781) |
| Linear Contrails | 1,041,126 (1,502,508) | 326,048 (482,967) | 489,770 (353,760) |
| Unique Flights Contributing to Contrail Pixels (per-frame) | 2,205,919 | 678,224 | 719,265 |
| Unique Flights Contributing to Linear Contrails (per-frame) | 606,127 | 189,514 | 205,359 |

**A22   Contrail Lifetimes in SynthOpenContrails**

In Figure A7 we show the lifetimes of contrails as they are filtered through the stages of the SynthOpenContrails generation pipeline. We want to stress that this figure is documentation of the dataset, and not any sort of scientific claim of the true detectable lifetimes of contrails. We see that the peak age for both rasterization and linearization is around 1.5 hours after formation, where 35% of total contrail-forming flight km are rasterized, and 17% are linearized. These numbers increase to

43% and 21% if they are measured relative to the just the contrail-forming flight km that are not yet extinct. If, rather than weighting by contrail-forming flight km, we weight by CoCiP-predicted energy forcing (EF) of the contrail, we see a peak of 48% of EF being rasterized and 26% linearized. CoCiP shows the rate of contrail extinction being fairly linear in time, while the synthetic rasterizations and linearizations fall-off exponentially, presumably due to the application of Equation A4. We note that this is very different than what Driver et al. (2024) found. There are a few likely reasons for this. One is that we do



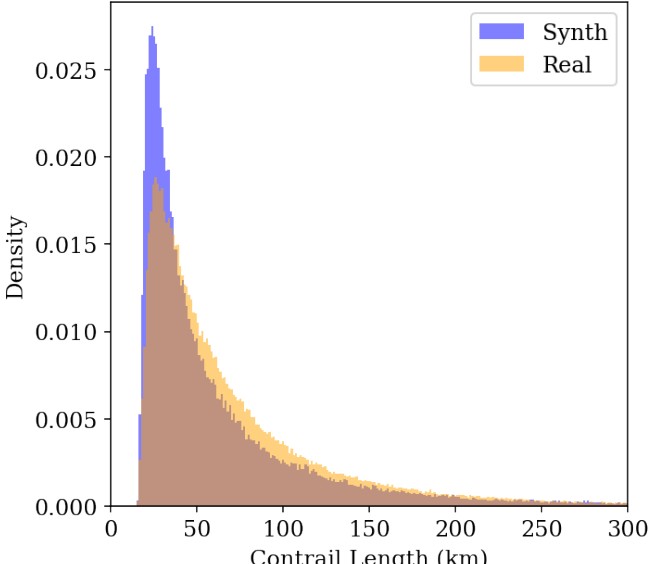

**Figure A5.** Histograms of linear contrail lengths in the space-time region defined by the validation split. The orange distribution is from the detector run on real satellite imagery and the blue is SynthOpenContrails.

not attempt to rasterize the full lifetime of each contrail. This data is generated from the test set, which rasterizes 9 hour time spans, but some of the contrails formed before the start of the time span and others form close to the end, so their chances of rasterization and linearization are lower than contrails whose full lifespan is captured. Similarly, some of our contrails advect into or out of the spatial region that is rasterized over the course of their lifetime, also limiting their rasterized lifetime. Furthermore, Driver et al. (2024) essentially used the CoCiP outputs without modification, whereas we are manipulating them.

The HandleOutbreaks subroutine in in particular likely has a major impact here. Finally, it is important to point out that Driver et al. (2024) was operating under idealized background conditions of clear skies and at-nadir resolution, whereas we are not. Figure A7 does, on its own, indicate that there is room for improvement on SynthOpenContrails's handling of very young narrow contrails with very high optical depth. Currently they can still be rasterized if they happen to align well with the centers of the pixels, even with supersampling applied. In fact, we see a few rasterized pixels corresponding to contrails that are just a

few seconds old, which is likely not realistic.

### A23 Missing Flights Sensitivity

We performed a sensitivity analysis on the assumption we made in Section 2.4.3 that we should exclude a random sample of 20% of the flights used to generate SynthOpenContrails. In Figure A8 we show the results of rerunning the single-frame and CoAtSaC algorithms with different fractions of flights available, without retuning any parameters. Unsurprisingly, both

contrail- and flight-level recall increase linearly with the fraction of flights available to them. Precision is also affected for CoAtSaC, but substantially less than recall. Somewhat surprisingly, the single-frame algorithm's precision is essentially un-





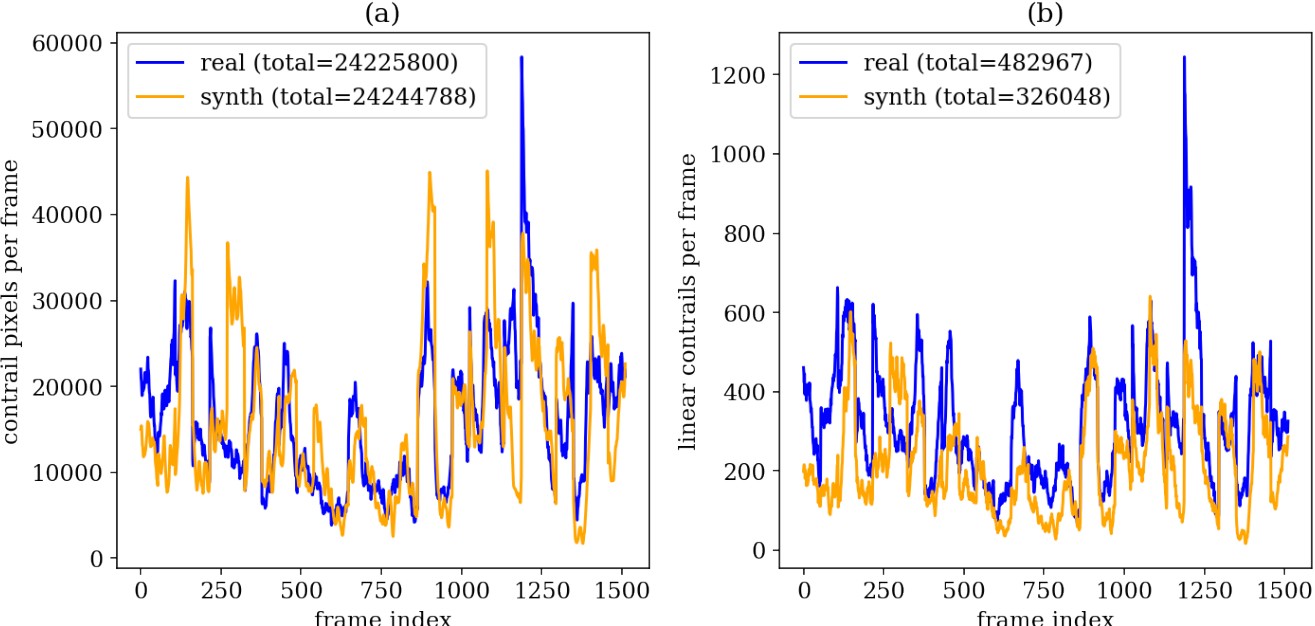

**Figure A6.** Comparisons of contrail statistics between real data and SynthOpenContrails, shown for satellite frames in the validation split. Blue is from running our detector on real GOES-16 ABI imagery and red is from SynthOpenContrails. (**a**) The number of contrail pixels per frame. (**b**) The number of linear contrails per frame.

changed by the fraction of flights available. In general, the relative ordering of the algorithms does not change for any of the metrics depending on the fraction of flights available. If the true fraction of flights missing from the FlightAware.com database were to become available, and the missing flights turn out not to have a bias with respect to contrail formation properties, then
these results might provide a more accurate estimate of the real-world performance of these algorithms.

**A24   Parameter Values for CoAtSaC**

Table A4 shows the values used for each parameter of CoAtSaC.

**A25   Changes to the Tracking Algorithm**

For the tracking algorithm in Chevallier et al. (2023), we made the following changes to make it work with the SynthOpen-
Contrails. The advection method used was a reimplementation of that used in Geraedts et al. (2024), using ERA5 nominal data on pressure levels. The tracking algorithm was designed to operate on contrail instance masks, which is not an explicit output of SynthOpenContrails. It does implicitly provide something similar, since the Linearize subroutine already calculates the set of mask pixels it believes correspond to each linearized contrail, but these were still qualitatively quite different from the instance masks used in Chevallier et al. (2023). We therefore slightly adapted the tracking algorithm to operate directly on the



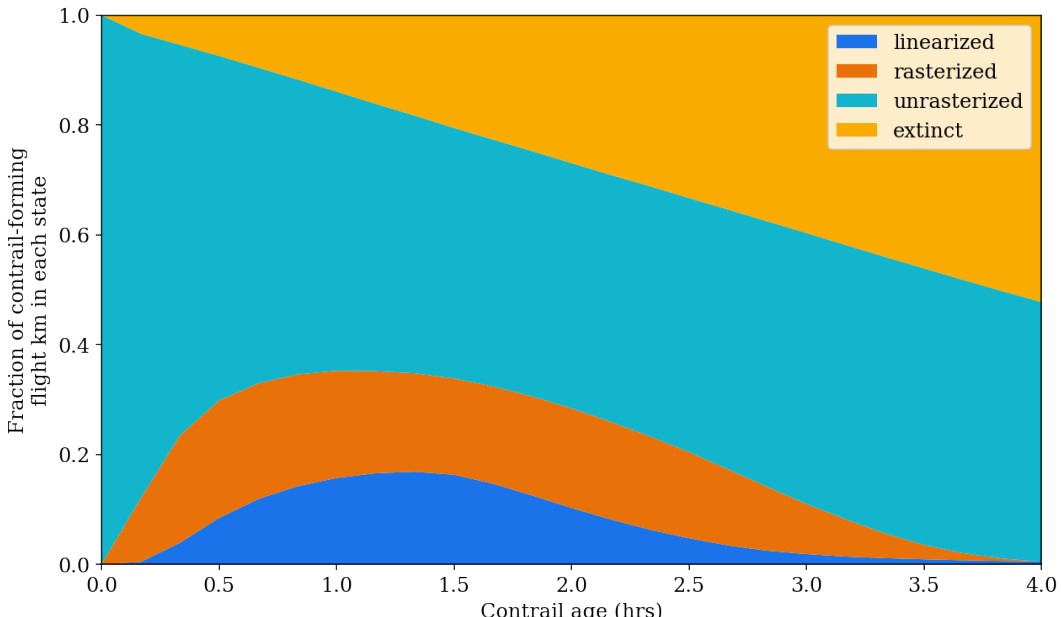

**Figure A7.** A plot showing the distribution of "states" of CoCiP-predicted contrails in the SynthOpenContrails test split over the first 4 hours of their lifetimes. A "contrail" unit here is a contiguous set of flight waypoints that CoCiP says forms contrails. The entire contrail is labeled as being rasterized if at least one waypoint in that range contributes to the synthetic contrail mask. It is considered linearized if a linear contrail is found that is determined to be formed by a set of waypoints that overlaps the CoCiP waypoint range. The y-axis in the plot scales each contrail's contribution by its fraction of total flight distance that CoCiP says formed contrails. This plot should not be interpreted as a scientific finding of the detectability of real contrails over their lifetime, but rather as documentation of the properties of SynthOpenContrails.

linearized contrails provided. This makes it more comparable with the other algorithms used here, but limits its performance somewhat. A future goal is to adapt SynthOpenContrails to emulate an instance segmentation model, as opposed to the global segmentation model emulated in the current approach. The parameters of the algorithm were otherwise kept exactly the same as in the original paper, although they originally were tuned for the GOES-16 ABI's Scan Mode 3, which provided an image every 15 minutes, and SynthOpenContrails uses the current Scan Mode 6a, with data every 10 minutes. Future work should use

the training and validation splits of SynthOpenContrails to further tune the parameters of the tracking algorithm. In Chevallier et al. (2023), the results are presented by applying a threshold on the minimum lifetime of the detected contrail, with the expectation that this improves precision. Here we present all results without that filter. The impact of that decision is discussed in Section A27 and Figure A13. Due to time and computational constraints we were only able to evaluate the tracking algorithm on a subset of the dataset time spans. The included time spans are specified in Table A7.





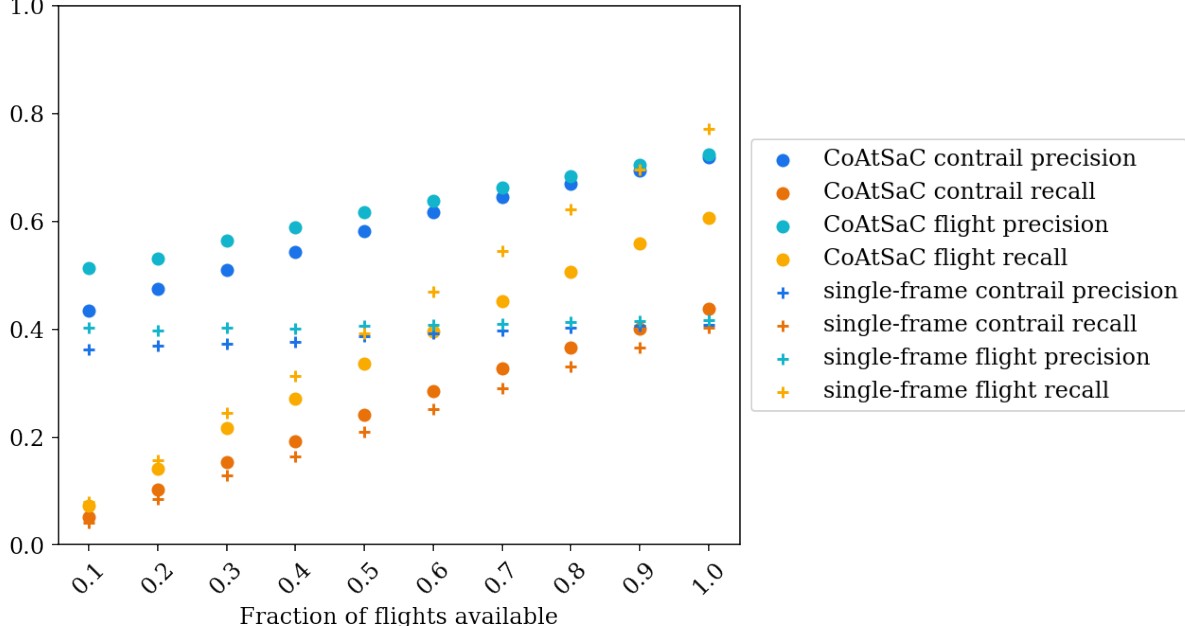

**Figure A8.** Performance of the CoAtSaC and single frame algorithms when different fractions of the total contrail-forming flights are available to them.

**Table A4.** The parameter values used for CoAtSaC.

| Parameter | Description | Value | Units |
|---|---|---|---|
| $T_s$ | Maximum value of Equation 2 considered in CoAtSaC. | 12 | unitless |
| $T_t$ | Maximum allowed temporal gap between single-frame attributions to be considered as a seed pair for the "fitting" stage. | 0.5 | hrs |
| $T_{dW/dt}$ | Maximum allowed slope between single-frame attributions to be considered as a seed pair for the "fitting" stage. | 13 | km/hr |
| $T_{res}$ | The maximum squared residual allowed for a single-frame attribution to be considered an inlier with respect to a fit line. | 3.5 | km |
| $C_{slope}$ | Coefficient of the line slope term in computing Equation 3. | 0.08 | unitless |
| $C_{int}$ | The coefficient of the intercept term in Equation 4. | 0.2 | unitless |
| $C_{sing}$ | The coefficient of the single-frame attribution score term in Equation 4. | 0.3 | unitless |
| $T_b$ | A threshold on the difference between Equation 4 values for different fits that include the same contrail detection, above which the higher scoring fit is rejected. | 0 | unitless |





**A26    Global Flight Recall Losses**

An investigation into the global flight recall losses, seen in Table 1, showed that the flights correctly attributed by the single-frame algorithm but not by CoAtSaC are almost all cases where a contrail was only detected in a single-frame, which CoAtSaC inherently can not attribute correctly. We investigated various ways to add handling for these to CoAtSaC, including simply using the single-frame attributions for any contrail detections not attributed by CoAtSaC, but all attempts resulted in substantially

lower precision. Of note, SynthOpenContrails may artificially amplify the number of contrails that are detectable in only one frame. Specifically, each time span within SynthOpenContrails defines a 4 dimensional box in space and time, and a contrail that advects into the box towards the end of its "linearizable" lifetime, or advects out of the box early in its "linearizable" lifetime, will only have a single linear contrail in the dataset, despite the fact that it would have been linearized in multiple frames if the boundaries of the space-time box had been shifted. While it is reasonable to assume that contrails that are truly

only detectable in one frame have a smaller warming impact than those detected in multiple frames, future research is needed to quantify this.

**A27    Performance as a Function of Contrail Properties**

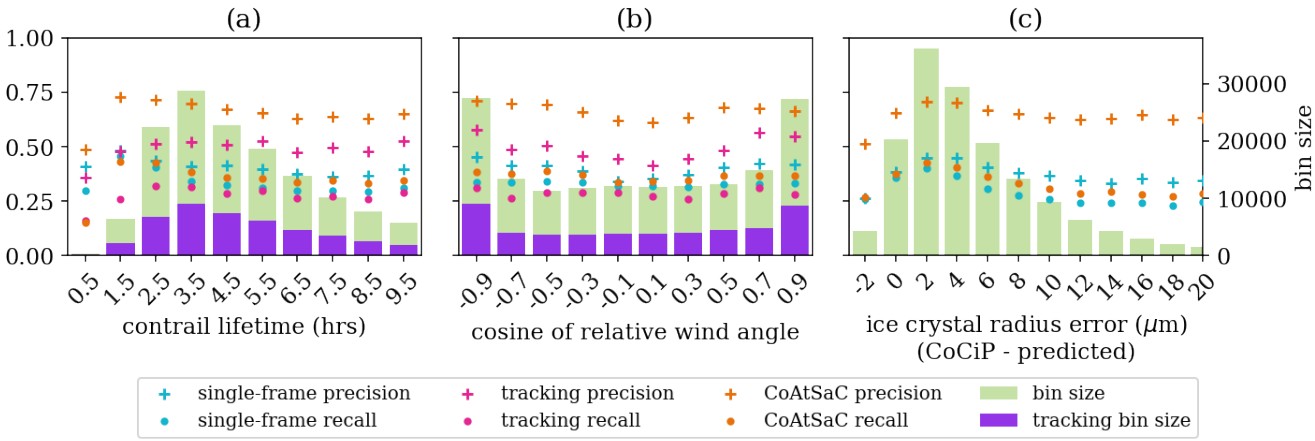

**Figure A9.** Plots in the same style as Figure 8, but binning by some additional properties. **(a)** shows performance binned by total lifetime of the contrail that was detected, as predicted by CoCiP, which is not the same as its detectable lifetime. **(b)** shows performance binned by the cosine of the wind direction relative to the flight heading for the true flight that formed the contrail. **(c)** shows the performance binned by the difference in contrail ice crystal radius between what CoCiP predicts and the prediction from the statistical function of age mentioned in Appenidx A4. The "tracking" algorithm is not plotted here, since we do not have access to its approximation.

Here we present a further analysis of the slicing the performance metrics along axes of various contrail properties, as shown in Section 3.1.1, including Figure 8, Figure A9, and Figure A10.



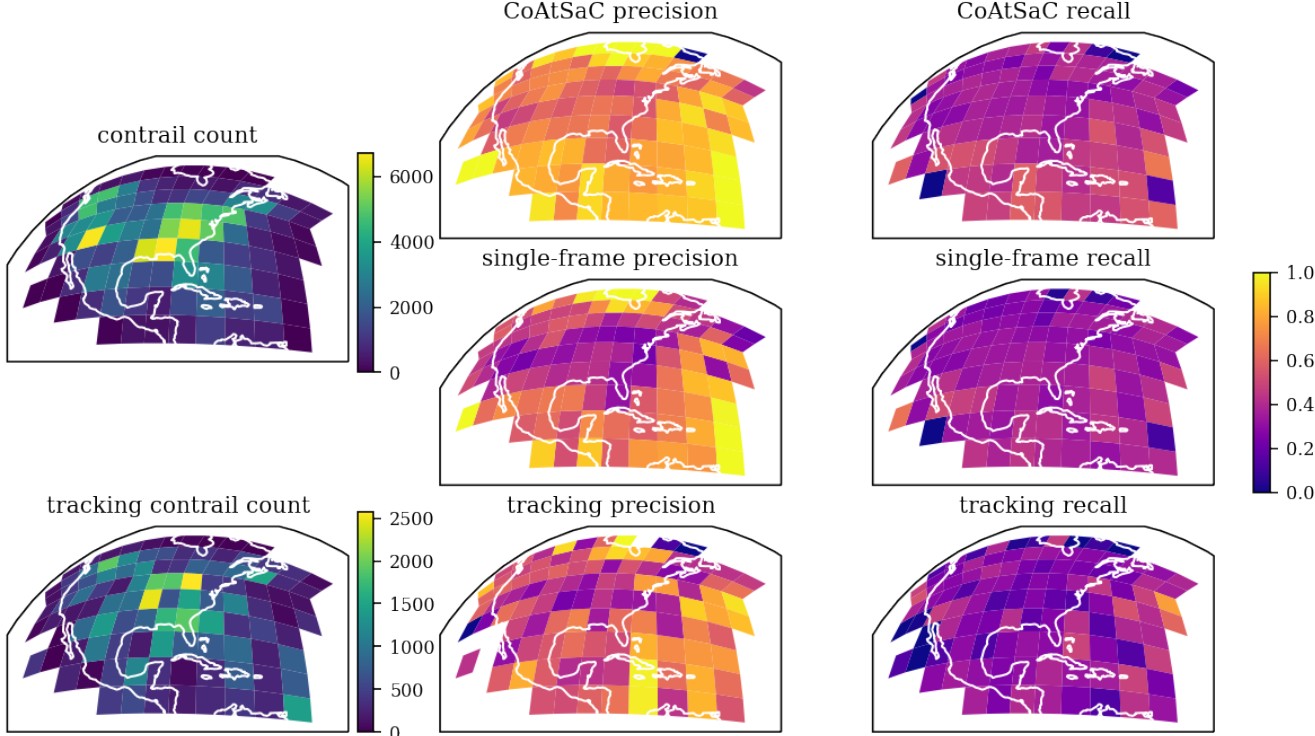

**Figure A10.** Contrail-detection-level performance metrics of each attribution algorithm binned geographically by level 7 S2 Geometry (Google, 2024) cell within the analysis region, rendered from the GOES-16 ABI perspective. Note that the bin sizes are the same for the CoAtSaC and single-frame algorithms, as shown in the "contrail count" plot, but the "tracking" algorithm is only evaluated on a subset of the data, so its bin sizes are shown separately, with a different scale.

One observation is that across virtually all points on these plots, CoAtSaC outperforms the tracking algorithm, which outperforms the single-frame algorithm. The exceptions are the bins with high proportions of contrails only detectable in a single frame, and therefore are inherently impossible for CoAtSaC to handle. Even in these cases, CoAtSaC's precision is still higher, even though its recall is lower. There are isolated additional data points where the single-frame algorithm performance seems higher, but these are generally very small bins where the metrics are not statistically significant.

One of the other dominant effects here is that precision, and to a lesser degree, recall, decreases with higher contrail density for all algorithms. This is most visible in Figure 8(a) and in Figure A10. It is likely also responsible for the performance metric shifts across time-based bins, like when looking at seasonal or diurnal effects in Figures 8(e) and 8(f), since in these cases the higher contrail counts imply higher spatial density. It is notable that the special handling for contrail outbreaks in SynthOpenContrails generation likely has a big impact on how high the density can get, and consequently it may have removed

many contrails where CoAtSaC would have performed the worst.



The effect of contrail density is so dominant that it makes it very hard to answer some other questions using this data. For example, does performance degrade with decreased spatial resolution as you approach the edge of the disk that the satellite captures, perhaps due to increased error in the position of the detected contrails? In our data with the GOES-16 ABI this would be seen in the north-western US and Canada in the upper-left corners of Figure A10, but this region also has above-average
contrail-density, so further investigation is required to disentangle these effects.

Contrail altitude also seems to have an impact on the performance of both algorithms, as can be seen in Figure 8(b). As mentioned in Section 2.3.1, the weather data input to CoCiP was inadvertently missing pressure levels between 450-975 hPa, which likely caused a small secondary peak of contrails around 6 km altitude, due to the weather conditions for contrail formation and persistence being interpolated down to implausibly low altitudes. We opted not to include the roughly 1000
contrails in this plot since it made the rest of the plot hard to read. Within the more plausible altitude buckets, there is a clear trend of performance increasing up until around 11.5 km, and then decreasing again. It is possible that this is again a contrail density effect, but Meijer et al. (2024b) showed that contrail altitudes generally decrease with increasing latitude within this region, and the regions of highest contrail density are in the middle latitudes, so we would expect the opposite effect. It also stands to reason that the performance fall-off seen in Figure A9(c) due to ice crystal radius approximation error
(see Appendix A4) leading to incorrect sedimentation rate has a correlation with altitude. We in fact see this correlation in Figure A11, where the mean error decreases with increasing altitude. We do not see the error going back up above 11 km, though, so it does not explain everything. The tracking algorithm uses a similar but not identical method for approximating ice crystal radius than the other two algorithms. It also shows the performance increasing with altitude, even past the point where performance starts decreasing for the other methods. That said, its bin sizes are very small at the higher end of the altitude
distribution, so that may just be noise.

It may be tempting to conclude that Figure A9(c) combined with Figure A11 indicate that the approximation of ice crystal radius used in the both the single-frame and CoAtSaC algorithm is detrimental. It is important to point out, however, that this is comparing to "ground-truth" that is generated from NWP data, and the entire purpose of the age-based approximation is that this data is known to have inaccuracies, so matching it exactly would not necessarily translate to better performance on real
data, but would trivially improve performance on SynthOpenContrails. Further study is needed to characterize this component of the error and whether something is needed beyond just using different ERA5 EDA members in order to make a synthetic dataset better able to model true sedimentation rates.

Figure A12 provides further visibility into how altitude factors into each algorithm's results. Figure A12(a) shows the ground-truth distribution of contrail formation altitudes in SynthOpenContrails, binned by flight-levels (flights in North America gen-
erally cruise at intervals of 1000 feet, measured as barometric altitudes, and labeled by dividing this altitude in feet by 100, so flight-level 350 means 35,000 ft). The top panel shows the overall distribution and the lower panel shows the subset that the tracking dataset is evaluated on. Each bin is overlaid with the fraction of contrails in the bin that each algorithm attributes correctly. There is no substantial difference in performance between flight-levels for any algorithm, and the differences between algorithms reflect the global contrail recall differences. We observe that essentially all of the contrails are formed above
flight-level 300, and those few that aren't are likely due to the aforementioned weather interpolation error. There is also an



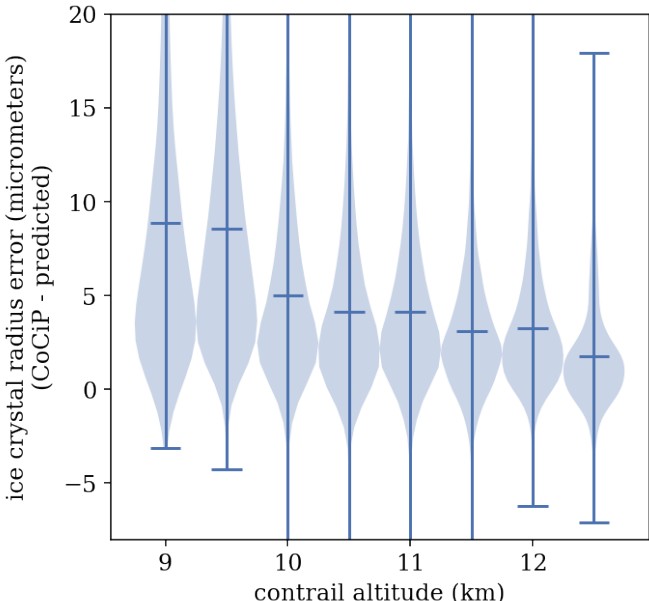

**Figure A11.** A violin plot showing the distribution of ice crystal radius error between what CoCiP predicts and the predictions from the statistical function of age mentioned in Appendix A4, binned by contrail altitude. The horizontal lines indicate the mean of the distribution.

alternating effect in bin size between "even" (even multiples of 1000 feet) and "odd" flight-levels, where the even flight-level bins are generally substantially smaller than their neighboring odd flight-level bins. Within North America, the even flight-levels are assigned to flights heading south or west, while the odd flight-levels are assigned to flights heading north or east. This may indicate different rates of producing detectable contrails based on the degree to which the flight heading is aligned with the prevailing winds, although we note that this effect is not seen in Figure A9(b). Further study is needed to explain this phenomenon, and to understand if it is also present in real data or an artifact of CoCiP.

Figure A12(b) shows the distribution of formation flight-levels for the segments of each flight that are incorrectly attributed to contrails. These again look fairly similar across algorithms. We note, however, that all three have non-trivial numbers below flight-level 300. The single-frame algorithm has the highest rate, at 10.5%, followed by the tracking algorithm, with 7.8%, and CoAtSaC with 7.1%. This demonstrates that incorporating the temporal dynamics into the attribution can reduce these seemingly implausible attributions.

Figure A12(c) looks at the altitudes at the time of contrail observation, rather than formation. Specifically, it again looks only at the attributions to incorrect flights, and subtracts the ground-truth altitude of the center of the contrail at the time of observation from the altitude of the incorrectly attributed segment of the advected flight. All of the algorithms show a fairly wide spread, indicating that adding an external signal for observed contrail altitude could help a lot, even if it were only accurate to within, say, a kilometer. The secondary peaks, especially visible in the single-frame distribution, are likely tied to the flight-level quantization of the original flight tracks. In the single-frame results we can identify the peaks corresponding to



three flight-levels in each direction, whereas the other two algorithms only clearly show one in each direction. This is, again, likely a result of incorporating temporal dynamics, since the likelihood of having the same wind speed at different flight-levels may decrease the further apart the flight-levels are. An additional observation is that the distributions are not symmetrical. The single-frame algorithm has 9.2% of it incorrect matches where the true contrail altitude is more than 2 km above the advected flight, but only 3.8% in the reverse direction. The tracking algorithm is 6.7% versus 2.1%, and CoAtSaC is 4.5% vesus 2.2%. Generally this shows that slightly fewer of CoAtSaC's errors are at substantially incorrect altitudes, which is again attributable to wind speeds being more correlated at nearby altitudes. The asymmetry is likely a result of contrails forming near the upper range of commercial flight cruising altitudes, which provides a relatively small upper bound on how far above a contrail an incorrectly attributed advected flight can be, but there is a much wider range of altitudes available for incorrect attributions lower than the contrail.

Contrail age is the other axis that seems heavily negatively correlated with attribution performance, as shown in Figure 8(d). The single-frame algorithm has a simple explanation for this, which is that it has an explicit term in its score function that makes a flight that has advected for longer less likely to match a contrail. CoAtSaC's behavior is less straightforward. We speculate that it may be tied to contrails growing wider and less linear with age, and therefore the linearization becomes less consistent. For example, if the contrail starts to curve, either the linearization will keep it as a single contrail and join the endpoints, which would likely produce very different $W$ values than when it was more linear, or it could split it into two smaller linearizations, where one would have a younger implied age and the other an older implied age than the full contrail would have had, making the $W$ by implied-age plot less linear. For the single-frame and CoAtSaC algorithms the performance artificially goes to 0 at 2 hours because flights are only advected for that long, so any detected contrail older than that can only be attributed to incorrect flights. The tracking algorithm allows for longer advection, so it has non-zero performance past 2 hours, but both precision and recall decline rapidly on these older contrails.

A related effect is shown in Figure A9(a), where performance is assessed based on the total CoCiP-reported lifetime of the detected contrail. The units here are still contrail detections, so detections of the same contrail will appear in the corresponding histogram bin multiple times, and presumably the longer-lived contrails appear more times. This is artificially flattened out by the age-based decay of optical depth in Equation A4, however. If we ignore the first bin, which is nearly empty, Figure A9(a) is just a stretched out version of Figure 8(d).

Contrail length does have a meaningful impact on performance, with performance improving monotonically with increased length for all metrics except for single-frame recall. This is shown in Figure 8(c). The improved performance with increased length makes sense in the multitemporal context of CoAtSaC, since longer contrails are more likely to persist in multiple satellite frames just due to the time it takes to form them end-to-end. As they evolve they are also likely to produce more stable linearizations and $W$ values over time, due to being better constrained by additional contrail mask pixels and flight waypoints, respectively. The decrease in single-frame recall for longer contrails may be tied to longer contrails generally being less linear, since the wind fields are not uniform over larger spatial regions. Even with a perfectly linear flight path, the advection will make the contrail non-linear, but, up to a point, the linearization process will still make it into a single linear contrail. This will





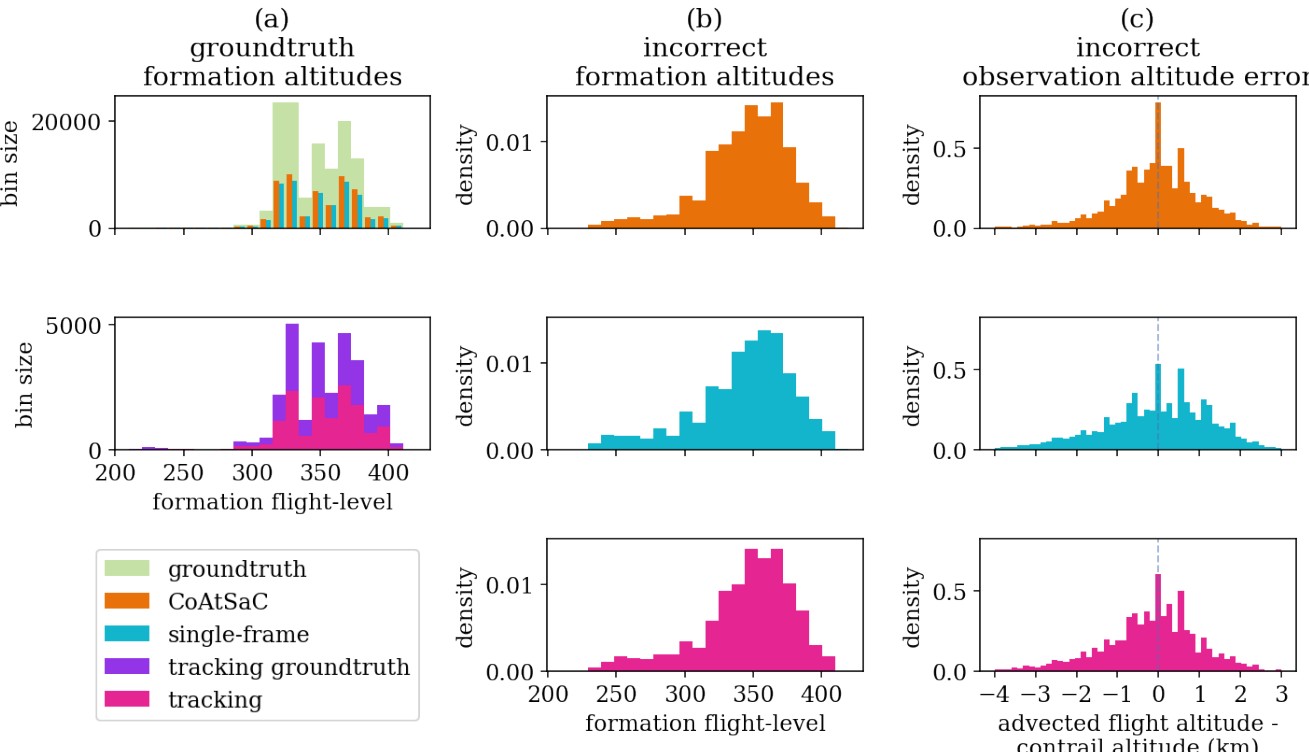

**Figure A12.** Altitude-related distributions of the attributions from all 3 algorithms. In **(a)**, the top panel shows the distribution of flight-levels at which the SynthOpenContrails contrails were formed in green, weighted by the number of frames each contrail is detected in. The orange and blue bars show the fraction of contrail detections from each bin that are correctly attributed by the CoAtSaC and single-frame algorithms, respectively. The lower panel shows flight-level distribution of the subset that the tracking algorithm was evaluated on in purple, and the fraction of each bin that the tracking algorithm attributed correctly in pink. **(b)** shows the distribution of flight levels of the flight segments incorrectly attributed to a contrail detection by each algorithm. **(c)** looks at the time of contrail observation, rather than formation, and shows the distribution of altitude error, as measured by the difference between the altitude of the incorrectly attributed advected flight and the altitude of the contrail, from each algorithm.

negatively impact the fit term of Equation 2 because the rigid transform can not make a non-linear advected flight path become linear.

We investigated the hypothesis that the relative angle between the flight heading and the wind direction impacts attribution performance. This was motivated by the fact that contrails that are advecting directly along the original flight path are difficult for humans to attribute in most existing visualization methods. Furthermore, given that the advection is almost entirely in the $v$ direction (as in the $v - w$ plane, not the conventional $u - v$ wind direction vectors), this could hurt an algorithm dependent on wind error only in the $w$ direction. As we show in Figure 8(f), none of the algorithms seem to suffer in this scenario.





Performance on some metrics is actually slightly higher when the flight is flying directly into or along with the wind, as opposed to perpendicular to it.

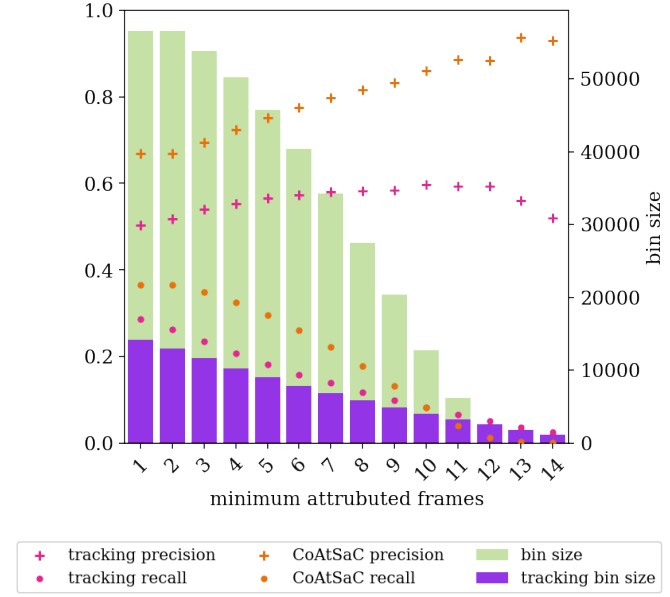

**Figure A13.** The impact on contrail-level precision and recall when only considering attributions that see a contrail in at least a minimum number of frames. The single-frame algorithm is not presented here, since it does not link attributions across frames.

Finally, we assess the impact of requiring that contrails be attributed in at least a certain number of frames in order to be considered a match. Both the CoAtSaC and tracking algorithms have a notion of linking together contrail detections in multiple frames to claim that they are observations of the same physical contrail. We hypothesize that those are attributed in more frames will be higher confidence, and therefore, dropping those attributions with fewer frames would increase precision. As shown in Figure A13, this largely holds true. CoAtSaC shows a fairly linear increase in precision as the threshold for minimum number of frames increases, approaching perfect precision at the upper end of the range, but recall decreases quite rapidly. The tracking algorithm shows more modest gains in precision, and even reduces somewhat at the high end, but its recall does not decrease quite as rapidly as it does for CoAtSaC. It appears that this could be a valuable lever for an attribution use-case that needs very high precision, at the expense of recall.

## A28 Time Spans

We document here the time spans used for all aspects of this work. All dates and times are UTC. The time spans are divided into train, validation, and test splits, presented in Table A5, Table A6, and Table A7, respectively. For each span here, there are a number of derived time spans applied for different purposes. These are documented in Table A8.





**Table A5.** Time spans in the train set

| Start time (UTC) | End time (UTC) |
| --- | --- |
| 2019-04-18 08:00 | 2019-04-19 06:00 |
| 2019-04-29 20:00 | 2019-04-30 18:00 |
| 2019-05-09 02:00 | 2019-05-10 00:00 |
| 2019-05-13 20:00 | 2019-05-14 18:00 |
| 2019-05-25 08:00 | 2019-05-26 06:00 |
| 2019-06-08 08:00 | 2019-06-09 06:00 |
| 2019-06-29 02:00 | 2019-06-30 00:00 |
| 2019-07-07 02:00 | 2019-07-08 00:00 |
| 2019-07-21 14:00 | 2019-07-22 12:00 |
| 2019-07-27 14:00 | 2019-07-28 12:00 |
| 2019-08-05 14:00 | 2019-08-06 12:00 |
| 2019-08-19 02:00 | 2019-08-20 00:00 |
| 2019-09-14 08:00 | 2019-09-15 06:00 |
| 2019-09-25 02:00 | 2019-09-26 00:00 |
| 2019-10-07 02:00 | 2019-10-08 00:00 |
| 2019-10-16 14:00 | 2019-10-17 12:00 |
| 2019-11-01 20:00 | 2019-11-02 18:00 |
| 2019-11-15 14:00 | 2019-11-16 12:00 |
| 2019-11-24 02:00 | 2019-11-25 00:00 |
| 2019-12-06 14:00 | 2019-12-07 12:00 |
| 2019-12-14 14:00 | 2019-12-15 12:00 |
| 2019-12-22 20:00 | 2019-12-23 18:00 |
| 2020-01-16 14:00 | 2020-01-17 12:00 |
| 2020-01-23 14:00 | 2020-01-24 12:00 |
| 2020-02-07 08:00 | 2020-02-08 06:00 |
| 2020-02-19 14:00 | 2020-02-20 12:00 |
| 2020-03-08 14:00 | 2020-03-09 12:00 |
| 2020-03-25 02:00 | 2020-03-26 00:00 |



**Table A6.** Time spans in the validation set

| Start time (UTC) | End time (UTC) |
|------------------|----------------|
| 2019-04-21 02:00 | 2019-04-21 06:00 |
| 2019-04-26 08:00 | 2019-04-26 12:00 |
| 2019-05-06 14:00 | 2019-05-06 18:00 |
| 2019-05-18 02:00 | 2019-05-18 06:00 |
| 2019-05-31 20:00 | 2019-06-01 00:00 |
| 2019-06-14 20:00 | 2019-06-15 00:00 |
| 2019-06-22 14:00 | 2019-06-22 18:00 |
| 2019-07-11 08:00 | 2019-07-11 12:00 |
| 2019-07-15 14:00 | 2019-07-15 18:00 |
| 2019-07-31 02:00 | 2019-07-31 06:00 |
| 2019-08-11 20:00 | 2019-08-12 00:00 |
| 2019-08-28 08:00 | 2019-08-28 12:00 |
| 2019-09-17 14:00 | 2019-09-17 18:00 |
| 2019-09-29 20:00 | 2019-09-30 00:00 |
| 2019-10-05 08:00 | 2019-10-05 12:00 |
| 2019-10-22 02:00 | 2019-10-22 06:00 |
| 2019-11-05 14:00 | 2019-11-05 18:00 |
| 2019-11-21 08:00 | 2019-11-21 12:00 |
| 2019-11-28 20:00 | 2019-11-29 00:00 |
| 2019-12-10 08:00 | 2019-12-10 12:00 |
| 2019-12-28 02:00 | 2019-12-28 06:00 |
| 2020-01-04 08:00 | 2020-01-04 12:00 |
| 2020-01-13 20:00 | 2020-01-14 00:00 |
| 2020-01-27 02:00 | 2020-01-27 06:00 |
| 2020-02-11 20:00 | 2020-02-12 00:00 |
| 2020-02-24 08:00 | 2020-02-24 12:00 |
| 2020-03-12 20:00 | 2020-03-13 00:00 |
| 2020-03-29 14:00 | 2020-03-29 18:00 |





**Table A7.** Time spans in the test set. All time spans were used in the evaluation of the single frame and CoAtSaC algorithms. Only the time spans indicated in the third column were used in the evaluation of the tracking algorithm of Chevallier et al. (2023).

| Start time (UTC) | End time (UTC) | Included in tracking algorithm evaluation |
|---|---|---|
| 2019-04-15 02:00 | 2019-04-15 06:00 | No |
| 2019-04-22 14:00 | 2019-04-22 18:00 | No |
| 2019-05-03 20:00 | 2019-05-04 00:00 | No |
| 2019-05-10 14:00 | 2019-05-10 18:00 | No |
| 2019-05-22 08:00 | 2019-05-22 12:00 | Yes |
| 2019-06-05 14:00 | 2019-06-05 18:00 | No |
| 2019-06-27 08:00 | 2019-06-27 12:00 | Yes |
| 2019-07-02 14:00 | 2019-07-02 18:00 | No |
| 2019-07-19 20:00 | 2019-07-20 00:00 | Yes |
| 2019-07-24 08:00 | 2019-07-24 12:00 | Yes |
| 2019-08-03 02:00 | 2019-08-03 06:00 | No |
| 2019-08-16 20:00 | 2019-08-17 00:00 | Yes |
| 2019-09-11 08:00 | 2019-09-11 12:00 | No |
| 2019-09-20 08:00 | 2019-09-20 12:00 | Yes |
| 2019-10-02 14:00 | 2019-10-02 18:00 | Yes |
| 2019-10-10 02:00 | 2019-10-10 06:00 | Yes |
| 2019-10-27 20:00 | 2019-10-28 00:00 | Yes |
| 2019-11-11 02:00 | 2019-11-11 06:00 | No |
| 2019-11-18 20:00 | 2019-11-19 00:00 | Yes |
| 2019-12-02 02:00 | 2019-12-02 06:00 | Yes |
| 2019-12-17 20:00 | 2019-12-18 00:00 | No |
| 2019-12-19 08:00 | 2019-12-19 12:00 | No |
| 2020-01-10 14:00 | 2020-01-10 18:00 | No |
| 2020-01-20 08:00 | 2020-01-20 12:00 | Yes |
| 2020-02-03 02:00 | 2020-02-03 06:00 | No |
| 2020-02-16 02:00 | 2020-02-16 06:00 | No |
| 2020-03-03 14:00 | 2020-03-03 18:00 | Yes |
| 2020-03-19 08:00 | 2020-03-19 12:00 | Yes |



**Table A8.** Derived time spans from those specified in Table A5, Table A6, and Table A7 for different applications. The start and end times of each span in the other tables are referenced here as $t_1$ and $t_2$, respectively.

| Application | Start time | End time |
|---|---|---|
| Attribution statistics | $t_1$ | $t_2$ |
| Attribution flight loading | $t_1 - 2$hrs | $t_2$ |
| Attribution contrail detection loading | $t_1 - 2$hrs | $t_2 + 2$hrs |
| Synthetic flight loading | $t_1 - 6$hrs | $t_2 + 3$hrs |
| Synthetic rasterization | $t_1 - 2$hrs | $t_2 + 3$hrs |

*Author contributions.* AS performed most of the algorithm design and implementation for the CoAtSaC algorithm as well as the SynthOpenContrails dataset, and also led the analysis and paper-writing. VM contributed regular feedback on the approach and wrote parts of the paper. RC adapted the tracking algorithm to work on SynthOpenContrails and ran the evaluations, and also contributed to the paper writing. AD and KM implemented the initial version of the synthetic contrails generation pipeline, upon which the SynthOpenContrails was built. SG helped design and implement the CoAtSaC algorithm and provided regular feedback on all aspects of the project. KM helped design and implement the SynthOpenContrails dataset, and also provided regular feedback on all aspects of the project.

*Competing interests.* The authors declare the following financial interests/ personal relationships which may be considered as potential competing interests: Some authors are employees of Google Inc. as noted in their author affiliations. Google is a technology company that sells computing services as part of its business.

*Acknowledgements.* We thank Tristan Abbott, Zebediah Engberg, and Marc Shapiro at Breakthrough Energy for their assistance with building and using the PyContrails library, as well as providing feedback on the overall approach and details along the way. We also thank Tharun Sankar for his help in adapting PyContrails for our use, Sebastian Eastham for his guidance throughout the project, and Dilip Krishnan for the initial suggestion of applying RANSAC to this problem.



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
