# Peer review of "Benchmarking and improving algorithms for attributing satellite-observed contrails to flights"

_EGUsphere, 2024_

## Referee Comment (RC1)

**Summary**

In the work "Benchmarking and improving algorithms for attributing satellite-observed contrails to flights", the authors demonstrate an improvement in attributing contrail detections to segments of generating flights—by utilising the way that the difference between a contrail's location and the position of an advected flight track compounds as the advection time is increased. As well as enabling attribution of contrails in groups (which minimises detections available for 'false positive' attribution), the offset is used to attribute a group of detections more-confidently to the flight than using 'single-frame' attribution methods, using the fact that the offset (interpretable for a match as 'advection error') tends to zero at the time of the flight.

This algorithm is benchmarked against other attribution algorithms (and tuned) using "synthetic contrail observations"—specifically, linear geometries based on a rasterised optical thickness field using output from CoCiP (Schumann, 2012, A 'Contrail Cirrus Prediction' model).

**General comments**

This study signficantly furthers attempts at contrail attribution, developing a benchmarking framework, a new, performant model, and an interesting exploration of its physical basis. A comprehensive dataset of attributed contrails is needed to benchmark attribution algorithms, but remains unachievable in the immediate term. This work highlights this issue effectively and constructs an alternative using model outputs, and appears to be the first comprehensive dataset effective in highlighting the limitations of current attribution algorithms, and distinguishing between them.

These methods and associated conclusions are clearly described in the text, including the underlying assumptions, and are well-captured by the abstract and title. The results support the conclusions made.

The manuscript is long, and is frequently repetitive, despite the clarity and importance of the results. In particular, many appendices are included; these are overused and unstructured. Some appendices contain analysis beyond the manuscript or important to include in the body to support discussions. Nonetheless, these are very valuable results, with a methodology which seems largely sound. Although a particularly long list of comments has been included below, most are minor suggestions, and I look forward to seeing the final version. It is recommended for publication following minor revisions.

**Specific comments**

The larger points of concern are presented here, and are followed by many more minor points and suggestions, several of which are stylistic and are left to the authors' discretion.

1. Many appendices have been presented in an unstructured way (i.e. all part of Appendix A; not listed the same order they are referenced), and often go beyond the analysis or discussion included in the paper, these further analyses should be included in the manuscript body or removed. A number of suggestions have been included in the list below. They should be appropriately structured into sections and subsections. A supplement should be considered for anything that cannot reasonably be included (though a supplement should also be supporting, and not significant analysis and discussion omitted from the body).

2. Lines 268–273 (and Fig. 4, 5(a,b)): Clarify why one needs to group contrails attributed across a flight, rather than starting with Fig. 5(b), wherein only overlapping waypoints are considered. Given the single-frame attributions are the result of advecting flight waypoints, ungrouping overlapping waypoints doesn't represent an important step for the new algorithm. For Fig. 4 it would be more valuable to present ambiguous cases from overlapping waypoints.

3. Line 381–383: The ERA5 ensemble control run is not independent of the ERA5 operational analysis data. Using accurate weather data (i.e. output from the same model) would mean that the metrics were not benchmarking the differences in weather models. Although it is true that this would lead to overestimates of precision, it is already acknowledged that the benchmark is only appropriate for relative comparisons.
   The current approach also means that a future algorithm could falsely perform well only by using the weather data used to produce the benchmarking dataset.
   The applicability of the data is somewhat discussed in Appendix A7—but the errors between models do not occur randomly, so only matching the distribution doesn't mean that the advection error is realistically distributed.
   A better alternative to this approach could be to use the same ERA5 data used for the attribution algorithm, but perturb it by a field representing some estimated error.

4. Lines 478–471: Specify why it is valuable for 'per-flight' recall and precision to be evaluated without requiring attributions to be correct. Will this falsely reward inaccurate attributions in flight-dense regions? In the prose definitions, the precision and recall are hard to differentiate—Appendix A19 is clearer and should be moved to the body of the manuscript.
   Table 1: The per-frame and global quantities struggle to distinguish between relative performance between different algorithms, other than the single-frame algorithm's flight recall. Given the expectation that global quantities are biased towards dense scenes, and the flight recall does not impose accurate attributions, could it be the case that this algorithm is only being rewarded for making false attributions in dense scenes? (This seems to be indicated by Appendix A26's decreased precision seen when including single-frame attributions for ambiguous cases, and Fig. A8's ever-increasing flight recall for the single-frame attribution). The analysis of Appendix A26 should be in the body of the manuscript.

Given the otherwise-similar relative performance of the algorithms, can a single approach (per-frame or global) be presented as the best way to benchmark?

5. Section 3.1: Specify whether all the benchmarked algorithms were tuned on the SynthOpenContrails, and if this would have an impact on their relative performance. Some of the context of Appendix A25 is important for the body of the manuscript, including the fraction of time spans that were used for the tracking algorithm, and whether this has any impact on the results (i.e. how do the other algorithms perform on this subset).

6. Table 1: The standard deviations in the per-frame quantities are significant, as expected based on the variability in Fig. A6. Would this variability be even higher if stratified by 'similar scenes' (Fig. A6 shows similar statistics in consecutive frames with occasional jumps). Perhaps a per-time-span average would better capture variability? Can estimates of uncertainty in precision and recall be made? Are more time spans needed to accurately benchmark?

7. Line 292: Discuss why the slope is an indicator that a fit contains repeated detections of the same contrail—that a real contrail associated with a specific flight would have less error in advection and therefore a shallower slope? Clarify whether a steep negative slope is considered a 'lower Ssc'. Clarify whether Cslope has any impact for Ssc (is it constant across candidate fits?), if its impact is only in Sfit, it could be removed from equation 3 and only included in equation 4. The preference for maximum number of detections above a high Ssc could be clarified by specifying that Ssc is high unless Nslope=max(Nslope). Altogether, could lines 286–297 could be simplified as 'selecting the candidate fit with the shallowest gradient with the largest number of inliers'?

Minor comments and suggestions follow. Many of them are stylistic, which might help to improve clarity. More substantial points are in bold

**Abstract**

The abstract clearly describes the aims and motivations of the work. It could be useful for the order of the abstract (i.e. benchmark and new algorithm) to be the same as the text.

Line 8: 'synthetic contrail observations'—rephrasing as 'synthetic contrail detections' could make clearer that these are geometry definitions rather than imitating satellite data.

**Line 9–10:** emphasise that the metrics are appropriate only for relative performance, and are not appropriate for making inference about these 'real world' applications (per Section 2.4.3).

**Introduction**

Lines 22–58: This part of the introduction motivates contrail attribution using the need to relate contrail observations to the forming aircraft, to be able to relate contrail properties to aircraft properties, for "validating contrail forecasts" (line 49), validate avoidance results, and for contrail altitude estimation.
A shorter version of this part would be sufficient to motivate contrail attribution, and could be structured to highlight the value of attribution more specifically. Further methodological comments are below, although abbreviating this section could negate the need to treat these in the text while keeping the study sufficiently motivated.

**Line 25**: The frequency and morphology of ISSRs indicates that trajectory modifications are minor, and could lead one to conclude that few flights cause persistent contrails, but it's not clear from this that the contrail impact that does occur could be largely mitigated from deviating only a small subset of contrail-forming flights in practice—this is a modelling result that may differ by region (e.g. Teoh et al., 2020).

**Lines 29–35**: Here, the modelling results establishing the physical basis for trajectory modification (which assume that meteorological data allows for contrail forecasts to be correct in terms of the statistics of contrail occurrence), are conflated with the application of the data for trajectory avoidance in practice (which may require a "perfect" forecast for success in individual cases).

Lines 63–76: The differences between these algorithms could be highlighted and their deficiencies relative to this study, rather than briefly summarising the full methodology which is similar in each case. In particular differences between this study and the Gryspeerdt et al. (2024) approach are not made clear.

**Lines 62, 70, 75:** NWP weather data should be disambiguated from reanalysis weather data (such as ERA5).

Lines 88–112: This discussion of contrail avoidance monitoring and validation is an effective and simpler motivating application than supporting model validation. This part of the introduction might be better placed earlier in the introduction, with the other motivating aims.

**Line 114:** Does the new algorithm improve on the scalability of Geraedts et al. (2024)?

**Lines 114–115:** It doesn't follow that scalability and lack of relative benchmarking limit the application of attribution, it seems likely that the performance of attribution (and detection) methods are a more significant factor. If anything, existence of applications indicates that if an algorithm worked well, it would see use.

Lines 115–121: This outline at the end of the introduction could be expanded by including 'signposting' reference to specific sections, rather than the current approach of outlines at the top of each section, which makes the manuscript feel repetitive.

**Methods**

This section could benefit from restructuring. In particular, work for which a suitable citation exists is described in more detail than necessary in section 2.2.2. It may also benefit from separating into separate sections: Section 2 being the Contrail-to-Flight Attribution algorithm containing Section 2.2, and any necessary information from 2.4.3 (i.e. that it can be tuned), then a separate section 3 to introduce the synthetic contrail detections. This would enable e.g. the splitting of Section 2.2.3 into simpler subsections.

Lines 123–125, 134–135: These section introductions outside of subsequent subsections aren't necessary, and lead to repetition. Signposting at the end of the introduction is the most appropriate place for this context.

Section 2.1: This section is a little confusing when separated from the context where the notation is used, and the notation is not so complex that it is required—it could be removed, with symbols introduced at the time they are used, or including one of the appendices' symbols tables instead.

**Contrail to flight attribution algorithm**

Sections 2.2.1 and 2.2.2: These sections largely follow Geraedts et al. (2024). It could be worth highlighting this even more strongly at the beginning of Section 2.2.1. Section 2.2.2 describes the algorithm in considerable detail which might not be needed to understand or reproduce the improvements shown here. As long as the $W$ parameter were sufficiently introduced as the cross-track offset between the advected flight track and detected contrail, relying on the Introduction's description of this algorithm may even be sufficient, particularly if adapted to include Fig. 2. The discussions of lines 177–181 and 209–215 are valuable in motivating this study and should not be removed in simplifying this section—if the authors decide not to significantly simplify here, these could at least be brought to be more prominent.

**Line 139:** This is the best-achieved spatial resolution of GOES-R's ABI and occurs at nadir—clarify that this won't be achieved over the region studied.

**Line 143:** 'several time spans'—specify how many.

**Line 164:** Refine 'separated by roughly 10 hPa'—this is not true of ERA5 L137 model levels.

Line 190–191: The use of $u$ and $w$ as coordinate names is slightly confusing as they are conventionally associated with wind speeds, but their use is understandable for consistency with Geraedts et al. (2024). It might be worth particularly specifying them as being spatial.

Line 200: clarify $W$ and $V$ are distances, not translations and that $\theta$ is an angle.

Line 207: As discerning them is a key strength of CoAtSaC, it would be valuable to show how often the previous algorithm was forced to rely on the 'additional logic' to disambiguate detections, or discuss this in the context of the per-contrail recall.

Section 2.2.3: As mentioned above, this section (which introduces the new insights of CoAtSaC) could benefit from restructuring to feature near the start of its own main body section (potentially enabling contrail grouping to be separated from the full algorithm). The section itself is a little repetitive, for example, the 'fitting' methodology is described in brief (ln. 218–220), in figure captions (Figs 3,5), in a description of the figure in the text (ln 225–245), and in the implementation (ln 268–312). While the visualisations are effective and valuable, as long as they have clear captions, one clear in-text explaination should be sufficient.

Fig. 3: Panel (a) may be clearer if the individual frames were plotted on separate smaller axes, especially in clarifying cases such as Contrail D, whose associate flight track is a full-frame over-advected, but similar colours make this hard to distinguish.

Line 233: It might be worth noting whether the corresponding waypoints of flight 2 occurred before or after contrail 1 was first detected.

Lines 247–250: 'we acknowledge that we do not know the ages of the advected contrails'—this is confusing following the description of Fig. 3 because implied contrail age is already clear to be the time information necessary, as each analysis is working with a specific hypothesis flight whose time information is known, as is the time of the detection. Clarify this introduction of 'implied contrail age'.

Fig. 4: May benefit from a different palette from Fig. 3. (a) Clarify whether 1 or 2. (c) Clarify whether additional causes could be evolution of the particular contrail, advection error in the $v$ direction from a later contrail, and whether this is a match in CoAtSaC. (d) Clarify whether Sattr threshold is actually ignored in this work/whether speaking of 'available to match to other flights' under different algorithms. Clarify that speaking of 'contrail detections' rather than contrails, particularly (g)–(i).

Fig. 5: (e) clarify whether the offset parameter should ever be decreasing when the contrail ages (as is the case for the maroon fit), and if not, whether such contrails could be removed—clarify the benefit of instead using a 'fit score'. The inclusion of this fit (and reliance on a conflicting attribution) seems to be at odds with the claim of lines 242–244, that the near-zero $W$-intercept is fundamental to the algorithm.

**Line 257:** Clarify if this slope condition is equivalent to ensuring that no two simultaneous, spatially-separated contrail detections can be attributed to the same generating flight?

Line 280: More meaningful notation than $m$ and $b$ could be used. Clarify if $m$ and $dW/dt$ are the same quantity.

Line 305: clarify that 'each flight' is 'each group of flight waypoints with overlapping single-frame attributed contrails'.

Fig. 6: Does Fig. 3 not highlight a similarly rejectable scenario for contrail 1 as matched to flight 2? If so, this figure may not be necessary.

**Line 307:** What is the impact of including Ssc to this score? Although enhancing confidence in this being a single contrail, shallow slopes reduce the confidence in the 'low $W$-intercept' condition.

**Line 340:** If scalability is critical, it would be good to clarify if this approach is any more scalable than the algorithms mentioned in the introduction, specifically Geraedts et al. (2024).

Lines 344–345: Clarify if any other attribution attempts propose such a methodology.

**Synthetic contrail benchmark dataset**

Line 347: The value of the benchmark dataset goes beyond tuning the hyperparameters—the segue could be removed.

**Line 364:** The name 'SynthOpenContrails' is chosen to parallel the 'OpenContrails' dataset of Ng et al. (2023). Make explicit that the new dataset is not suitable for benchmarking contrail detection algorithms, only attribution algorithms, and that a performant contrail detection is assumed.

Line 374–375: Clarify 'flight loading purposes'.

Line 380: Clarify 'weather data to use' for what.

**Tuning and Benchmarking**

Lines: 440–442: This introduction is not necessary.

**Line 455:** Could whether or not it is the case that multiple sets of parameters match the real data be specified with more confidence? Is there any motivation to the specific set chosen?

Line 459: The content of Appendix A21 would be valuable in the body of the manuscript. Comment on whether the train split was used for manual tuning, and why a better match could not be obtained in terms of contrail pixels and linear contrails.

**Line 453:** Specify what it means for the metrics to be computed globally, the region of study is the Contiguous US.

Line 474: Fig. A6 might be valuable to include in the manuscript body. Suggest that this is then used to justify calculating per-frame quantities when results are introduced rather than here, to avoid 'teasing' the results.

Lines 481–483: Clarify this. It might be better placed as part of an applicability discussion after the results are presented.

Lines 485–489: This is a repeated introduction in abstract, and isn't necessary.

Line 498: The results of the sensitivity could be very briefly summarised here. Given the relatively straightforward outcome, the appendix could be removed. If the contents is kept, specify which precision is meant for the result of the single-frame algorithm (line 941)

**Results**

**Section 3.2:** Although performance relative to each other is approximately constant, the relative performance between bins of different contrail properties is interesting and could be expanded by including some of Appendix A26—which includes significant analysis and discussion beyond the contents of the manuscript's body. The discussions of contrail density and altitude are especially important. Given the error in meteorological data is hypotehsised as having significant impact, could this be resolved? The discussion of contrail age and width is also valuable—does this speak to limitations of using linear objects as contrails?

**Technical corrections**

In Algorithm 1, using e.g. italicised text rather than mathematics text may improve legibility.

Line 431: 'subrouting' should read 'subroutine'.

**References**

Geraedts, Scott et al. (2024). "A Scalable System to Measure Contrail Formation on a Per-Flight Basis". In: *Environmental Research Communications* 6.1, p. 015008. ISSN: 2515-7620. DOI: 10.1088/2515-7620/ad11ab.

Gryspeerdt, Edward et al. (2024). "Operational Differences Lead to Longer Lifetimes of Satellite Detectable Contrails from More Fuel Efficient Aircraft". In: *Environmental Research Letters* 19.8, p. 084059. ISSN: 1748-9326. DOI: 10.1088/1748-9326/ad5b78.

Ng, Joe Yue-Hei et al. (2023). *OpenContrails: Benchmarking Contrail Detection on GOES-16 ABI.* http://arxiv.org/abs/2304.02122. arXiv: 2304.02122 [cs].

Schumann, U. (2012). "A Contrail Cirrus Prediction Model". In: *Geoscientific Model Development* 5.3, pp. 543–580. ISSN: 1991-9603. DOI: 10.5194/gmd-5-543-2012.

Teoh, Roger et al. (2020). "Mitigating the Climate Forcing of Aircraft Contrails by Small-Scale Diversions and Technology Adoption". In: *Environmental Science & Technology* 54.5, pp. 2941–2950. ISSN: 0013-936X. DOI: 10.1021/acs.est.9b05608.

---

## Referee Comment (RC2)

Overall evaluation:

In this manuscript, the authors present a novel algorithm capable of producing synthetic contrail observations that can be used as benchmark for evaluating contrail attribution algorithms, as well as an enhanced contrail-to-flight attribution algorithm which is highly-scalable and shows significant improvement over previous ones when evaluated using the aforementioned benchmark. The topic is interesting and of good novelty. Meanwhile, the algorithms that are developed are of good scientific and technical value. The only major shortcoming of this manuscript as a journal publication is the structuring of contents, with so many appendices separating some critical information from the main contents that add to the readers' difficulty in understanding the already novel algorithms. Therefore, I would recommend a minor revision from the authors to make this manuscript better reader-friendly to the general peers in and outside the field before it gets published.

General comments:

1. The synthetic contrail dataset generated as SynthOpenContrails, as mentioned by the authors, can be used for benchmarking contrail attribution algorithms. This is a more general concept comparing to an enhanced attribution algorithm, as the latter should also be evaluated by the former to demonstrate its superiority. Therefore, I would suggest the authors put the relevant contents describing SynthOpenContrails in front of those on CoAtSaC. This would also correspond to the order of the two algorithms' appearances in the current title and abstract, which is more logically reasonable.

2. There are too many appendices in the manuscript, making the article look like more of a script of program codes with multiple subroutines responsible for different functions, rather than a journal article in the field of geoscience. While it is somewhat common to have appendices attached as 'extensions' from  specific contents in the main text, they should not include critical information so that it would not affect the understanding of paper even when removed from manuscript. For the current manuscript, however, some appendices can definitely be merged into the main text such as A1~A3, and A5, while some are beyond the analysis (A22 and part of A27) that should be removed and potentially restructured as a new paper if the authors would like to. Therefore, it is strongly recommended that the authors consider restructure the appendices and merge those vital information into the manuscript as subsections.

Specific comments:

1. For subsections 2.2.1 to 2.2.3, the contents within different subsections are kind of mixed up and duplicate, not fully corresponding to the titles of each subsection. Take 2.2.1 as an example, the details of training and validating procedure, such as data splitting, shouldn't appear in the 'Data' subsection, but in the subsection 2.4 or an independent subsection. Also discussions on the error sources of the flight advection simulation algorithm is not related to 'data' but an independent subsection.

2. I would recommended that the authors merge Appendix A5 into subsection 2.2.3 and incorporate it into the main contents, as both parts are associated with enhancements over the single-frame attribution algorithm by Geraedts et al. (2024).

3. Fig. 5(b), could you provide more details on how did you separate the mixed attributions into different groups? From the figure there are a lot of points around $W=0$ line, which are difficult to separate from others from Fig. 5(a) but are ultimately divided into different groups.

4. Equation (4), please clarify what impact does $S_{SC}$, or the slope have in the fit score? Do you expect the slope as low as possible to gain more confidence in the fitting? What's the relative impact of slope compared to the intercept, like a high $S_{SC}$ with a low $|b|$ against a low $S_{SC}$ with a high $|b|$.

5. Line 338, is the threshold for $S_{fit}$ constant for the algorithm, or is to be customized when scaled to different parts of the world and global usage?

6. This is only a suggestion. For Algorithm 1, it's more common in my experience to use a flowchart rather than a pseudocode describing the work and logic flow. The current display resembles a technical report or a User's Guide, but not as reader-friendly as a journal article in the field of geophysics, especially to readers from different backgrounds.

7. Subsection 3.1.1, I would suggest the authors merge Appendix A27 into this subsection in the main content. As the comparison of validation results among the algorithms is pretty obvious, the contents can be summarized and shortened with the main properties and relevant comparisons.

8. For subsection 3.2, it's hard to illustrate if there is improvement given that the 'truth' labels are also generated from the algorithm, as is mentioned by the authors, rather than stand-alone observations. Also, this is not really an analysis or result of the SynthOpenContrails, but some application prospects. Therefore, I would suggest this part of information to be relocated to the discussion part in the section of conclusion rather than an independent subsection. If the authors insist on keeping it an independent subsection, some further elaboration and analysis should be given on the model trained by Sonabend et al. as well as the difference in forecasts from using different labels for training.

---

## Author Comment (AC2)

AC: We thank the reviewer for their time and thorough comments, which we think have lead to a significant improvement of the manuscript. We have revised the manuscript following both reviewers' suggestions, which were generally in agreement with each other. Below we respond to the individual reviewer comments and suggestions (RC) with either author comments (AC), manuscript changes (MC), or both, as appropriate.

Overall evaluation:

In this manuscript, the authors present a novel algorithm capable of producing synthetic contrail observations that can be used as benchmark for evaluating contrail attribution algorithms, as well as an enhanced contrail-to-flight attribution algorithm which is highly-scalable and shows significant improvement over previous ones when evaluated using the aforementioned benchmark. The topic is interesting and of good novelty. Meanwhile, the algorithms that are developed are of good scientific and technical value. The only major shortcoming of this manuscript as a journal publication is the structuring of contents, with so many appendices separating some critical information from the main contents that add to the readers' difficulty in understanding the already novel algorithms. Therefore, I would recommend a minor revision from the authors to make this manuscript better reader-friendly to the general peers in and outside the field before it gets published.

General comments:

RC1. The synthetic contrail dataset generated as SynthOpenContrails, as mentioned by the authors, can be used for benchmarking contrail attribution algorithms. This is a more general concept comparing to an enhanced attribution algorithm, as the latter should also be evaluated by the former to demonstrate its superiority. Therefore, I would suggest the authors put the relevant contents describing SynthOpenContrails in front of those on CoAtSaC. This would also correspond to the order of the two algorithms' appearances in the current title and abstract, which is more logically reasonable.

AC1. Thank you, we have adopted this suggestion and we believe that this has substantially improved the flow of the paper.

MC1. Swapped the order of the synthetic dataset section and the attribution algorithm section, and then divided up the subsections of the "Tuning and Benchmarking" section into the other sections, as appropriate.

RC2. There are too many appendices in the manuscript, making the article look like more of a script of program codes with multiple subroutines responsible for different functions, rather than a journal article in the field of geoscience. While it is somewhat common to have appendices attached as 'extensions' from specific contents in the main text, they should not include critical information so that it would not affect the understanding of paper even when removed from manuscript. For the current manuscript, however, some appendices can definitely be merged into the main text such as A1~A3, and A5, while some are beyond the analysis (A22 and part of

A27) that should be removed and potentially restructured as a new paper if the authors would like to. Therefore, it is strongly recommended that the authors consider restructure the appendices and merge those vital information into the manuscript as subsections.

AC2. We have integrated appendices A1, A5, A19, A20, A21, A24, A26, and part of A27 directly into the manuscript body. We feel that A2 and A3 should remain in the appendix, as including them in the manuscript body might cause the reader to have a hard time understanding what we did because there is too much interspersed discussion of why we did it.

We have dropped appendix A22 entirely, as we agree that it is perhaps better suited to a followup paper. We replaced appendix A23 with a short summary of it in the main text.

The remaining appendices have been reorganized into logical sections.

MC2. We integrated appendices A1, A5, A19, A20, A21, A24, A26, and part of A27 directly into the manuscript body. We have removed Appendix A22 and its references. We replaced appendix A23 with a short summary of it in the main text. Reorganized remaining appendices into logical sections.

Specific comments:

RC3. For subsections 2.2.1 to 2.2.3, the contents within different subsections are kind of mixed up and duplicate, not fully corresponding to the titles of each subsection. Take 2.2.1 as an example, the details of training and validating procedure, such as data splitting, shouldn't appear in the 'Data' subsection, but in the subsection 2.4 or an independent subsection. Also discussions on the error sources of the flight advection simulation algorithm is not related to 'data' but an independent subsection.

MC3. Restructured much of this section. Flight advection is now its own subsection, outside of Data. The dataset splits are now discussed in the "Tuning the Synthetic Dataset Parameters" section.

RC4. I would recommended that the authors merge Appendix A5 into subsection 2.2.3 and incorporate it into the main contents, as both parts are associated with enhancements over the single-frame attribution algorithm by Geraedts et al. (2024).

MC4. Moved Appendix A5 into the main text.

RC5. Fig. 5(b), could you provide more details on how did you separate the mixed attributions into different groups? From the figure there are a lot of points around W=0 line, which are difficult to separate from others from Fig. 5(a) but are ultimately divided into different groups.

AC5. Thank you for pointing out that this is confusing. Each attribution is associated with a range of advected waypoints that ostensibly formed the detected contrail. Flights can of course

form contrails multiple times along the flight path. If we imagine a scenario where a flight formed 2 contrails an hour apart from each other, and there was no advection error, the contrail age x W plot would put all attributions along the W=0 axis, and it would be impossible to separate them. What we therefore do is divide up the attributions into groups such that there are no common waypoints attributed to multiple groups. In the example given, this would now give us 2 groups, each with some subset of the attributions along the W=0 axis, and we can then trivially produce the correct fits. As this point was confusing to both reviewers, and it is essentially an implementation detail, we have removed the discussion of the separation into groups, and instead just stated that we start from sets of single-frame attributions with overlapping waypoints. We also then removed Fig 5(a) and the other groups from Fig 5(b).

MC5. Simplified the process of producing the groups of attributions for running the Fitting stage to just say "gathering all single-frame contrail attributions that are attributed to overlapping sets of waypoints for the same flight." Removed Fig 5(a) and the subplots from Fig 5(b) that showed other groups. Updated the caption for Fig 5.

RC6. Equation (4), please clarify what impact does SSC, or the slope have in the fit score? Do you expect the slope as low as possible to gain more confidence in the fitting? What's the relative impact of slope compared to the intercept, like a high SSCwith a low |b| against a low SSC with a high |b|.

AC6. This was a term that the tuning algorithm had the option to set to 0, but did not. Given that it's black-box tuning it's hard to know exactly why it did that. We can speculate that it might come into play in scenes with a large number of short-lived contrails (maybe similar to Fig 4(g)), where the supremacy of the number of inliers term during the fit generation phase might still produce a fit with a relatively large slope that joins detections of many different physical contrails. This term allows the S_fit for such fits to be high, and then ideally be rejected in the "Rejecting" phase.

MC6. Added this speculation to the text.

RC7. Line 338, is the threshold for Sfit constant for the algorithm, or is to be customized when scaled to different parts of the world and global usage?

AC7. The threshold value of 3 is not tuned and was chosen for consistency with (Geraedts et al. 2024) to make downstream analysis easier. The tuning and resulting performance would have been the same if we had tuned with any other threshold, but the resulting parameter values would change. Further research is required to determine whether the tunable parameter values will need to change in other regions of the world. The most likely regional difference that would cause re-tuning to be needed is if there is consistently higher flight and contrail density, which we are likely to find in parts of Europe. The performance fall-off of all 3 algorithms with increased contrail density, as shown in Fig 9(a) and elsewhere, is, to our knowledge, the first time that this density effect has been quantified, and we hope that this will motivate future research into attribution methods suitable for high density regions.

RC8. This is only a suggestion. For Algorithm 1, it's more common in my experience to use a flowchart rather than a pseudocode describing the work and logic flow. The current display resembles a technical report or a User's Guide, but not as reader-friendly as a journal article in the field of geophysics, especially to readers from different backgrounds.

AC8. We thank the reviewer for the suggestion. The algorithm pseudocode has been replaced by a flow chart. We further added a flow chart describing the CoAtSaC algorithm to help better structure that section of the manuscript.

MC8. Replaced algorithm with a flow chart. Added flow chart for the CoAtSaC algorithm.

RC9. Subsection 3.1.1, I would suggest the authors merge Appendix A27 into this subsection in the main content. As the comparison of validation results among the algorithms is pretty obvious, the contents can be summarized and shortened with the main properties and relevant comparisons.

MC9. Integrated parts of Appendix A27 into the main text.

RC10. For subsection 3.2, it's hard to illustrate if there is improvement given that the 'truth' labels are also generated from the algorithm, as is mentioned by the authors, rather than stand-alone observations. Also, this is not really an analysis or result of the SynthOpenContrails, but some application prospects. Therefore, I would suggest this part of information to be relocated to the discussion part in the section of conclusion rather than an independent subsection. If the authors insist on keeping it an independent subsection, some further elaboration and analysis should be given on the model trained by Sonabend et al. as well as the difference in forecasts from using different labels for training.

MC10. Removed this section and added a line about it in the Conclusions section.

---

## Author Comment (AC3)

AC: We thank the reviewer for their time and thorough comments, which we think have lead to a significant improvement of the manuscript. We have revised the manuscript following both reviewers' suggestions, which were generally in agreement with each other. Below we respond to the individual reviewer comments and suggestions (RC) with either author comments (AC), manuscript changes (MC), or both, as appropriate.

**Summary** In the work "Benchmarking and improving algorithms for attributing satellite-observed contrails to flights", the authors demonstrate an improvement in attributing contrail detections to segments of generating flights—by utilising the way that the difference between a contrail's location and the position of an advected flight track compounds as the advection time is increased. As well as enabling attribution of contrails in groups (which minimises detections available for 'false positive' attribution), the offset is used to attribute a group of detections more-confidently to the flight than using 'single-frame' attribution methods, using the fact that the offset (interpretable for a match as 'advection error') tends to zero at the time of the flight. This algorithm is benchmarked against other attribution algorithms (and tuned) using "synthetic contrail observations"—specifically, linear geometries based on a rasterised optical thickness field using output from CoCiP (Schumann, 2012, A 'Contrail Cirrus Prediction' model).

**General comments** This study signficantly furthers attempts at contrail attribution, developing a benchmarking framework, a new, performant model, and an interesting exploration of its physical basis. A comprehensive dataset of attributed contrails is needed to benchmark attribution algorithms, but remains unachievable in the immediate term. This work highlights this issue effectively and constructs an alternative using model outputs, and appears to be the first comprehensive dataset effective in highlighting the limitations of current attribution algorithms, and distinguishing between them. These methods and associated conclusions are clearly described in the text, including the underlying assumptions, and are well-captured by the abstract and title. The results support the conclusions made. The manuscript is long, and is frequently repetitive, despite the clarity and importance of the results. In particular, many appendices are included; these are overused and unstructured. Some appendices contain analysis beyond the manuscript or important to include in the body to support discussions. Nonetheless, these are very valuable results, with a methodology which seems largely sound. Although a particularly long list of comments has been included below, most are minor suggestions, and I look forward to seeing the final version. It is recommended for publication following minor revisions.

**Specific comments**

The larger points of concern are presented here, and are followed by many more minor points and suggestions, several of which are stylistic and are left to the authors' discretion.

RC1. Many appendices have been presented in an unstructured way (i.e. all part of Appendix A; not listed the same order they are referenced), and often go beyond the analysis or discussion included in the paper, these further analyses should be included in the manuscript body or

removed. A number of suggestions have been included in the list below. They should be appropriately structured into sections and subsections. A supplement should be considered for anything that cannot reasonably be included (though a supplement should also be supporting, and not significant analysis and discussion omitted from the body).

MC1. We have done the following to address these issues:
- Restructured appendix into 6 sections: "Synthetic Dataset Design Decisions," "Synthetic Dataset Generation," "Attribution Algorithm Design Decisions," "Modifications to Previously Published Attribution Algorithms," "Performance as a Function of Contrail Properties," and "Time Spans and Dataset Splits."
- Fully removed Appendices A22 and A23.
- Integrated Appendices A1, A5, A19, A20, A21, A24, A26, and part of A27 into the main text.

RC2. Lines 268–273 (and Fig. 4, 5(a,b)): Clarify why one needs to group contrails attributed across a flight, rather than starting with Fig. 5(b), wherein only overlapping waypoints are considered. Given the single-frame attributions are the result of advecting flight waypoints, ungrouping overlapping waypoints doesn't represent an important step for the new algorithm. For Fig. 4 it would be more valuable to present ambiguous cases from overlapping waypoints.

AC2. We appreciate you pointing out the confusion regarding the grouping process. In many ways this is just an implementation detail, as an implementation that went directly to the sets of overlapping waypoints, without first grouping by flight, would arrive at exactly the same result. In light of this, we've decided to drop this detail from the paper and from Fig 5, to avoid confusion.

The plots in Fig 4 are already just showing overlapping waypoints. We have clarified this in the caption.

MC2. Removed Fig 5(a) and the other groups in 5(b). Removed the text discussion of first grouping by flight and then grouping by overlapping waypoints. Updated the caption for Fig 4 to indicate that each plot is only showing attributions with overlapping waypoints.

RC3. Line 381–383: The ERA5 ensemble control run is not independent of the ERA5 operational analysis data. Using accurate weather data (i.e. output from the same model) would mean that the metrics were not benchmarking the differences in weather models. Although it is true that this would lead to overestimates of precision, it is already acknowledged that the benchmark is only appropriate for relative comparisons. The current approach also means that a future algorithm could falsely perform well only by using the weather data used to produce the benchmarking dataset. The applicability of the data is somewhat discussed in Appendix A7—but the errors between models do not occur randomly, so only matching the distribution doesn't mean that the advection error is realistically distributed. A better alternative to this approach could be to use the same ERA5 data used for the attribution algorithm, but perturb it by a field representing some estimated error.

AC3. We agree that it would be ideal to have all wind error characteristics (including spatio-temporal covariances in the errors) between the ERA5 operational analysis and the ERA5 ensemble control run be similar to the wind error characteristics between the ERA5 operational analysis and the real atmosphere. During the development of this work, we did consider using a perturbed wind field as suggested by the reviewer,but ultimately did not adopt it because perturbing only the wind field can lead to physically implausible weather data. For example, inconsistencies between winds (especially wind shear) and locations of ice supersaturation could negatively impact the error characteristics of the CoCiP outputs, or a perturbed wind field may result in numerical instabilities during advection. Because of this we decided perturbing only the wind field could be counterproductive to the goal of having well-matched error characteristics. One recently-researched option to address this potential for physical inconsistency/implausibility would be to generate entire new weather ensemble members using machine-learned generative diffusion methods (e.g. https://www.nature.com/articles/s41586-024-08252-9), so we have added a discussion of this to the relevant appendix section, but we leave this to future work as it is a substantial new endeavor.

It is correct that a future attribution algorithm that used the same weather data as was used to generate the synthetic dataset would receive an artificial boost in performance. We are not particularly worried about people trying to "game" this benchmark intentionally, so have added an extra line to the text to warn against doing this accidentally. We also added a recommendation that future algorithms also use the ERA5 nominal weather data to avoid metric changes conflating the weather data differences with the attribution performance, especially as an improvement solely due to weather data differences is unlikely to translate when applied to real contrail detections.

MC3. Augmented discussion of wind error characteristics in the relevant appendix; Added a line warning against using the same weather data for an attribution algorithm as was used to generate the synthetic data, and explicitly recommending always using ERA5 nominal for benchmark metric computation.

RC4. Lines 478–471: Specify why it is valuable for 'per-flight' recall and precision to be evaluated without requiring attributions to be correct. Will this falsely reward inaccurate attributions in flight-dense regions? In the prose definitions, the precision and recall are hard to differentiate—Appendix A19 is clearer and should be moved to the body of the manuscript. Table 1: The per-frame and global quantities struggle to distinguish between relative performance between different algorithms, other than the single-frame algorithm's flight recall. Given the expectation that global quantities are biased towards dense scenes, and the flight recall does not impose accurate attributions, could it be the case that this algorithm is only being rewarded for making false attributions in dense scenes? (This seems to be indicated by Appendix A26's decreased precision seen when including single-frame attributions for ambiguous cases, and Fig. A8's ever-increasing flight recall for the single-frame attribution). The analysis of Appendix A26 should be in the body of the manuscript. Given the otherwise-similar

relative performance of the algorithms, can a single approach (per-frame or global) be presented as the best way to benchmark?

AC4. For certain downstream applications, such as evaluating a contrail avoidance trial, the binary determination of whether a flight formed a contrail is likely the important piece of information. Flight recall and flight precision measure this, and they are similar to metrics used in Geraedts et al. (2024) and Sonabend et al. (2024). One could imagine a version of these metrics that operated on individual flight waypoints, and it should be possible to compute such metrics using this synthetic dataset, but our experience shows that there is a lot of noise along the boundaries of the attributed waypoint ranges, which can result in these metrics being noisier and harder to interpret.

Inaccurate attributions would only reward the flight-level metrics if the flight had indeed formed a contrail *and* the correct contrail had not also been attributed. Otherwise they would either hurt flight-level precision or have no effect. Conceivably they might hurt flight-level recall in aggregate, if the incorrect attribution resulted in failing to attribute the true forming flight for that contrail. Because of these ambiguities, we recommend monitoring the combination of the four metrics, as contrail-level precision would absolutely be hurt by these incorrect attributions, as we see in the results for the single-frame algorithm.

Regarding the question of whether the single-frame algorithm is being rewarded in flight recall only for false attributions in dense scenes, the new figure that we've added in the Results section shows that the single frame algorithm has higher flight recall across all time-spans, not just those with high contrail density, but its flight precision varies tremendously. This, coupled with the other analyses described in Appendix A26, suggest that the dominant cause of this effect is contrails that are only detected in a single frame, which account for a large fraction of the flights that form contrails, and therefore have a large impact on flight recall, but are also harder to attribute correctly, hence the losses in flight precision.

As discussed in more detail in AC6, we have removed the separate per-frame and global metrics, in favor of a single block-bootstrap approach.

We have moved the contents of Appendices A19 and A26 into the manuscript body.

MC4. Moved Appendices A19 and A26 into the main manuscript body.

RC5. Section 3.1: Specify whether all the benchmarked algorithms were tuned on the SynthOpenContrails, and if this would have an impact on their relative performance. Some of the context of Appendix A25 is important for the body of the manuscript, including the fraction of time spans that were used for the tracking algorithm, and whether this has any impact on the results (i.e. how do the other algorithms perform on this subset).

AC5. We did not re-tune the other algorithms using SynthOpenContrails. We assume that this would improve the algorithms, but only to a point.

Regarding the subset of time spans used for evaluating the tracking algorithm, we have now specified in the text that it is precisely half the time-spans. As it turns out, these time spans are not a representative sample. Consequently, we have added the results of evaluating the other 2 algorithms on just this subset, which improves their performance across all metrics. We also added a figure that shows per-time-span performance of all algorithms, and shows that the subset that the tracking algorithm was evaluated on skews slightly towards the "easier" scenes.

MC5. Specified that the other algorithms were not re-tuned using SynthOpenContrails. Specified the fraction of time spans used for evaluating the tracking algorithm and added rows to the results table for the other algorithms being evaluated on this subset. Also added a figure showing per-time-span performance of each algorithm.

RC6. Table 1: The standard deviations in the per-frame quantities are significant, as expected based on the variability in Fig. A6. Would this variability be even higher if stratified by 'similar scenes' (Fig. A6 shows similar statistics in consecutive frames with occasional jumps). Perhaps a per-time-span average would better capture variability? Can estimates of uncertainty in precision and recall be made? Are more time spans needed to accurately benchmark?

AC6. Thank you for this suggestion. As you point out, nearby frames are highly correlated, and the variance is primarily between time spans. As such, we have switched away from having the separate "global" and "per-frame" metrics, and instead compute the metric using block-bootstrap with each multi-hour time span as the block. This provides a single central estimate of each metric, along with confidence intervals for the estimate. We have also added a figure showing the value of each metric computed over just individual time spans, which gives a sense of the variance across time spans for each algorithm.

MC6. Switched the method of metric computation to block-bootstrap over time spans, replacing the separate "global" and "per-frame" methods. Update the Metrics and Results sections accordingly. Also added a new figure showing the metrics computed per time span.

RC7. Line 292: Discuss why the slope is an indicator that a fit contains repeated detections of the same contrail—that a real contrail associated with a specific flight would have less error in advection and therefore a shallower slope? Clarify whether a steep negative slope is considered a 'lower Ssc'. Clarify whether Cslope has any impact for Ssc (is it constant across candidate fits?), if its impact is only in Sfit, it could be removed from equation 3 and only included in equation 4. The preference for maximum number of detections above a high Ssc could be clarified by specifying that Ssc is high unless Nslope=max(Nslope). Altogether, could lines 286–297 could be simplified as 'selecting the candidate fit with the shallowest gradient with the largest number of inliers'?

AC7. The reviewer makes some excellent points, both that we should have indicated absolute slope in the S_sc calculation (the implementation already used absolute slope), and that C_slope is not needed for the Eq 3. The reason for C_slope's initial inclusion in Eq 3 was that

we experimented with using a linear combination that included other terms, but the tuning set all of the other terms' coefficients to 0, making the C_slope value here irrelevant. As such, we have moved C_slope to only appear in Eq 4 and corrected the text to use the absolute slope.

As for why we prefer a shallower slope, this is mostly relevant in cases like Fig 4(f), where there are many nearby short-lived contrails. A steep-sloped fit, even below the $T_{dW/dt}$ threshold, could cut across the different physical contrails and achieve many inliers. We have added this explanation to the text. As already noted in the text, this only comes into play as a tie-breaker between fits with the same number of inliers.

We have substantially simplified lines 286-297, in line with the reviewer's suggestions.

MC7. Removed Eq 3 and just used absolute slope directly where it had been referenced previously. Clarified that it is "absolute slope" and not just slope. Replaced S_sc in Eq 4 with C_slope|dW/dt|. Added physical explanation for why preferring shallow slopes is helpful. Simplified the description of the process for determining the best fit. Replaced S_sc in Fig 5 with dW/dt.

**Minor comments and suggestions follow**. Many of them are stylistic, which might help to improve clarity. More substantial points are in bold

**Abstract**
RC8. The abstract clearly describes the aims and motivations of the work. It could be useful for the order of the abstract (i.e. benchmark and new algorithm) to be the same as the text.

AC8. We have followed Reviewer 2's suggestion of reordering the sections to be first the benchmark and then the new algorithm. It therefore now aligns with the abstract.

RC9. Line 8: 'synthetic contrail observations'—rephrasing as 'synthetic contrail detections' could make clearer that these are geometry definitions rather than imitating satellite data.

MC9. Replaced 'synthetic contrail observations' with 'synthetic contrail detections'.

RC10. **Line 9–10:** emphasise that the metrics are appropriate only for relative performance, and are not appropriate for making inference about these 'real world' applications (per Section 2.4.3).

MC10. Added the phrase "although the metrics do not directly inform real-world performance."

**Introduction**
RC11. Lines 22–58: This part of the introduction motivates contrail attribution using the need to relate contrail observations to the forming aircraft, to be able to relate contrail properties to aircraft properties, for "validating contrail forecasts" (line 49), validate avoidance results, and for contrail altitude estimation. A shorter version of this part would be sufficient to motivate contrail

attribution, and could be structured to highlight the value of attribution more specifically. Further methodological comments are below, although abbreviating this section could negate the need to treat these in the text while keeping the study sufficiently motivated.

AC11. We agree with the reviewer here. In light of this comment and RC16, we have shortened this part of the introduction (lines 22 - 58) to lines (22 - 46), and placed more emphasis on "monitoring and validating contrail avoidance" as a motivation for this work.

RC12. **Line 25:** The frequency and morphology of ISSRs indicates that trajectory modifications are minor, and could lead one to conclude that few flights cause persistent contrails, but it's not clear from this that the contrail impact that does occur could be largely mitigated from deviating only a small subset of contrail-forming flights in practice—this is a modelling result that may differ by region (e.g. Teoh et al., 2020).

MC12. Removed this line.

RC13. **Lines 29–35:** Here, the modelling results establishing the physical basis for trajectory modification (which assume that meteorological data allows for contrail forecasts to be correct in terms of the statistics of contrail occurrence), are conflated with the application of the data for trajectory avoidance in practice (which may require a "perfect" forecast for success in individual cases).

MC13. Rephrased this sentence as "These studies do however make use of forecast and reanalysis data to quantify the climate impact of contrails, and do not account for inaccuracies therein (Gierens et al., 2020; Geraedts et al., 2024; Meijer, 2024). Therefore, the benefits achievable by performing contrail avoidance with existing models may differ from those quantified in such studies."

RC14. Lines 63–76: The differences between these algorithms could be highlighted and their deficiencies relative to this study, rather than briefly summarising the full methodology which is similar in each case. In particular differences between this study and the Gryspeerdt et al. (2024) approach are not made clear.

AC14. We have rewritten this section to highlight the differences between the various prior studies, and what we suggest are deficiencies addressed by our work.

MC14. Rewrote this section to be less of a literature review and more focused on what is relevant to this study.

RC15. **Lines 62, 70, 75:** NWP weather data should be disambiguated from reanalysis weather data (such as ERA5).

MC15. Updated all uses of NWP to either the more general "weather model" or to specify reanalysis, as appropriate.

RC16. Lines 88–112: This discussion of contrail avoidance monitoring and validation is an effective and simpler motivating application than supporting model validation. This part of the introduction might be better placed earlier in the introduction, with the other motivating aims.

AC16. Please see AC11.

RC17. **Line 114:** Does the new algorithm improve on the scalability of Geraedts et al. (2024)?

AC17. It does not improve the scalability. The statement in the line referenced is in comparison to manual attribution approaches, such as those used in Sonabend et al. (2024). It does, however, improve on the scalability over previous multi-frame automated attribution approaches like Chevallier et al. (2023).

Sonabend-W, A., Elkin, C., Dean, T., Dudley, J., Ali, N., Blickstein, J., Brand, E., Broshears, B., Chen, S., Engberg, Z., et al.: Feasibility test of per-flight contrail avoidance in commercial aviation, Communications Engineering, 3, 184, 2024.

RC18. **Lines 114–115:** It doesn't follow that scalability and lack of relative benchmarking limit the application of attribution, it seems likely that the performance of attribution (and detection) methods are a more significant factor. If anything, existence of applications indicates that if an algorithm worked well, it would see use.

AC18. Performance is absolutely a significant contributor. The intention of this statement was to say that the inability to measure performance has led to unwillingness to rely on automated approaches. We have reworded to clarify.

MC18. Softened the language slightly to allow for there being other factors, and clarified that the way that the lack of benchmark contributes is in the ability to assess the performance of the attribution algorithms.

RC19. Lines 115–121: This outline at the end of the introduction could be expanded by including 'signposting' reference to specific sections, rather than the current approach of outlines at the top of each section, which makes the manuscript feel repetitive.

MC19. Added signposting at the end of the introduction.

**Methods**
RC20. This section could benefit from restructuring. In particular, work for which a suitable citation exists is described in more detail than necessary in section 2.2.2. It may also benefit from separating into separate sections: Section 2 being the Contrailto-Flight Attribution algorithm containing Section 2.2, and any necessary information from 2.4.3 (i.e. that it can be

tuned), then a separate section 3 to introduce the synthetic contrail detections. This would enable e.g. the splitting of Section 2.2.3 into simpler subsections.

AC20. Thank you, this was a very helpful suggestion. See also AC23.

MC20. Promoted the Methods section subsections to be top-level sections. Added a flow-chart overview of the CoAtSAC algorithm and used that to better organize the relevant section, adding many more subsection headings throughout. Also moved the synthetic dataset tuning and metric definitions into the benchmark section, and the attribution algorithm tuning into the attribution algorithm section, so there is no longer a separate "Tuning and Benchmarking" section.

RC21. Lines 123–125, 134–135: These section introductions outside of subsequent subsections aren't necessary, and lead to repetition. Signposting at the end of the introduction is the most appropriate place for this context.

MC21. Added signposting at the end of the introduction and removed or shortened section introductions to the extent that we felt they didn't provide additional value.

RC22. Section 2.1: This section is a little confusing when separated from the context where the notation is used, and the notation is not so complex that it is required—it could be removed, with symbols introduced at the time they are used, or including one of the appendices' symbols tables instead.

MC22. Removed this section and moved the symbol tables into the main text.

**Contrail to flight attribution algorithm**
RC23. Sections 2.2.1 and 2.2.2: These sections largely follow Geraedts et al. (2024). It could be worth highlighting this even more strongly at the beginning of Section 2.2.1. Section 2.2.2 describes the algorithm in considerable detail which might not be needed to understand or reproduce the improvements shown here. As long as the W parameter were sufficiently introduced as the cross-track offset between the advected flight track and detected contrail, relying on the Introduction's description of this algorithm may even be sufficient, particularly if adapted to include Fig. 2. The discussions of lines 177–181 and 209–215 are valuable in motivating this study and should not be removed in simplifying this section—if the authors decide not to significantly simplify here, these could at least be brought to be more prominent.

AC23. We have made clearer where we are exactly following Geraedts et al. (2024), both in terms of the Data section and the section describing the algorithm.

We have also moved the discussion of simulated flight track advection into its own subsection, outside of the data section, which we hope will increase the focus on it, as the error incurred by this approach is key to the CoAtSaC algorithm.

We have somewhat abbreviated Section 2.2.2, but we are reluctant to force a reader who is not intimately familiar with Geraedts et al. (2024) to repeatedly refer to that paper in order to understand this one. In our opinion, the summary we provide is the minimum amount of detail needed in order to understand the parts of Geraedts et al that the CoAtSaC algorithm depends on.

MC23. Added clarifications about where we follow Geraedts, et al. Moved the discussion of simulated flight track advection to its own subsection. Abbreviated section 2.2.2 slightly.

RC24. **Line 139**: This is the best-achieved spatial resolution of GOES-R's ABI and occurs at nadir—clarify that this won't be achieved over the region studied.

MC24. Clarified that this is nadir spatial resolution.

RC25. **Line 143**: 'several time spans'—specify how many.

MC25. Replaced "several" with "84"

RC26. **Line 164**: Refine 'separated by roughly 10 hPa'—this is not true of ERA5 L137 model levels.

AC26. Thank you for flagging this. We investigated and discovered that the weather data that we believed to be the L137 model levels is actually the 37 pressure levels. This means that all benchmarked algorithms are now using the same weather data source. We updated the manuscript accordingly.

MC26. Changed the model

RC27. Line 190–191: The use of u and w as coordinate names is slightly confusing as they are conventionally associated with wind speeds, but their use is understandable for consistency with Geraedts et al. (2024). It might be worth particularly specifying them as being spatial.

MC27. Clarified that it is a "2D spatial coordinate system" and added the following warning: "(we adopt the axis names from Geraedts et al. (2024) but caution not to confuse them with the conventional usage of these variables for directional wind speeds)"

RC28. Line 200: clarify W and V are distances, not translations and that θ is an angle.

MC28. Changed to "translation distances" and "rotation angles".

RC29. Line 207: As discerning them is a key strength of CoAtSaC, it would be valuable to show how often the previous algorithm was forced to rely on the 'additional logic' to disambiguate detections, or discuss this in the context of the per-contrail recall.

AC29. On the SynthOpenContrails test set, the Geraedts et al. (2024) algorithm eliminates 18% of attributions that would otherwise have been below the threshold using the 'additional logic' (which is simply eliminating any attributions that have an $S\_attr$ score more than 1 higher than the lowest score for a given contrail detection). Eliminating this logic results in the single frame algorithm achieving a (global) contrail-precision 34.9%, contrail-recall 33.8%, flight-precision 38.6%, and flight-recall 63.3%, so a slight increase in recall at the expense of a substantial drop in precision.

RC30. Section 2.2.3: As mentioned above, this section (which introduces the new insights of CoAtSaC) could benefit from restructuring to feature near the start of its own main body section (potentially enabling contrail grouping to be separated from the full algorithm). The section itself is a little repetitive, for example, the 'fitting' methodology is described in brief (ln. 218–220), in figure captions (Figs 3,5), in a description of the figure in the text (ln 225–245), and in the implementation (ln 268–312). While the visualisations are effective and valuable, as long as they have clear captions, one clear in-text explaination should be sufficient.

MC30. We have restructured much of this section around a new flow chart. We've added additional subsection headings that reference the flow chart. We've attempted to remove as much of the repetition as possible, while still including illustrative figures that help provide intuition for what each component of the algorithm is contributing.

RC31. Fig. 3: Panel (a) may be clearer if the individual frames were plotted on separate smaller axes, especially in clarifying cases such as Contrail D, whose associate flight track is a full-frame over-advected, but similar colours make this hard to distinguish.

AC31. Thank you for pointing this out. We have updated the figure to use a categorical color map so that adjacent colors are more distinct. It seems to us that this resolves the issue without plotting each frame independently, which we think runs the risk of making the temporal progression harder to understand.

MC31. Changed to a categorical color map.

RC32. Line 233: It might be worth noting whether the corresponding waypoints of flight 2 occurred before or after contrail 1 was first detected.

MC32. Added this discussion, pointing out how the evidence from Flight 1 can be used to reject attributing F, G and K to Flight 2.

RC33. Lines 247–250: 'we acknowledge that we do not know the ages of the advected contrails'—this is confusing following the description of Fig. 3 because implied contrail age is already clear to be the time information necessary, as each analysis is working with a specific hypothesis flight whose time information is known, as is the time of the detection. Clarify this introduction of 'implied contrail age'.

MC33. Rewrote these lines to make them clearer and specifically reference the plots in Fig 3.

RC34. Fig. 4: May benefit from a different palette from Fig. 3. (a) Clarify whether 1 or 2. (c) Clarify whether additional causes could be evolution of the particular contrail, advection error in the v direction from a later contrail, and whether this is a match in CoAtSaC. (d) Clarify whether Sattr threshold is actually ignored in this work/whether speaking of 'available to match to other flights' under different algorithms. Clarify that speaking of 'contrail detections' rather than contrails, particularly (g)–(i).

AC34. We have addressed most points. We changed the color palette for Fig3 (and dropped Fig 6) so they no longer make this one confusing. For (c) we have clarified that this contrail was in fact caused by a later flight that passed in the advection path of this flight. CoAtSAC does correctly (we believe) attribute the contrail to the later flight, but in our opinion that information does not belong in the caption at this stage of the paper, where the CoAtSAC algorithm hasn't yet been described.

MC34. Clarified that Fig 4(a) is 2 contrails. Clarified that (c) is a case of another flight flying near the advection path, but that such cases can also be caused by occlusion or low wind-shear conditions causing the contrail to remain too narrow to be detected until later in life. Clarified that (d) means out of match range for the single-frame algorithm. Reworded that (g)-(i) are "higher contrail detection density."

RC35. Fig. 5: (e) clarify whether the offset parameter should ever be decreasing when the contrail ages (as is the case for the maroon fit), and if not, whether such contrails could be removed—clarify the benefit of instead using a 'fit score'. The inclusion of this fit (and reliance on a conflicting attribution) seems to be at odds with the claim of lines 242–244, that the near-zero W-intercept is fundamental to the algorithm.

AC35. There are cases where the intercept is substantially non-zero, which is usually a result of linearization of a curved contrail, where the W value appears to decrease. The fact that the $S_{fit}$ value for this fit ends up very close to the threshold is a reflection of the automated tuning process determining that there are a few cases where such fits are correct and it is desirable not to categorically eliminate them at this stage, since the majority of cases where they are incorrect are handled by the "Rejecting" stage. This is actually a key benefit of the "Rejecting" stage, in that it allows "Fitting" to be more permissive in ambiguous situations.

MC35. Added a line explaining this to the caption and as motivation for the "Rejecting" phase.

RC36. **Line 257:** Clarify if this slope condition is equivalent to ensuring that no two simultaneous, spatially-separated contrail detections can be attributed to the same generating flight?

AC36. We are assuming the reviewer is referring to line 275 here. The slope condition does not prevent two simultaneous, spatially-separated contrail detections from being attributed to the

same generating flight. As the text explains, its goal is to provide an upper-bound on the reasonable rate of W growth per satellite frame, since if it were unbounded, we would frequently end up with fits that span multiple physical contrails in regions with a high density of short-lived contrails.

It is completely allowed, and sometimes desirable for two simultaneous, spatially-separated contrail detections to be attributed to the same flight, even for the same group of overlapping waypoints. A common example is a contrail that has split in two, lengthwise (either due to detection error or due to physical processes). This is in fact part of the reason for allowing the residual ($T_{res}$) when computing the inliers for a fit, since a contrail that is detected as a singular contrail in one frame and splits into two in the next frame would end up with attributions $\{(Age_1, W_1), (Age_2-d, W_2), (Age_2+d, W_2)\}$, where d is a small time delta, because the implied ages of the split contrails would be slightly different, but we still want to capture both in the fit. We have added a brief discussion of this where the residual term is introduced.

MC36. Added brief discussion of handling contrails that split as they age.

RC37. Line 280: More meaningful notation than m and b could be used. Clarify if m and dW/dt are the same quantity.

MC37. Replaced m with dW/dt and b with $W_{t=0}$.

RC38. Line 305: clarify that 'each flight' is 'each group of flight waypoints with overlapping single-frame attributed contrails'.

MC38. Clarified that this is the group.

RC39. Fig. 6: Does Fig. 3 not highlight a similarly rejectable scenario for contrail 1 as matched to flight 2? If so, this figure may not be necessary.

MC39. Dropped Fig 6 and updated "Rejection" discussion to reference Fig 3 instead.

RC40.  **Line 307**: What is the impact of including Ssc to this score? Although enhancing confidence in this being a single contrail, shallow slopes reduce the confidence in the 'low W-intercept' condition.

AC40. This was a term that the tuning phase had the option to set to 0, but did not. Given that it's black-box tuning it's hard to know exactly why it did that. We speculate that it might come into play in scenes with a large number of short-lived contrails (maybe similar to Fig 4(g)), where the supremacy of the "number of inliers" term during the fit generation phase might still produce a fit with a relatively large slope that joins detections of many different physical contrails. This term allows the $S_{fit}$ for such fits to be high, and then ideally be rejected in the "Rejecting" phase.

MC40. Added this speculation to the text.

RC41. **Line 340:** If scalability is critical, it would be good to clarify if this approach is any more scalable than the algorithms mentioned in the introduction, specifically Geraedts et al. (2024).

AC41. CoAtSAC is comparably scalable to Geraedts et al. (2024), in the sense that it can be parallelized in exactly the same way. It is, however, going to be strictly slower, as it for the most part runs the Geraedts et al algorithm and then adds post-processing. The scalability claims are intended to be in comparison to the tracking-based approaches like Chevallier et al. (2023) and to manual attribution approaches.

MC41. Clarified that the scalability of CoAtSAC is comparable to Geraedts et al (2024).

RC42. Lines 344–345: Clarify if any other attribution attempts propose such a methodology.

MC42. Specifically referenced Chevallier 2023 here.

**Synthetic contrail benchmark dataset**

RC43. Line 347: The value of the benchmark dataset goes beyond tuning the hyperparameters—the segue could be removed.

MC43. Replaced the first 2 sentences of the paragraph with "Ideally we would use a dataset of ground-truth contrail attributions in geostationary imagery to tune and evaluate our attribution algorithm."

RC44. **Line 364:** The name 'SynthOpenContrails' is chosen to parallel the 'OpenContrails' dataset of Ng et al. (2023). Make explicit that the new dataset is not suitable for benchmarking contrail detection algorithms, only attribution algorithms, and that a performant contrail detection is assumed.

MC44. Added the text: "While the resulting dataset takes the form of contrail labels corresponding to satellite imagery, due to the aforementioned caveats it is not suitable for training contrail detection models, and is intended only for use in contrail attribution algorithms, where the labels need not align with actual satellite radiances."

RC45. Line 374–375: Clarify 'flight loading purposes'.

MC45. Replaced with "When selecting the candidate flights to form the synthetic contrails"

RC46. Line 380: Clarify 'weather data to use' for what.

MC46. Replaced with "weather data that will be used to determine synthetic contrail formation, dynamics, and evolution from the candidate flights"

**Tuning and Benchmarking**

RC47. Lines: 440–442: This introduction is not necessary.

MC47. Removed.

RC48. **Line 455:** Could whether or not it is the case that multiple sets of parameters match the real data be specified with more confidence? Is there any motivation to the specific set chosen?

AC48. It's hard to be more confident about this, both because the tuning is rather expensive, and because the qualitative comparisons are necessarily subjective. We did, however, add a suggestion for how one might adjust the parameters to get a comparably good match.

MC48. Added suggestion for how to adjust parameters to get a comparably good match.

RC49. Line 459: The content of Appendix A21 would be valuable in the body of the manuscript. Comment on whether the train split was used for manual tuning, and why a better match could not be obtained in terms of contrail pixels and linear contrails.

AC49. We have inlined most of Appendix 21 in the main text.

As stated on line 454, the validation set was used for tuning the dataset parameters. This was largely due to the fact that the train set contains about 5 times as many satellite frames as the validation set, and the dataset is slow to regenerate and scales roughly with the number of frames being generated. We were less concerned about overfitting in this process, as the tuning process was entirely manual and the parameter values have physical meanings, but we agree that the more rigorous approach would have been to use a subset of the training set.

Appendix 21 (now the main text) includes a discussion of the imperfect match in contrail pixel and linear contrail counts. It hypothesizes that the bluntness of the approach to contrail outbreak handling makes getting a better match difficult, and that a more refined approach is likely to improve this substantially, but requires further research.

MC49. Inlined most of Appendix A21 in the main text.

RC50. **Line 453:** Specify what it means for the metrics to be computed globally, the region of study is the Contiguous US.

AC50. We are assuming that this was referring to line 473. The intent here was the other definition of "global," meaning that it is computed uniformly over the entire dataset. As we have switched away from having the two versions of the metric, this term is no longer used.

RC51. Line 474: Fig. A6 might be valuable to include in the manuscript body. Suggest that this is then used to justify calculating per-frame quantities when results are introduced rather than here, to avoid 'teasing' the results.

MC51. Fig A6 is now in the main text, before this line.

RC52. Lines 481–483: Clarify this. It might be better placed as part of an applicability discussion after the results are presented.

MC52. Added the following clarifying text: "This affects the case where a given flight formed one or more contrails according to CoCiP, but, due to the dataset's post-processing steps, no detections of its contrails ended up in SynthOpenContrails. If an attribution algorithm were to attribute a synthetic detection to this flight, it would hurt the per-flight precision and not increase its per-flight recall."

RC53. Lines 485–489: This is a repeated introduction in abstract, and isn't necessary.

AC53. This is the first place where it is spelled out how one should use the synthetic dataset to tune or benchmark an attribution algorithm. We are not certain what part of the paper this is repeating.

RC54. Line 498: The results of the sensitivity could be very briefly summarised here. Given the relatively straightforward outcome, the appendix could be removed. If the contents is kept, specify which precision is meant for the result of the single-frame algorithm (line 941)

MC54. Added summary of sensitivity analysis in the main text body. Clarified that both precision metrics behave the same way. Removed the appendix.

**Results**

RC55. **Section 3.2**: Although performance relative to each other is approximately constant, the relative performance between bins of different contrail properties is interesting and could be expanded by including some of Appendix A26—which includes significant analysis and discussion beyond the contents of the manuscript's body. The discussions of contrail density and altitude are especially important. Given the error in meteorological data is hypotehsised as having significant impact, could this be resolved? The discussion of contrail age and width is also valuable—does this speak to limitations of using linear objects as contrails?

AC55. We are assuming that the reviewer intended to write Appendix A27 instead of A26. We have now incorporated the most significant sections of that appendix to the main text.

We're unsure exactly which significant impact of meteorological data error the reviewer is referring to. We'll note that there is much ongoing research aiming to improve the

meteorological data as it relates to contrails. One such example is Wang (2025). This work is beyond the scope of our current study, however.

Wang, Z., Bugliaro, L., Gierens, K., Hegglin, M. I., Rohs, S., Petzold, A., Kaufmann, S., and Voigt, C.: Machine learning for improvement of upper-tropospheric relative humidity in ERA5 weather model data, Atmos. Chem. Phys., 25, 2845–2861, https://doi.org/10.5194/acp-25-2845-2025, 2025.

MC55. Added a suggestion that there may be benefits to moving away from linearized contrail detection. Added portions of Appendix A27 to the main text.

**Technical corrections**

RC56. In Algorithm 1, using e.g. italicised text rather than mathematics text may improve legibility.

AC56. As per the suggestion of Reviewer 2, we replaced Algorithm 1 with a flow diagram.

RC57. Line 431: 'subrouting' should read 'subroutine'.

MC57. Fixed typo.

**References**

Geraedts, Scott et al. (2024). "A Scalable System to Measure Contrail Formation on a Per-Flight Basis". In: Environmental Research Communications 6.1, p. 015008. issn: 2515-7620. doi: 10.1088/2515-7620/ad11ab.

Gryspeerdt, Edward et al. (2024). "Operational Differences Lead to Longer Lifetimes of Satellite Detectable Contrails from More Fuel Efficient Aircraft". In: Environmental Research Letters 19.8, p. 084059. issn: 1748-9326. doi: 10.1088/1748- 9326/ad5b78.

Ng, Joe Yue-Hei et al. (2023). OpenContrails: Benchmarking Contrail Detection on GOES-16 ABI. http://arxiv.org/abs/ 2304.02122. arXiv: 2304.02122 [cs].

Schumann, U. (2012). "A Contrail Cirrus Prediction Model". In: Geoscientific Model Development 5.3, pp. 543–580. issn: 1991-9603. doi: 10.5194/gmd-5-543-2012.

Teoh, Roger et al. (2020). "Mitigating the Climate Forcing of Aircraft Contrails by Small-Scale Diversions and Technology Adoption". In: Environmental Science & Technology 54.5, pp. 2941–2950. issn: 0013-936X. doi: 10 . 1021 / acs . est . 9b05608.

---

## Referee Report (RR1)

Thanks to the authors for their thoughtful and productive response, and the changes made. The revised manuscript offers a significant improvement over the original, and the changes made have substantially improved the clarity, making for an enjoyable read.

Two previously raised issues persist, which can be rectified with language changes alone:

- RC3: The manuscript still implies the independence of the ensemble control and operational ERA5 data. These datasets have the same underlying model, so will exhibit shared systematic biases. The appendix analysis only establishes the distribution of the errors is similar, but doesn't establish their independence.
  Hence, the benchmark is not representative of how different attribution algorithms perform when faced with errors in advection-critical quantities in ERA5 data.
  This only needs acknowledging in relevant portions of the text, specifically:
    - Sect. 2.1.2
    - Lines 118-119 (where that advection error characteristics are said to be realistic – flaws of contrail model and of the meteorological data upstream are also baked in)
    - Line 238: "real weather" minimises these errors.
  This feature of the benchmarking is important for this work, because the character of advection error is the basis for CoAtSaC, so a simpler advection error could feasibly (though not necessarily) suit it particularly well, and cause it to benchmark better. Therefore, this limitation should also be emphasised with the presentation of relative benchmarks in Sect. 4.1 (around line 614).
  Along similar lines, the limitation of attribution when using pixel values (line 286) ought to be emphasised in section 4.1 as it is relevant for comparing these algorithms.
- RC13: The revised version still doesn't accurately reflect the approach of the modelling studies (in line 23). This is representative of some flaws in the logical flow of the introduction.
  The model studies do scale the humidity in order to correct the frequency of ISSR occurrence and the RHi distribution within them. Therefore, it is not true that the studies "do not account for inaccuracies therein" (line 28).
  However, it is true that the model studies don't use a humidity field that is correct in terms of e.g. the spatial distribution of ISSRs. Therefore, they establish a representative population of contrails (which are a suitable basis for their conclusions as listed in the manuscript), but fail in predicting the occurrence of specific individual contrails (which would be required to compare observations and model output).
  As a result, the statement "the benefits achievable by performing contrail avoidance with existing weather forecasts by differ from those quantified" doesn't follow – making these conclusions doesn't need individual matches. Further, line 32 seems to imply that these issues could be resolved with observations, but it is the deficiency in weather data that restricts model/observation comparisons. Perhaps observations would be better presented as an alternative impact assessment method, as long as attributions to flights can be made.

And some technical points/clarifications:

- Line 90: "for comparing a contrail forecast model to satellite observations". Perhaps change this to "for comparing properties of generating aircraft to satellite observations", to avoid the need to relitigate whether a contrail forecast model produces a population that is comparable on an individual basis.
- Line 161: 2 waypoints per pixel doesn't follow immediately unless a value is stated for $C_{Tflight}$. It would be clearest to state it here as well as having it in the table.
- Line 221: Clarify if the smoothing imposed is a spatial Gaussian blur (if so, standard deviation of 1 in which distance units), or a Gaussian noise in opacity (in which case, s.d. of 1 seems large, is this justified?)
- Sect. 4.2.2: This section, moved from an appendix, is a little dense (especially lines 660-694). Perhaps Fig. 13 could be removed. Furthermore, it is not clear from Fig. 12b that the altitude dependence is very strong, especially for the precision.

---

## Author Response (AR2)

We thank the reviewer for their further efforts in ensuring the correctness of our work. We believe that we have addressed all of the issues raised.

> Thanks to the authors for their thoughtful and productive response, and the changes made. The revised manuscript offers a significant improvement over the original, and the changes made have substantially improved the clarity, making for an enjoyable read.
>
> Two previously raised issues persist, which can be rectified with language changes alone:
>
> RC3: The manuscript still implies the independence of the ensemble control and operational ERA5 data. These datasets have the same underlying model, so will exhibit shared systematic biases. The appendix analysis only establishes the distribution of the errors is similar, but doesn't establish their independence.
>
> Hence, the benchmark is not representative of how different attribution algorithms perform when faced with errors in advection-critical quantities in ERA5 data.
>
> This only needs acknowledging in relevant portions of the text, specifically:
> - Sect. 2.1.2
> - Lines 118-119 (where that advection error characteristics are said to be realistic - flaws of contrail model and of the meteorological data upstream are also baked in)
> - Line 238: "real weather" minimises these errors.

We have audited the manuscript to ensure that the claims about advection error of ERA5 vs. the synthetic contrails being similar to that of ERA5 vs. real contrail detections mention these potential biases and do not overstate their realism.

> This feature of the benchmarking is important for this work, because the character of advection error is the basis for CoAtSaC, so a simpler advection error could feasibly (though not necessarily) suit it particularly well, and cause it to benchmark better. Therefore, this limitation should also be emphasised with the presentation of relative benchmarks in Sect. 4.1 (around line 614).

Done

Along similar lines, the limitation of attribution when using pixel values (line 286) ought to be emphasised in section 4.1 as it is relevant for comparing these algorithms.

Done

RC13: The revised version still doesn't accurately reflect the approach of the modelling studies (in line 23). This is representative of some flaws in the logical flow of the introduction.

The model studies do scale the humidity in order to correct the frequency of ISSR occurrence and the RHi distribution within them. Therefore, it is not true that the studies "do not account for inaccuracies therein" (line 28).

However, it is true that the model studies don't use a humidity field that is correct in terms of e.g. the spatial distribution of ISSRs. Therefore, they establish a representative population of contrails (which are a suitable basis for their conclusions as listed in the manuscript), but fail in predicting the occurrence of specific individual contrails (which would be required to compare observations and model output).

As a result, the statement "the benefits achievable by performing contrail avoidance with existing weather forecasts by differ from those quantified" doesn't follow - making these conclusions doesn't need individual matches. Further, line 32 seems to imply that these issues could be resolved with observations, but it is the deficiency in weather data that restricts model/observation comparisons. Perhaps observations would be better presented as an alternative impact assessment method, as long as attributions to flights can be made.

We have reworked this section to focus less on evaluation and improvement of prediction models, which is not the direct subject of this study, and instead focus on using observations to evaluate operational contrail avoidance. We believe that the reworked text should address the concerns raised here.

And some technical points/clarifications:
- Line 90: "for comparing a contrail forecast model to satellite observations". Perhaps change this to "for comparing properties of generating aircraft to satellite observations", to avoid the need to

relitigate whether a contrail forecast model produces a population that is comparable on an individual basis.

We reworded to "comparing the per-flight predictions of a contrail forecast model to satellite observations, for the purposes of assessing the model's utility for operational avoidance."

- ○ Line 161: 2 waypoints per pixel doesn't follow immediately unless a value is stated for C_Tflight. It would be clearest to state it here as well as having it in the table.

Done

- ○ Line 221: Clarify if the smoothing imposed is a spatial Gaussian blur (if so, standard deviation of 1 in which distance units), or a Gaussian noise in opacity (in which case, s.d. of 1 seems large, is this justified?)

Clarified that this is a spatial Gaussian blur with a standard deviation of 1 pixel.

- ○ Sect. 4.2.2: This section, moved from an appendix, is a little dense (especially lines 660-694). Perhaps Fig. 13 could be removed. Furthermore, it is not clear from Fig. 12b that the altitude dependence is very strong, especially for the precision.

We have moved Fig 13 into an appendix along with all of the discussion about it, and replaced it with a short summary of the key findings. We also softened the language regarding the trends observed in Fig. 12b.